# Streamlined single-cell proteomics by an integrated microfluidic chip and data-independent acquisition mass spectrometry

Sofani Tafesse Gebreyesus [1,2,3,9], Asad Ali Siyal [1,4,5,9], Reta Birhanu Kitata [1], Eric Sheng-Wen Chen[1], Bayarmaa Enkhbayar[4,6], Takashi Angata [6], Kuo-I Lin [7], Yu-Ju Chen [1,3,4,8 ✉] & Hsiung-Lin Tu [1,2,4,8 ✉]

Single-cell proteomics can reveal cellular phenotypic heterogeneity and cell-specific functional networks underlying biological processes. Here, we present a streamlined workflow combining microfluidic chips for all-in-one proteomic sample preparation and data-independent acquisition (DIA) mass spectrometry (MS) for proteomic analysis down to the single-cell level. The proteomics chips enable multiplexed and automated cell isolation/counting/imaging and sample processing in a single device. Combining chip-based sample handling with DIA-MS using project-specific mass spectral libraries, we profile on average ~1,500 protein groups across 20 single mammalian cells. Applying the chip-DIA workflow to profile the proteomes of adherent and non-adherent malignant cells, we cover a dynamic range of 5 orders of magnitude with good reproducibility and <16% missing values between runs. Taken together, the chip-DIA workflow offers all-in-one cell characterization, analytical sensitivity and robustness, and the option to add additional functionalities in the future, thus providing a basis for advanced single-cell proteomics applications.

[1] Institute of Chemistry, Academia Sinica, Taipei 11529, Taiwan. [2] Nano Science and Technology Program, Taiwan International Graduate Program, Academia Sinica, Taipei 11529, Taiwan. [3] Department of Chemistry, National Taiwan University, Taipei 10617, Taiwan. [4] Chemical Biology and Molecular Biophysics Program, Taiwan International Graduate Program, Academia Sinica, Taipei 11529, Taiwan. [5] Department of Chemistry, National Tsing Hua University, Hsinchu 30013, Taiwan. [6] Institute of Biological Chemistry, Academia Sinica, Taipei 11529, Taiwan. [7] Genomics Research Center, Academia Sinica, Taipei 11529, Taiwan. [8] Genome and Systems Biology Degree Program, Academia Sinica and National Taiwan University, Taipei 10617, Taiwan. [9] These authors contributed equally: Sofani Tafesse Gebreyesus, Asad Ali Siyal. ✉email: yujuchen@gate.sinica.edu.tw; hltu@gate.sinica.edu.tw

Rapidly developing single-cell omics-based molecular measurements have revolutionized modern biological research[1,2]. As proteins are functional workhorses of the cell, proteomic profiling provides a direct snapshot of the dynamic biological network to complement the genomics and transcriptomics architecture[3]. However, the sensitivity of proteomic profiling is limited due to the wide dynamic range of proteome constituents and the lack of a viable protein amplification strategy[4]. Targeted protein analyses have enabled sensitivity down to the single-cell level, but their multiplexity is often limited and depends on antibody availability and quality[5–8]. Mass spectrometry (MS)-based proteomic approaches offer label-free analysis with high specificity and deep proteomic coverage, which has been shown in several studies to reach single-cell sensitivity[8–14]. However, multistep processing in most traditional MS workflows often results in significant sample loss, linking trade-offs between high proteome coverage and accessible sample size[8,15].

Microproteomics workflows aimed at handling minute samples have been widely developed to expand MS-based proteomic analysis toward limited input samples (<1000 cells)[16]. For example, filter-aided sample preparation (FASP), inStage-Tip (iST), integrated proteome analysis device (iPAD), and single-pot solid-phase-enhanced sample preparation (SP3) reported protocols that combine cell lysis, protein digestion, and/or detergent removal to improve proteome identification at the level of a few hundred cells[17–20]. Alternatively, sample preparations on nanoliter droplets have been developed to enhance proteome profiling sensitivity, including oil-air droplet (OAD) and digital microfluidic (DMF-SP3) chips; these methods effectively reduced adsorptive loss and identified 1063 and 2500 proteins from 100 and 500 cells, respectively[21,22]. Similarly, the nanodroplet processing platform (nanoPOTS) achieved proteome coverage of over 3000 proteins from 10 to 100 cells by incorporating the match-between-runs (MBR) feature[23]. Its recent extension with ultralow-flow nanoLC and high-field asymmetric ion mobility spectrometry (FAIMS) coupled to the Orbitrap Eclipse instrument reported sensitive profiling of 1475 protein groups with MBR from a single cell[13]. Incorporation of isotopic labeling to extend the multiplexity ability was demonstrated at the level of 1000 proteins in primary cells[12,24]. With these advances, nonetheless, a fully automated workflow, starting from multiplexed cell capturing and imaging, cell lysis, and protein digestion to peptide desalting, all integrated within a single device to realize proteomic analysis for low-input samples has not yet been established, despite the prospect to substantially minimize sample loss and achieve high reproducibility and sensitivity.

Microfluidic devices based on multilayer soft lithography use custom chip integration and hydraulic actuations to achieve precise μL-to-nL fluid manipulation and are ideal platforms to execute a complex protocol[25–27]. However, microfluidics has not been explored for streamlined proteomics workflows primarily due to challenges associated with reagent compatibility for one-pot protocols, concerns of mixing in confined space, and overall system integration. To study the cellular or phenotypic status, sufficient proteome coverage is critical and may require higher cell numbers due to relatively limited proteome coverages at the single-cell level. Therefore, in this study, chips with different cell capacities were constructed to facilitate experiments with optimal profiling depth for different cell inputs. Specifically, an integrated proteomics chip (iProChip, 1–100 cells) and its extended version for single-cell capacity (SciProChip) were designed and coupled with data-independent acquisition (DIA) MS as streamlined nanoproteomics (nanogram of cells) pipelines. These chips are designed as automated stations for the entire proteomic workflow, offering built-in features including quantifiable cell capture

and imaging, complete cell lysis, protein digestion, and peptide desalting. In situ cell counting allows quantification of the number of captured cells. Thus, the cell number can be variable to achieve proteome coverage of interest. Following chip processing, we showed that DIA MS, which detects all precursors and fragments in the entire $m/z$ range within isolation windows, enabled all retrospective peptide mapping against spectral libraries and offered 2.3-fold superior coverage than conventional data-dependent acquisition (DDA) mode[28]. Importantly, the SciProChip-DIA workflow characterized 1500 ± 131 protein groups from individual single cells with 1% false discovery rate (FDR). Furthermore, the analytical performance and versatility of iProChip-DIA were demonstrated using both human adenocarcinoma cells (PC-9) and chronic B cell leukemia cells (MEC-1), whose size differences were readily quantified using the built-in cell imaging feature. The results revealed a performance of 5 orders of proteome coverage, >100-fold quantification range, good reproducibility (Pearson correlation of 0.88–0.98) and low between-run missing values (<16%). The presented workflow illustrates a unique implementation of microfluidic devices with all-in-one functionality to achieve automated and streamlined proteomic preparation, which offers high sensitivity and reproducibility for limited input samples, including a single cell.

## Results

**Design and characterization of the iProChip and streamlined microproteomics workflow.** To provide a streamlined microproteomic pipeline for mass-limited samples, we designed a microfluidic device as an integrated proteomics chip (iProChip) to offer all-in-one functionality from cell input to complete proteomic sample processing. The iProChip has a two-layer, push-up geometry and allows accurate fluid manipulation via 34 valves controlled by a custom program, thereby offering an automated protocol for precise and systematic control (Fig. 1a–f and Supplementary Fig. 1a)[29]. The chip is composed of 9 units to enable multiplexed proteomic experiments running in parallel. Each unit contains a cell capture, imaging and lysis chamber, a protein reduction, alkylation and digestion vessel, and a peptide desalting column (Fig. 1c and Supplementary Fig. 1b). All units share 9 inlets and 2 outlets, allowing programmed delivery of reagents and simultaneous sample processing to increase assay throughput. The cell trap is made up of arrays of 10, 50 and 100 wedge-shaped twin pillars spaced by 5 μm for rapid size-based cell capture in 5, 8.5, and 11 nL chambers, respectively (Fig. 1e)[30]. A circular chamber with a radius of 1 mm and a height of 100 μm (312 nL) was fabricated to accommodate the entire proteomic workflow, including cell lysis, protein reduction, alkylation and digestion in a single step (Fig. 1f and Supplementary Fig. 2). Note that the calculated surface-to-volume ratio for iProChip is larger than that of existing single-cell devices, such as nanoPOTS, yet it still exhibits a substantially reduced surface area by >90% in comparison to the microscale vial-based workflow (Supplementary Table 1)[11]. For peptide desalting, a 2.5 cm-long column with a cross-section of 200 μm × 25 μm was fabricated by packing reversed-phase C18 beads into the microchannel prepatterned with 5 μm filters to perform on-chip clean-up of digested peptides (Fig. 1e, f and Supplementary Movie 1). To increase proteome coverage, we applied a deep single-shot profiling strategy that integrates direct- and library-based DIA analysis using an Orbitrap mass spectrometer. We developed a spectral library resource complementarily established by hybrid DDA-DIA datasets using either cancer cell lines or immune cells consisting of different proteome compositions, which can serve as a digital map to theoretically recover all peptides in the $m/z$ and retention time domains of DIA data (Fig. 1g). Specifically,

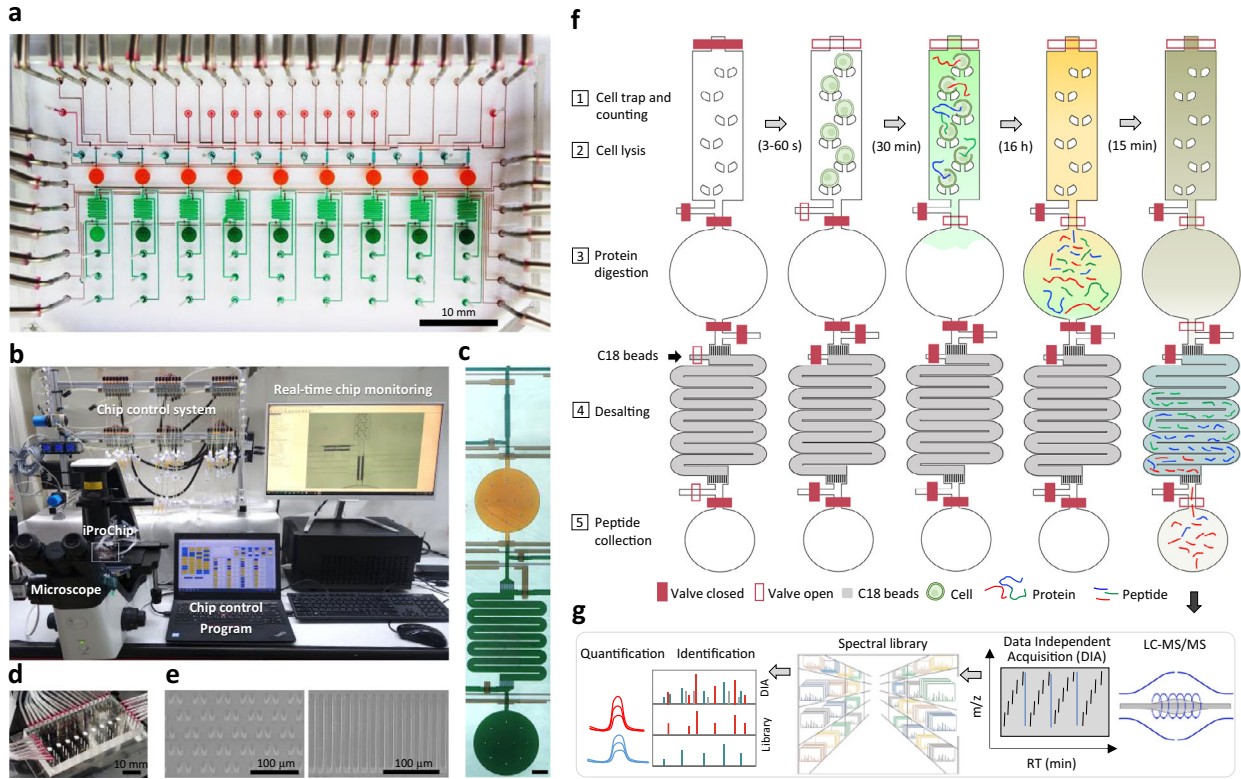

**Fig. 1 Schematics of the integrated proteomics chip and streamlined workflow for nanoproteomics. a** A bright-field image of the integrated proteomics chip (iProChip), where cell capture chambers (cyan), reaction vessels (orange), on-chip SPE columns (green), sample collection ports (dark green), and control layers (brown) are shown. **b** The entire system set-up for iProChip operation. **c** A close-up view of a single operation unit. Scale bar: 300 μm. **d** A ready-to-use iProChip mounted on the microscope. **e** SEM images of cell capturing pillars (left) and C18 filters in the SPE column (right). These images are representative of two chips that were observed using SEM. **f** Operational procedures of iProChip for streamlined sample preparation, including (1) cell trapping, imaging, and counting, (2) cell lysis, (3) protein digestion, (4) desalting, and (5) peptide collection. **g** Proteomic analysis using data-independent acquisition-based liquid chromatography–tandem mass spectrometry (LC-MS/MS) and spectral library search.

spectral libraries constructed from cell lines with different cell numbers were tested and optimized to maximize the number of identified and quantified proteins.

In the first step of the streamlined workflow, the cell trapping efficiency was determined using non-small-cell lung cancer (NSCLC) PC-9 cells ("Methods"). Using optimal cell density (500 cells/μL), desired numbers of cells (1–100) for each unit can be trapped in 10–60 s. The average percentage of cells captured from traps containing a single cell were 100, 92 ± 3, and 89 ± 8% for chambers with 10, 50, and 100 traps, respectively. The targeted capture efficiency for all units reached ~100% after counting traps containing 1 (~90%) and 2 or 3 cells (~10%), establishing it as an absolute quantifiable module to perform simple and fast size-based cell isolation (Fig. 2a, b, "Methods," and Supplementary Movie 2). Compared to external stand-alone cell sorters, such a built-in module offers simple, rapid, and efficient cell isolation. Additionally, we also showed that by using a lower cell density (25 cells/μL) operated at 3 psi, such cell chambers allow precise capture of lower numbers of cells at the level of 1 and 5 cells (Supplementary Fig. 3). The cell trapping capability of iProChip was also evaluated to characterize the cell usage efficiency (defined as numbers of trapped cells/numbers of total injected cells) and minimum numbers of cells needed for iProChip operation ("Methods"). Using a total of either 5 or 10 μL cell solution (25 cells/μL), the results showed that cell usage efficiency ranged from ~4 to 44% for capturing 1–100 cells (Supplementary Table 2 and Supplementary Fig. 4). Next, we sought to characterize whether reagents can mix efficiently in the

closed vessel during cell lysis and protein digestion. Three mixing approaches, including vortexing, shaking (by a plate shaker), and passive diffusion, were tested (Supplementary Fig. 5). Using imaging analysis, the relative mixing index (RMI) was calculated to assess the mixing performance ("Methods")[31]. The results showed that it took 11, 16, and 30 min for vortexing, shaking, and diffusion-mixing to reach 75% RMI, indicating that all three mixing strategies were sufficient to accommodate reactions within minutes to hours of reaction kinetics, which fit the timescale of conducting the proteomics workflow (Fig. 2c, d and Supplementary Fig. 5). Although vortex mixing was found to provide faster mixing, mixing by shaking was used in subsequent experiments due to its flexibility in handling and sufficient reaction timescale.

Another integration to the miniaturized device is the on-chip peptide desalting module (Fig. 2e and "Methods"). The effectiveness of desalting was evaluated by processing ~10 cells, where the profile of the desalted sample showed reproducible typical peptide ion profiles, while the nondesalted sample showed the predominant presence of detergent peaks (Supplementary Fig. 6). The loading capacity and peptide recovery (%) of the desalting module showed a linear correlation from 0.125 to 1 μg with ~89% recovery (Fig. 2f and "Methods"). Assuming that a typical mammalian cell contains 200 pg proteins[17], the capacity of the desalting column is thus anticipated to capture peptides from approximately 4000 cells. Furthermore, the concern of compromised sample retrieval due to preferential flow was evaluated by flowing a colored dye through the C18 bead-packed column. The results showed that 9 psi was the minimal flow pressure to

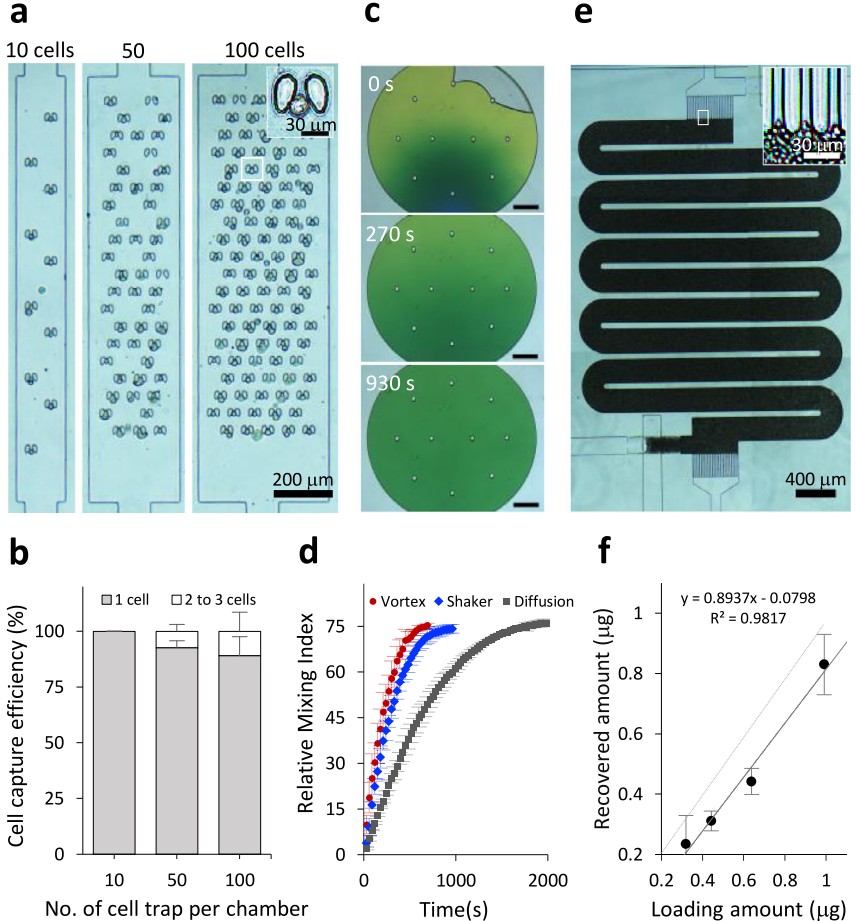

**Fig. 2 Performance characterization of the iProChip. a** Bright-field images of non-small lung cancer PC-9 cells captured in 10, 50, and 100 cell chambers. Top right: a zoom-in image of a trapped cell. **b** Characterization of the cell capture efficiency for separate capture chambers. Data are presented as mean values ± SD ($n = 3$ independent measurements). **c** Representative time-lapse images of a reaction vessel filled with green dye during mixing-by-shaking characterization. Scale bars: 300 μm. **d** Comparison of mixing efficiency in the reaction vessel. Data are presented as mean values ± SD ($n = 3$ independent experiments). **e** A bright-field image of the SPE column packed with C18 beads. Top right: a close-up view near the C18 filter. **f** Desalting recovery efficiency of the on-chip SPE column. Data are presented as mean values ± SD ($n = 3$ independent experiments). Source data are provided as a Source data file.

overcome the preferential flow, and 11 psi was used in the following workflow (Supplementary Fig. 7 and "Methods").

**Integration of iProChip with DIA MS**. To operate the iProChip for proteomic workflow, precise numbers of cells, including 1, 5, 10, 50, and 100 cells, were captured using built-in cell traps. Parallel processing of the cells trapped in capture chambers was performed by dispensing and incubating with cocktail buffer containing RapiGest, tris(2-carboxyethyl)phosphine hydrochloride (TCEP), and chloroacetamide (CAA), which was specifically adapted to achieve one-pot cell lysis, protein reduction, and alkylation to minimize sample loss (Supplementary Figs. 2 and 8 and Movie 3). Subsequent protein digestion and acidification were conducted in the reaction vessel, and digested peptides were then subjected to multiplexed desalting by passing through the C18 column for 15 min. Note that before each experiment, the cell chamber and reaction vessel were coated with bovine serum albumin (BSA) to minimize adsorptive losses of peptides. For subsequent liquid chromatography (LC)–MS/MS analysis, single-shot DIA-MS acquisition parameters, including isolation window, resolution, peptide amount, and LC–MS/MS gradient were optimized to enhance proteomic profiling coverage in low numbers of cells ("Methods")[32].

To allow deep profiling and enhanced identification of low abundance and cancer-relevant proteins by DIA, high-quality project-specific spectral libraries were constructed using lung cancer and human chronic lymphocytic leukemia (CLL) cell lines. The protein compositions and dynamic range may vary in bulk samples containing thousands of cells and a single cell, which likely affect the chromatographic time domain and DIA acquisition. Thus, we constructed both large-scale (1 μg) and small-scale (~10 cells) libraries from respective cell types, i.e., PC-9 and MEC-1 cells, and implemented them to analyze various numbers of cells. Specifically, the large-scale project-specific libraries of PC-9 and MEC-1 processed in bulk/dilution with DDA and DIA modes consisted of 6,345 protein groups (83,305 peptides) and 6261 protein groups (60,335 peptides) with 1% precursor and protein FDR, respectively. These large-scale libraries were used to analyze higher cells (i.e., >10 cells). For lower input samples (i.e., ~5 and single cells), we reasoned that a small-scale specific library should be beneficial for identification and quantification. To maximize the proteome profiling sensitivity at the single-cell level, we constructed small-scale spectral libraries from ~10 cells processed through iProChip as well as aliquots of ~1.5 ng (~10 cells) processed through bulk/dilution, yielding a depth of 2231 protein groups (14,054 peptides) and 2440 protein groups (11,720 peptides), respectively.

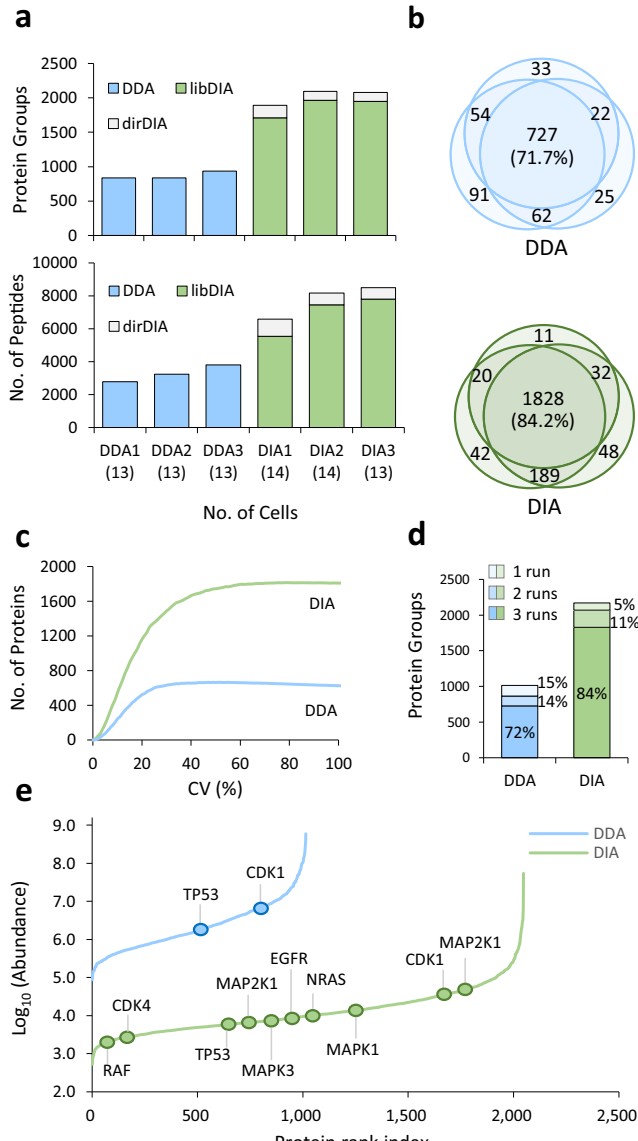

**Fig. 3 Comparison of identification coverage and quantitation performance of proteomic profiling of PC-9 cells by DIA and DDA methods. a** Comparison of protein groups and peptides identified by triplicate analysis using the data-dependent acquisition (DDA, light blue) and data-independent acquisition (DIA, light green) modes. Single-shot DIA was acquired and processed by both library-based DIA (libDIA) and direct DIA (dirDIA) approaches by Spectronaut. **b** Overlap of protein groups identified by DDA and DIA. **c** Distribution of coefficient of variation (CV%) for quantified protein groups by DDA and DIA. **d** Evaluation of missing values (%) of proteins identified and quantified in triplicate LC–MS/MS runs by DDA and DIA. **e** Assessment of dynamic range based on protein abundance rank and annotation of selected proteins related to lung cancer. Source data are provided as a Source data file.

To evaluate the performance of DIA-based quantitation for iProChip, analytical merits in sensitivity, proteome coverage, and reproducibility were systematically investigated by using iProChip to process 13–14 PC-9 cells and compared to the conventional DDA method (Fig. 3a). The identification and quantification analysis of all the datasets in this study were performed at a statistically stringent criterion of 1% FDR at the peptide-to-spectrum match (PSM) and protein levels. By the DDA method, an average of 869 protein groups (3280 peptides) were identified

in the triplicate analysis. In comparison, the direct DIA (dirDIA) approach using the Spectronaut tool identified 1409 protein groups (5174 peptides), whereas the library-assisted DIA (libDIA) approach using the large-scale PC-9 cell library showed significantly higher proteome coverage of 1874 protein groups (6929 peptides). Comparing the dirDIA and spectral library-based results, the superior quantitation of libDIA is likely due to more efficient detection of low-intensity peptide ions in DIA mode to match the corresponding peptide spectra in our library. By combining the complementary dirDIA and libDIA results, the overall identification coverage further increased to 2022 proteins (7757 peptides). The identification of 2.3-fold and 2.4-fold protein groups and peptides, respectively, by the DIA approach revealed its superior profiling coverage over the DDA approach (Fig. 3a). These results demonstrated that a single-shot DIA-based LC–MS/MS, complementarily processed by dirDIA and libDIA, improved the proteome identification coverage for the small-scale sample from the fully automated sample preparation in iProChip.

To evaluate the reproducibility of the iProChip-DIA workflow, we calculated the percentage of overlapping proteins between the triplicate analysis of 14 PC-9 cells. The results showed that 84.2% of the 2022 identified proteins and 71.7% of the 869 identified proteins were reproducibly detected by DIA and DDA, respectively, indicating higher reproducibility and proteome coverage with the iProChip-DIA approach (Fig. 3b). For evaluation of reproducible quantitation with CV ≤ 20%, DIA achieved significantly higher coverage of 1160 quantifiable proteins compared to 522 proteins quantified by DDA (Fig. 3c). Previous label-free quantification methods have commonly observed 10–50% between-run missing values, presenting a bottleneck for reproducible quantification across samples[33]. Additional comparison for run-to-run variabilities revealed fewer missing values, i.e., the number of proteins only quantified in one of triplicate runs, in the DIA result (16%) compared to that of DDA (28%) (Fig. 3d). Meanwhile, the wide dynamic range of proteome compositions presents another major bottleneck for deep profiling, especially for low-abundance proteins. Thus, we assessed the dynamic range based on the protein abundance rank. By DDA, the abundances of the 1014 identified proteins were found to span ~4 orders of magnitude, whereas 2170 identified proteins in DIA span ~5 orders of magnitude with coverage of important and low-abundance oncoproteins related to cancer. Notably, the FDA approved druggable targets for lung cancer, such as EGFR, MAP2K1, and MAP2K2, and proteins involved in the NSCLC pathway, including EGFR, NRAS, MAP2K1, MAP2K2, MAPK1, MAPK3, CDK4, and TP53, were readily identified in DIA, whereas only TP53 and CDK1 were detected in DDA using our approach (Fig. 3e)[34] (https://www.cancer.gov/about-cancer/treatment/drugs/lung). In summary, the DIA approach provides higher proteome profiling coverage, lower missing values, reproducible quantification, and a wider dynamic range than DDA at the level of 14 cells.

**Quantitative proteome profiling of mass-limited samples by iProChip and DIA-MS.** Next, we systematically evaluated the iProChip-DIA performance for proteomic profiling of PC-9 cells in chambers with 10, 50 and 100 traps, which may allow the experimental design of different sample inputs (Supplementary Data 1). As expected, the iProChip provided precise cell counting for each chamber to ensure unambiguous quantification (Fig. 4a). Combining libDIA and dirDIA analysis, on average, 4722 ± 10 protein groups (25,785 peptides) were identified from triplicate analysis from 106 ± 2 cells at 1% FDR. At chambers with lower numbers of cell traps, an average number of 3435 ± 262,

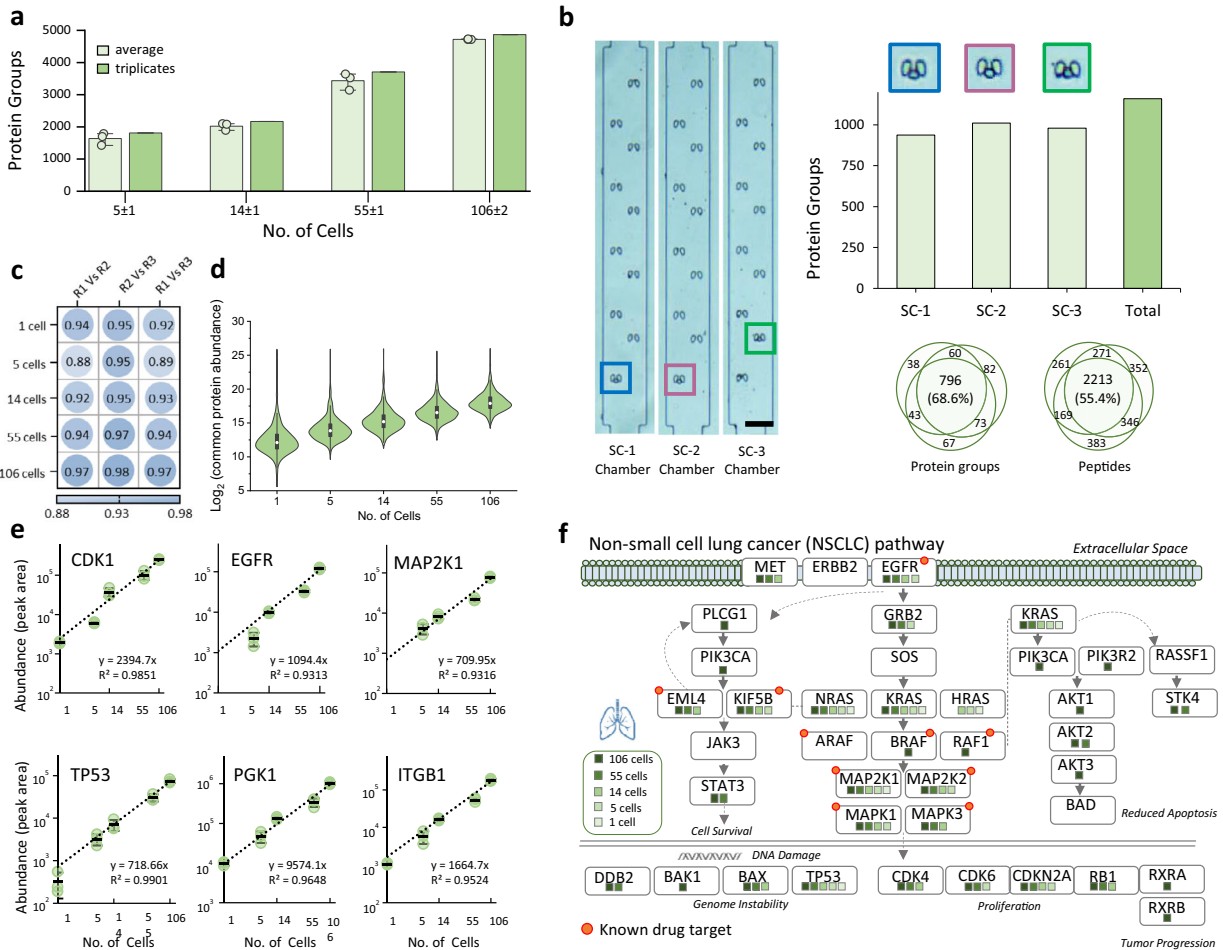

**Fig. 4 Analytical performance in sensitivity, reproducibility, and quantitation in proteomic profiling of a lower number of cells by iProChip-DIA. a** Identification summary of protein groups across different PC-9 cell numbers by iProChip-DIA. Data are presented as mean values ± SD ($n = 3$ independent cell samples for each condition). **b** Single PC-9 cell trapping using 10 cell capture chambers and corresponding cell image and triplicate analysis results of identified protein groups and coverage. SC: single cell; scale bar: 100 µm. **c** A heatmap showing the reproducibility of protein abundances obtained among different cell numbers and their replicates (R1–R3). **d** Distributions of total protein abundances in commonly identified proteins across different cell numbers. Each data was analyzed in $n = 3$ independent measurements. Center lines show the mean; box limits indicate the 25th and 75th percentiles; whiskers, 1.5× interquartile range. **e** Representative examples of lung cancer-related proteins showing quantitation of protein abundance calculated from peak area. Data are shown as mean ± SD from $n = 3$ independent measurements. **f** Identification coverage of proteins within the NSCLC pathway under different cell numbers using Kyoto Encyclopedia of Genes and Genomes (KEGG) database. Source data are provided as a Source data file.

2022 ± 114, and 1638 ± 191 protein groups were identified from 55 ± 1, 14 ± 1 and as low as 5 ± 1 cells, respectively (Fig. 4a and Supplementary Fig. 9). The overlap of identified protein groups in triplicate analyses was 77–93% from all cell numbers, illustrating the high reproducibility of the iProChip-DIA approach (Supplementary Fig. 10). Importantly, to characterize assay reproducibility across experiments, a separate batch of 10-cell samples was processed in an independent chip. The results showed a similar level of protein identification results in the two batches (Supplementary Fig. 11a), with 78–84% within-batch overlap and 71% between-batch overlap (Supplementary Fig. 11b–c), suggesting good reproducibility among separate experiments.

Encouraged by the promising sensitivity, we further pushed the profiling sensitivity at a single PC-9 cell using 10-cell capture chambers. An average of 976 ± 37 protein groups (3069 peptides) were identified from triplicates of a single PC-9 cell (Fig. 4b). The sum of triplicate measurements yielded the identification of 1160 protein groups (3995 peptides) from a single PC-9 cell. Comparing the identification coverage showed that 69% protein groups and 55% peptides were common among triplicate results.

To evaluate the background of the single-cell measurement, blank control samples (cell-free supernatant without trapped cells) were prepared for all sample preparation steps using iProChip and analyzed. Compared to the average of 88 proteins from similar studies[35], our results of an average of 58 proteins presented comparatively fewer proteins from the blank sample (Supplementary Fig. 12). Furthermore, the comparison showed a significant overlap between the 1-cell and 5-cell samples, both minimally overlapping with the blank sample. These results suggest low cross-contamination and false identifications.

To compare the iProChip with bulk sample processing, we directly processed ~100 cells in vial yet barely identified proteins with only an average of 86 protein groups (Supplementary Fig. 13). Comparison to dilution-based processing in vials (~50 µg lysate) was also performed by injecting small aliquots corresponding to 1.5 ng (~10 cells) and 15 ng (~100 cells) for DIA analysis ("Methods"). While the results showed comparable protein and peptide identification at 15 ng in both iProChip and bulk-dilution preparations, 2-fold more identifications were obtained from the iProChip workflow at 1.5 ng (Supplementary

Fig. 14). This higher performance gain of iProChip for smaller numbers of cells further validates the efficiency of our approach for limited cells. Additionally, when the data (1–106 cells) were compared to a conventional proteomics sample (~1 µg PC-9, ~6000 cells, >6000 proteins), 90–96% of the proteins identified from lower cells (1–106) using iProChip were common (Supplementary Fig. 15a). The fractions of the bulk proteins captured with iProChip showed an increasing trend correlated with cell numbers (Supplementary Fig. 15b). Interestingly, 75% of these 6000 proteins from 1 µg PC-9 cells were identified from 106 cells processed by iProChip, suggesting bulk-like comparable identification at a small sample scale. Further analysis of protein abundance showed that mainly abundant proteins were observed in lower cells, suggesting challenges of detecting low abundant proteins in low cell samples (Supplementary Fig. 15c).

To evaluate the analytical reproducibility of our approach for quantitative proteomics, the protein abundances in triplicate datasets were quantitatively compared by pairwise correlation analysis. The results showed good reproducibility (Pearson's correlation of 0.88–0.98) in the measured protein abundance for protein quantification by our iProChip-DIA workflow (Fig. 4c and Supplementary Fig. 16). To assess the quantitative performance, next, the distribution of overall protein abundances quantified in each cell number was calculated, which showed a log-linear correlation across different cell numbers (Fig. 4d). The capability of quantitative proteomic analysis was further evaluated at the individual protein level. Representative examples were selected from lung cancer-related oncoproteins, and their abundances were computed by Spectronaut[36]. The average protein abundances among representative lung cancer proteins, EGFR, CDK1, and MAP2K1, revealed good linearity between the measured protein abundance and increasing cell numbers (Fig. 4e). Most importantly, many quantified proteins, such as selected examples of TP53, ITGB1, PGK1, and MAPK1, showed good quantitative dependence (50-100-fold) between the protein abundance and cell number (1–106 cells) (Fig. 4e and Supplementary Fig. 17). Quantification at 50–100-fold magnitudes also demonstrated a wider dynamic range compared to conventional quantitative proteomic results on a bulk scale. In line with the aforementioned quantitative performances, iProChip-DIA enables a high degree of robustness, good reproducibility, and quantitative proteomic measurements down to the level of a single cell, a level of performance only achieved previously for ensemble measurements.

The identification of these proteins enabled us to map lung cancer-related signaling pathways searched against the Kyoto Encyclopedia of Genes and Genomes (KEGG) database[34]. A total of 329 pathways were enriched, such as the NSCLC pathway, metabolic pathways, pathways in cancer, spliceosome, viral carcinogenesis, proteoglycans in cancer, MAPK signaling, and apoptosis (Supplementary Fig. 18). The major lung cancer pathway, the NSCLC pathway, was enriched with coverage of a total of 29 proteins across different numbers of cells (Fig. 4f). Even at low cell numbers (14 ± 1 cells), 13 proteins, including the drug targets EGFR, MAP2K1, MAP2K2, MAPK1, MAPK3, and KIF5B, the tumor suppressor TP53, and other key signaling components (KRAS, CDK4, CDKN2A, EML4, KIF5B, NRAS, BAX, and RB1), were identified (Fig. 4f). In terms of sensitivity, EGFR, MAPK1, MAP2K1, MAP2K2, CDKN2A, TP53, KIF5B, and GRB2 proteins were still detected in as few as 5 cells, whereas MAP2K1, KRAS, and TP53 were even identified at the single-cell level (Fig. 4f). The confidence of protein identification for the above representative proteins is shown in their MS/MS spectra (Supplementary Fig. 19 and Supplementary Table 3). Based on a lung cancer model study, these results reveal the capability of the developed approach to provide protein coverage to study the cancer proteome and a wide range of cellular pathways at limited cell numbers.

**Application of iProChip-DIA for single leukemia cell proteomic profiling.** The general applicability of our iProChip-DIA platform was next demonstrated on the human B-CLL cell line MEC-1. From the perspective of methodology development, leukemia cells representing a heterogeneous cancer type are ideal models for developing sensitive proteomics tools, as they could readily complement various existing methods by delineating the system-wide profiles of phenotypic functionality[37]. When processing MEC-1 cells on-chip, the imaging-based cell trapping feature of iProChip revealed that MEC-1 cells were noticeably smaller than PC-9 cells (Supplementary Fig. 20). Combining dirDIA and libDIA using the MEC-1 spectral library, triplicate analyses of MEC-1 cells by iProChip-DIA analysis yielded averages of $3811 ± 362$, $931 ± 72$, and $455 ± 98$ protein groups from $117 ± 1$, $14 ± 1$ cells, and a single cell at 1% FDR, respectively (Fig. 5a, Supplementary Fig. 21, and Supplementary Data 2). The protein abundance was found to span ~5 orders of magnitude across different cell numbers, allowing the detection of important B cell surface markers CD20 and HLA molecules from as little as a single cell (Fig. 5b), while other key proteins, including CD19, CD22, CD47, and CD74, were identified from 14 and 117 cells (Fig. 5b). Functional annotation using UniProt showed that many proteins related to adaptive immunity, innate immunity, kinases, phosphatases, and Ig domains were identified from the single MEC-1 cell dataset, where the depth of protein coverage positively correlated with the cell number (Fig. 5c). By mapping 518 human kinases deposited in KinMap[38], 114 protein kinases were identified across all major branches of the kinase phylogenetic tree, such as tyrosine kinase (TK), TK-like kinases, serine/threonine protein kinases, casein kinase 1 (CDK1), and $Ca^{2+}$/calmodulin-dependent protein kinase (Fig. 5d). It was also noted that although MEC-1 cells were smaller than PC-9 cells, protein identification achieved good coverage and overlap (61–81%) using the iProChip-DIA approach, suggesting the versatility and robustness of our platform for different cell types (Supplementary Figs. 9c and 10b).

B cell receptor (BCR) signaling is crucial for mounting efficient adaptive immunity and is involved in the survival and growth of malignant B cells in B-CLL[39]. B cell activation is regulated via the interaction between the surface receptor complexes in BCRs and specific antigens[40]. The iProChip-DIA approach allowed the mapping of 83% of proteins within the BCR pathway and key BCR coreceptors, including CD19, CD21, CD22, and CD81 (Supplementary Fig. 22). Compared to a human immune cell proteomics study at a depth of >10,000 proteins using 28 primary hematopoietic cell populations by Rieckmann et al., 93% of the 4211 proteins identified from MEC-1 cells were in common, with an additional 309 proteins uniquely identified in this study, including a comparable number of key B cell surface receptors, such as CD19, CD21, CD22, CD81, FcgRIIB, and Igβ[41]. We further compared our data with B cell leukemia cells from Johnston et al.[42]. Even though the MEC-1 cell line also belongs to human B cell leukemia, only 51% of the protein groups commonly overlapped between the two datasets (Supplementary Table 4 and Supplementary Fig. 23). The common or unique proteins are likely due to cell type-dependent protein expression according to the Human Protein Atlas database (https://www.proteinatlas.org/). Nevertheless, B cell markers (e.g., CD19, CD20, and CD22) were commonly detected in both datasets and not detected in our PC-9 data. Similarly, lung cancer markers (e.g., EGFR and TP53) were not detected in MEC-1 cells (Supplementary Table 4). This supports the notion of cell-

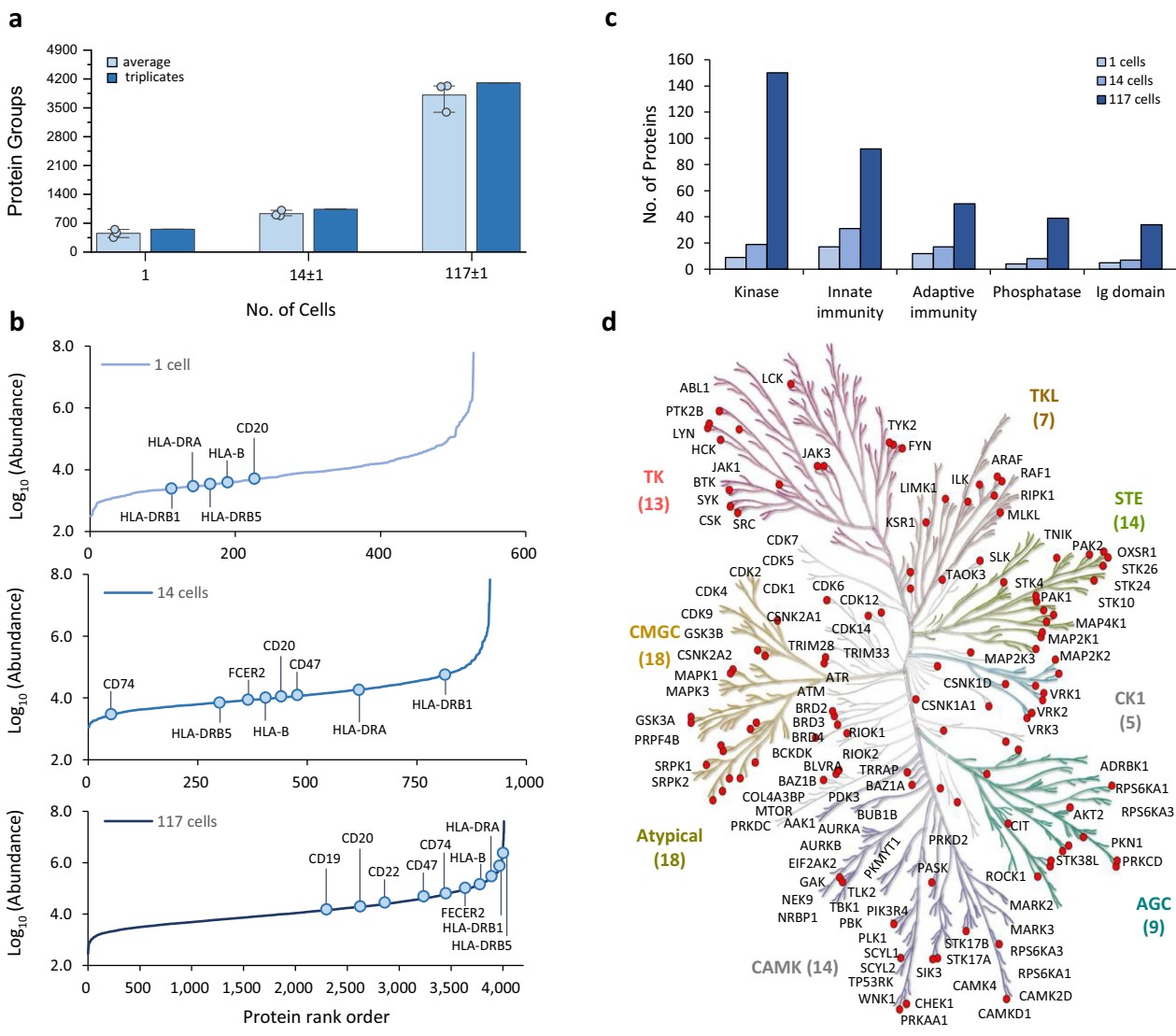

**Fig. 5 Application of iProChip-DIA for proteomic profiling of MEC-1 cells. a** Identification summary of protein groups across different MEC-1 cell numbers by iProChip coupled to DIA-MS. Data are presented as mean values ± SD ($n = 3$ independent cell samples for each condition). **b** Assessment of dynamic range based on protein abundance rank and annotation of selected proteins related to immune cancer markers. **c** Enrichment of immune-related and other functional classes against the UniProtKB database. **d** Kinase tree for mapping 114 kinases from the total cell numbers. The kinase families listed includes TK (tyrosine kinases), TKL (tyrosine kinase-like), CK1 (casein kinase 1), CAMK (calcium/calmodulin-dependent protein kinase), AGC (containing PKA, PKG, PKC families), CMGC (containing CDKs, MAPK, GSK, CLK families), and STE (serine/threonine kinases many involved in MAPK kinases cascade). Source data are provided as a Source data file.

dependent specificity of protein expression. Taken together, these results demonstrate the versatility of this approach for in-depth proteomic characterization of distinct cell types, thereby paving the way toward quantitative exploration of dissecting cellular heterogeneity in complex systems such as the tumor microenvironment.

**Enhanced sensitivity by single-cell integrated proteomics chip (SciProChip) and DIA MS.** Inspired by promising results in the proteome coverage and quantification of iProChip, we next sought to implement iProChip dedicated to 20-plex processing of single cells, which we termed single-cell iProChip (SciProChip) (Fig. 6a, b). This SciProChip was designed to include 20 chambers, each containing a single-cell trap, to facilitate precise and unattended capture of one cell for proteomic processing (Fig. 6c). The chip showed an improved cell usage efficiency of ~40% for

single-cell capture by optimal positioning of the cell traps and the narrower dimension of the chamber (Fig. 6d and "Methods"). Compared to iProChip, the total processing volume for Sci-ProChip was reduced from 312 to 78.5 nL, and the length of the C18-packed column was reduced from 2.5 to 1 cm, both of which helped to reduce sample loss. For compatibility of small cell input, the LC–MS/MS gradient time was reduced to 90 min.

Using SciProChip, we next performed proteome profiling of a series of single cells acquired from two batches of cell cultures using two chips (Supplementary Data 3). For the two batches, each containing 10 single-cell profiles, similar depths of an average of ~1500 ± 131 protein groups were reproducibly identified across 20 single cells. Compared to single-cell profiling by iProChip, significantly improved (1.53-fold) proteomic coverage was obtained by SciProChip (Fig. 6e). The accumulated number of all 10 single-cell datasets yielded a total identification of 1,792 and 1,995 protein groups from both batches of single

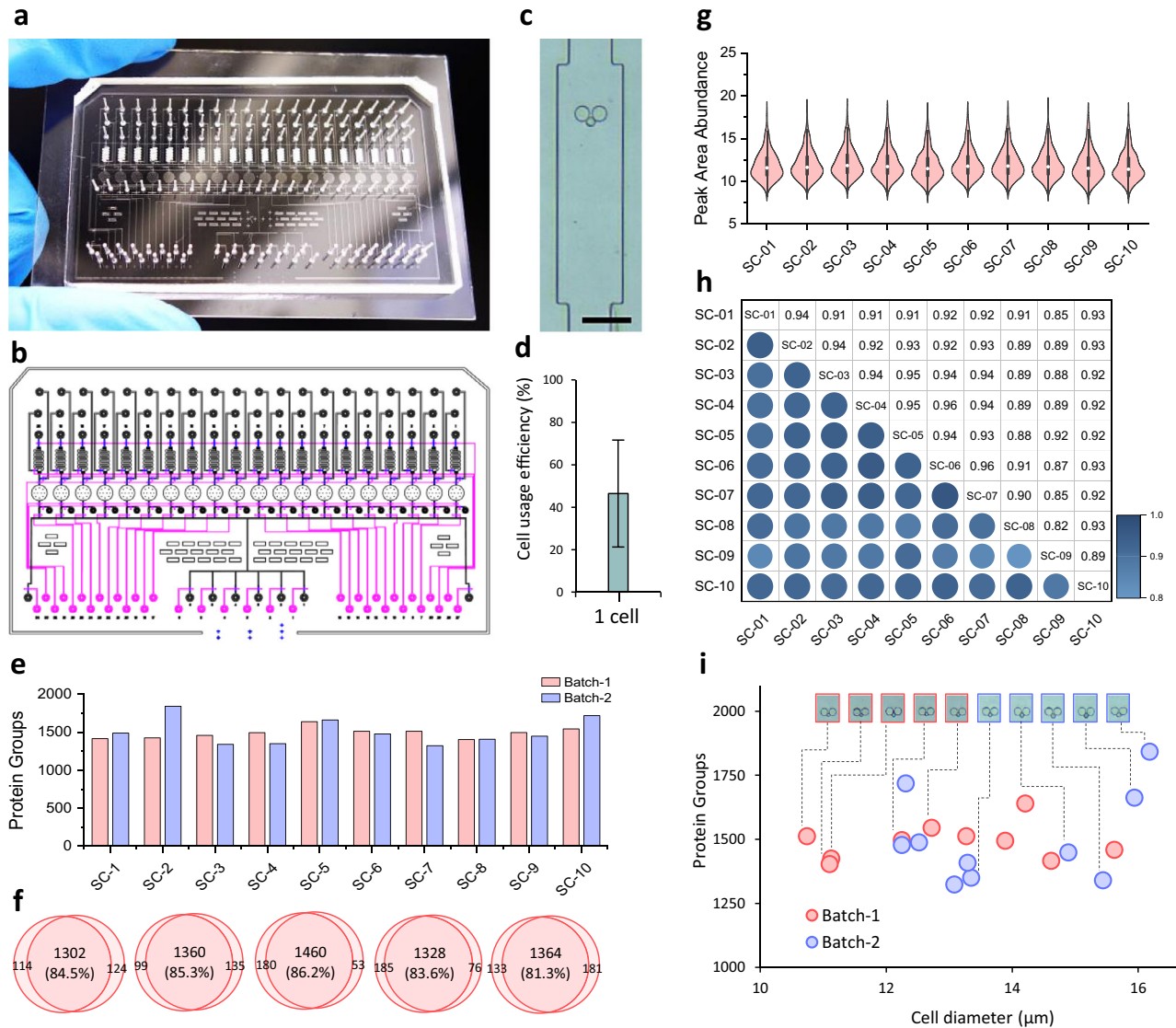

**Fig. 6 Single-cell integrated proteomics chip (SciProChip) for 20-plex single-cell proteomic analysis. a** A photograph of SciProChip for parallel processing of 20 single cells. **b** A schematic of the overall SciProChip layout. The control layer is shown in pink, while the flow layer is shown in black and blue. Note that SciProChip contains 20 operational units. **c** A bright-field image of a capturing chamber with a trapped cell. Scale bar: 120 μm. **d** The cell usage efficiency of SciProChip. Data are presented as mean values ± SD ($n = 28$ independent measurements). **e** Protein group identification of 10 single cells from 2 independent experimental batches. SC: single cell. **f** Overlaps of protein groups identified in replicate analyses and **g** distributions of total protein abundances in identified proteins of single cells shown in **e**. Each data was analyzed as an individual measurement. Centerlines show the mean; box limits indicate the 25th and 75th percentiles; whiskers, 1.5× interquartile range. **h** Pairwise Pearson's correlation showing the reproducibility of protein abundances obtained across different single cells. **i** Correlation between the estimated cell sizes and protein groups identified of 20 single cells. Source data are provided as a Source data file.

cells, respectively. Moreover, the identified protein groups achieved good overlap (81%-86%) between replicates of single-cell runs, indicating high reproducibility in the SciProChip-DIA workflow (Fig. 6f). Next, we evaluated the quantitation performance of single-cell replicates by calculating the overall protein abundances quantified in every single cell, which showed a consistent log abundance distribution across all datasets (Fig. 6g). The analytical reproducibility was assessed by computing the pairwise correlation of protein abundances between replicates, which showed a good Pearson correlation (Fig. 6h). We further explored the potential correlation between cell size and identified protein groups at the single-cell level. Intriguingly, the results showed that the number of identified proteins did not seem to correlate with the approximate size of individual captured cells (Fig. 6i). In summary, compared to iProChip, SciProChip

demonstrates higher cell usage efficiency and significantly improved proteomic coverage while maintaining good reproducibility for single-cell proteomic profiling.

## Discussion

We report herein all-in-one and fully automated devices capable of cell isolation, counting, imaging, and proteomic processing for in-depth microproteomics identification as well as quantification down to the single-cell level. Despite tremendous efforts, hundreds to approximately one-two thousand proteome coverage is limited to reach disease-specific proteins or pathways that are usually very low abundant. Thus, two chips were designed for different capacities from a single cell (SciProChip) to 10-100 cells (iProChip) to allow experiments that may require separate cell

inputs for sufficient proteomic depth to explore cell biology. Size-based sorting allows rapid and quantifiable cell isolation and imaging, obviating the necessity to use external cell sorters, such as fluorescence-activated cell sorting. Such a streamlined strategy circumvents limitations in sample loss during multistep sample transfers and minimizes surface absorption in conventional workflows, thus improving proteomic sensitivity with good reproducibility. The customized reagents for the presented workflow facilitate multiplexed on-chip cell lysis and protein digestion, followed by desalting to generate peptides ready for MS analysis. Importantly, the integrated SciProChip-DIA workflow demonstrated sensitive proteome coverage of 1500 ± 131 protein groups from a single PC-9 cell. Compared to recent studies for single-cell proteomic profiling, including the identification of 427 proteins using "DISCO"[35], 1475 protein groups using nanoPOTS with MBR[13], and ~1000 proteins using isobaric tandem-mass-tag (TMT)-based SCoPE2[12] and SCeptre[24], this study reported among one of the most sensitive proteome coverages for characterizing a single mammalian cell. Additionally, good analytical merits, including quantitation linearity, wide dynamic range in protein abundances, good between-replicate reproducibility, and low between-run missing values, were systematically benchmarked. Important druggable targets, biomarkers and signaling components related to either NSCLC or BCR signaling pathways were identified and quantified in low-scale samples, demonstrating the promising applicability of the chip-DIA workflow for both cell biology and clinical proteomics research.

Apart from the current study, there are several features that iProChip/SciProChip aims to further improve. For instance, size-based cell isolation of current chips could be combined with affinity-based cell sorting through surface marker functionalization to enable nanoproteomic analysis from a subpopulation of input cells. Meanwhile, the current workflow collected peptides from the chip and manually transferred them to a conventional autosampler to load ~90% of desalted peptides for LC–MS/MS analysis, which resulted in considerable sample loss. We believe such peptide transfer efficiency could be improved through the online interface of the chips directly with the autosampler on LC–MS/MS, which would reduce sample loss and increase overall sensitivity substantially[43,44]. High-throughput ability is important for single-cell characterization from a heterogeneous sample. The multiplexity of 20 single cells in the presented platform remains to be further increased. Such throughput may be further increased either by incorporating more single-cell processing units or by implementing protocols that allow multiplexed analysis such as TMT labeling. Other strategies to increase the proteomic coverage may benefit from using ultralow-flow nanoLC to enhance chromatographic resolution and FAIMS to effectively filter singly charged ions[13].

iProChip/SciProChip is designed as a versatile, scalable, and robust device in which existing proteomic methods can be readily integrated to fit different applications. Conceivably, in addition to static proteome analyses, our chips may serve as a platform for studying dynamic proteomic alterations upon cell stimulation, e.g., by ligand triggering, using cell imaging directly on-chip. Additional steps such as multiplex isobaric labeling and sub-proteome characterizations can be readily integrated to achieve sensitive and multiplexed proteomic profiling. Further incorporation of peptide enrichment components in the device can be designed to extend single-cell proteomics beyond the post-translational modification of the proteome. The chip-DIA approach is anticipated to find a variety of applications where only limited input samples are available, e.g., rare cell populations from clinical specimens. We envision that our approach can open a new avenue in bringing distinct functionalities into a single miniaturized platform to enable new ways of proteomic research at the single-cell level.

## Methods

**Materials and reagents**. Triethylammonium bicarbonate (TEABC), TCEP, CAA, chlorotrimethylsilane, hexamethyldisilazane (HMDS) and trifluoroacetic acid (TFA) were purchased from Sigma-Aldrich (St. Louis, MO, USA) and formic acid (FA) was bought from Honeywell Fluka (NC, USA) and were freshly prepared in ddH$_2$O each day before use. LC-MS grade acetonitrile (ACN) and Pierce™ BCA Protein Assay Kit was purchased from Thermo Fisher Scientific. RapiGest SF surfactant (Waters, MA, USA) was dissolved in a fresh 50 mM triethylammonium bicarbonate buffer with a concentration of 0.3% (w/v), aliquoted and stored at −30 °C until further use. Lys-C (MS grade) and Trypsin (MS grade) were bought from Promega (Madison, WI, USA). 5 μm C18 beads (300-Å pore size) were purchased from VDSpher® (VDS optilab, Chromatographie Technik GmbH, Berlin, Germany). AZ-40XT photoresist, SU-8 3025 photoresist and their developers were purchased from MicroChem (MA, USA). RTV 615 poly-dimethylsiloxane (PDMS) pre-polymer and curing agent were purchased from Momentive Performance Materials (NY, USA). Deionized (DI) water was purified using a Milli-Q Ultrapure Water Purification System (Millipore, Billerica, MA, USA).

**Cell culture**. The human lung adenocarcinoma cell line PC-9 was obtained from RIKEN BioResource Research Center (Japan, catalog number: RCB4455). PC-9 was cultured in RPMI-1640 medium supplemented with fetal bovine serum (FBS) (10% v/v), sodium bicarbonate (2% w/v), 1 mM sodium pyruvate, 100 units/mL penicillin, and 100 μg/mL streptomycin at 37 °C in a humidified 5% CO$_2$ incubator. The human B-CLL cell line MEC-1 was originally obtained from DSMZ GmbH (Germany, catalog number: ACC − 497) and cultured in Iscove's Modified Dulbecco's Medium supplemented with 10% heat-inactivated FBS, 100 units/mL penicillin, and 100 μg/mL streptomycin in a humidified incubator at 37 °C and 5% CO$_2$.

**Design and fabrication of the microfluidic chip**. Integrated proteomics chips, including iProChip and SciProChip, are two-layer PDMS devices with a top flow layer and a bottom control layer. The device layouts were designed using the AutoCAD software (Autodesk, USA). The flow layer contains a channel network that includes cell and buffer inlets, cell capture chambers, reaction vessels, desalting columns, and sample collection outlets (see chip schematics in Supplementary Fig. 1). Triplicate operational units are designated for 10, 50, and 100 cells in the iProChip while 20 single-cell units are designated in the SciProChip. The control layer contains 34 hydraulic microvalves to control the flow layer. To account for PDMS shrinkage, the flow layer layout was expanded by 1.5% relative to the control layer. The photo-mask for the flow layer and control layer was fabricated by the Advanced Nano/Micro Fabrication and Characterization Laboratory in Academia Sinica (Taiwan) and Taiwan Kong King Co., Ltd. respectively.

To fabricate master molds for the proteomics chips, regular photolithographic protocols were followed and performed on silicon wafers using an EVG-620 mask aligner[26]. Briefly, a 4-inch silicon wafer was cleaned thoroughly using acetone, isopropanol, and DI water, followed by dehydration (105 °C; 5 min) and HMDS coating to promote photoresist adhesion. The mold for the control layer was generated by spinning the negative photoresist SU 8-3025 (MicroChem, USA) at 4200 r.p.m. to obtain the height of 25 μm, followed by standard photo-patterning, developing and baking protocols. Meanwhile, for the flow layer mold, valve structures were firstly fabricated using the positive photoresist AZ 40XT (MicroChem, USA) spinning at 3500 r.p.m. to achieve the height of 25 μm, followed by standard protocols in photo-patterning, developing, baking and reflow to generate rounded features for effective valve closures[26]. Then, onto the same wafer, the rest of the flow layer features were generated using SU 8-3025 spinning at 4200 r.p.m. to obtain the height of 25 μm. To obtain a 100 μm height reaction chamber in the case of iProChip, the aforementioned SU 8-3025 protocol was repeated with a spin speed of 1100 r.p.m. to cast an additional 75 μm photoresist at the reaction chamber region using a dedicated photo-mask. Mercury match light at an exposure dose of 250 mJ/cm$^2$ was used for patterning all features.

The proteomics chips were prepared by casting an optically transparent soft elastomer PDMS onto patterned master molds. The wafer with either flow or control layer layout was pre-treated with trimethylchlorosilane in a fume hood for 15 min to ensure chip features were non-sticky to PDMS, and thus allowing multiple rounds of usage. The device has a push-up design, with a thick flow layer binding over a thin control layer. To make the flow layer, 60 g PDMS base and 6 g curing agent were thoroughly mixed by a mixer (Thinky ARE-310 Planetary Centrifugal Mixer) for 3 min at 2000 r.p.m. and 1.5 min at 2200 r.p.m., followed by degassing for 1 h using a desiccator before pouring onto the mold. To prepare the thin control layer, 10 g PDMS base and 1 g curing agent were thoroughly mixed, followed by spinning on the control layer mold in three steps: 300 r.p.m. for 20 s, 1800 r.p.m. for 50 s, and 0 r.p.m. for 10 s (Laurell WS-650HZ-23NPP/UD2 Spin coater). The control layer was then allowed to level off for 15 min on a horizontal surface, and both layers were baked in an 80 °C oven for at least 45 min. The thick flow layer was then peeled off from the wafer, followed by cutting and hole

punching (710 μm inner diameter biopsy puncher; Syneoco, USA). It was then activated using the oxygen plasma at the highest RF power for 1 min (Harrick plasma cleaner PDC-001-HP), before being aligned and bound to the thin control layer using a custom stereo-microscope with independent x-, y-, and z-alignment controller (Nikon-SMZ18). After baking in the 80 °C oven overnight, the bounded chip was trimmed, peeled off, and holes punched, before binding to a freshly plasma-treated 75 × 50 × 1 mm glass slide. The bounded chip was then placed in the 80 °C oven for at least 48 h before following experimental use.

**Preparation and characterization of the SPE column using BCA assay.** The desalting columns were prepared by slurry packing with 5 μm C18 beads in acetone (5 g/mL) with an input pressure of 13 psi, which typically takes 10–12 min (Supplementary Movie 1), followed by washing with methanol for 4–6 min and activated by buffer A (100% ACN + 0.1% TFA), buffer B (50% ACN + 0.1% TFA) and buffer C (0.1% TFA, ddH$_2$O) for 15 min. To examine the sample recovery efficiency of the on-chip desalting column, protein quantification was performed by BCA protein assay (Thermo Scientific, USA). Briefly, two batches of digested BSA peptides were prepared through serial dilutions to make final concentrations of 1.0, 0.5, 0.25, and 0.125 μg in 10 μL Buffer C. One batch was then subjected into the pre-activated SPE columns, and resulting samples were then collected, speedvac dried, and resuspended in 10 μL buffer C. Both samples were individually pipetted out (10 μL each) into a 96 well plate, and 190 μL of the BCA working reagent was added to each well, followed by thorough mixing on a plate shaker for 30 s and further incubation at 37 °C for 30 min. Finally, the plate was cooled to room temperature and the absorbance for each sample was measured at 562 nm by a plate reader (EnSpire™ Plate Reader). The sample recovery was determined by plotting the recovered peptide concentration versus input peptide concentration (Fig. 2f).

**Examination of mixing efficiency and preferential flow in the digestion chamber.** The mixing efficiency of the chip was apprehended through introducing blue and yellow food dyes sequentially into the digestion chambers, followed by applying either active mixing techniques including vortex-mixing (Unico-L-VM2000) and shaker-mixing (Eppendorf-Thermomixer F), or passive mixing through diffusion processes alone. To visualize and quantify the mixing performance, time-lapsed images of the digestion chamber filled with blue and yellow dyes were recorded every 30 s for all conditions (Supplementary Fig. 5). Using imaging analysis (Image J), the standard deviation of the pixel intensity was determined for images of the initial unmixed state, $\sigma_o$, and for individual time-lapsed images taken throughout the process, $\sigma$. The mixing efficiencies were then quantified as a non-dimensional parameter, RMI, according to the following Eq. (1)[31].

$$\text{RMI} = 1 - \frac{\sqrt{\frac{1}{N}\sum_{i=1}^{N}(I_i - <I>)^2}}{\sqrt{\frac{1}{N}\sum_{i=1}^{N}(I_{oi} - <I>)^2}} = 1 - \frac{\sigma}{\sigma_o} \qquad (1)$$

where $N$ is the total number of pixels, $I_{oi}$ is the local pixel intensity in the unmixed state, $I_i$ is the local pixel intensity in mixed state, and $<I>$ is the average intensity.

The fluid dynamics, especially the possibility of preferential flow as a result of (1) the circular shape of the digestion chamber, (2) dominated laminar flow inside the microfluidic channels, and (3) non-negligible flow resistance from the downstream integrated desalting column, were carefully examined to determine the optimal condition for subsequent sample retrieval. The time and corresponding pressures needed to push the colored fluid through C18 bead-packed desalting columns were found to be 30, 20, 13, 9, and 8 min for 9, 10, 11, 12, and 13 psi, respectively (Supplementary Fig. 7a). On the other hand, for an empty desalting column it took less than 35 s for dye molecules to be transferred for pressure >4 psi, confirming there was no preferential flow as a result of the circular geometry of the digestion chamber (Supplementary Fig. 7b). Based on such characterization, input pressure of 11 psi was used in the proteomics workflow for transferring digested peptides to the packed SPE column during the final sample clean-up.

**Examination of on-chip cell capture efficiency and validation of loading low concentration samples.** In order to characterize minimal numbers of cells needed to operate in this chip, we prepared vials containing PC-9 cells with a cell density of 25 cells/μL, and used a total volume of either 5 or 10 μL (which equals to 125 or 250 cells in each vial). The cell vial was then connected to the chip and directly flowed into the cell chamber at 3 psi to determine how much sample volume and cells are needed for capturing desired cell numbers (including 1, 5, 10, 50, and 100 cells) in individual chambers. The cell solution was injected into each chamber so that a fraction of cells were captured by wedge-shaped traps while the remaining cells simply passed by to the waste-outlets. A real-time chip monitoring by a bright-field microscope was used to inspect the cell flow and capture; the valves were closed when desired numbers of cells were captured in the chambers. Note that the imaging-ready feature of iProChip enables easy quantification of trapped cells. At this cell density, it took 3–40 s to capture 1–100 cells (Supplementary Table 2 and Supplementary Fig. 4). Our result showed 4–8% cell usage efficiency for capturing 1, 5, and 10 cells and 33–44% for trapping 50 and 100 cells. For SciProChip, the cell usage efficiency was characterized by examining the real-time video recorded

during single-cell trapping. For instance, if the first cell entering the chamber was immediately captured, then cell usage efficiency would be counted as 100%, whereas 50% means the second cell (out of total 2 cells) was captured and 33% means the third one (out of total 3 cells) was captured, and so on. A total of 28 single-cell capturing videos were analyzed.

**Proteomics workflow using the iProChip and SciProChip.** The PMDS chip was hooked up to the control system via 34 stainless steel connectors attached to Tygon tubings and then mounted onto an inverted microscope (Nikon-ECLIPSE-T s2). Working pressure of 28 to 30 psi was used to operate the control layer. To minimize non-specific absorption during proteomic processing, all modules of the iProChip except the desalting column were freshly coated with 0.1% BSA for 1 h, followed by phosphate-buffered saline (PBS) rinsing for 10 min and dried under the nitrogen stream. Afterward, desalting columns were packed with C18 beads (0.01 mg/mL in acetone), and cell capture chambers were degassed using PBS solution in order to achieve well dispersed cell flow into the chambers.

The proteomic processing steps start by introducing a cell suspension (500 or 25 cells/μL) into cell capture chambers under flow pressure of 3 psi, with which a quantifiable number of cells in the range of 1, 5, 10, 50, and 100 were trapped in a cell trapping chamber by controlling the injection time through real-time imaging. In the second step, cocktail lysing buffer (42 nL in the case of iProChip, or 20 nL in SciProChip) consisting of 0.3% RapiGest, 10 mM TCEP, 40 mM CAA was infused to individual cell capture chamber, followed by incubation for 30 min at 75 °C on a plate shaker (Eppendorf-Thermomixer F; 400 r.p.m.) (Supplementary Fig. 5 and Supplementary Movie 3). In the third step, Lys C (42 nL/20 nL, protein/Lysine-C 20:1 w/w) and trypsin (42 nL/20 nL, protein/trypsin 10:1 w/w) were sequentially infused into individual digestion vessels, followed by further incubation at 40 °C for 16 h on a plate shaker (400 r.p.m.). Finally, 25% FA (42 nL/20 nL, final 5% v/v) was infused to the individual chamber and incubated for 55 min at 40 °C to quench the enzymatic digestion. The peptide clean-up was carried out by the SPE columns preconditioned and equilibrated with buffer A, buffer B, and buffer C running for 15 min each. Then, the processed peptides were pushed from the digestion vessels to the activated SPE columns by buffer C at 11 psi for 15 min of desalting. Finally, buffer B was passed through SPE columns to elute the peptides into lo-binding vials, speedvac dried prior to subsequent LC-MS/MS analysis. The overall time for parallel processing of 9 samples (iProChip) and 20 samples (SciProChip) was ~20 h.

**Bulk proteomics workflow.** For comparison with chip-based proteomics, vial-based processing was performed for PC-9 cells collected with a concentration of $5 \times 10^5$/mL in a 10 mL tube and centrifuged at 194 × g for 3 min to remove the cell culture media. The cell pellet was further washed with 5 mL of PBS buffer three times. Next, the cell density was further diluted to a working concentration of 20,000/mL (i.e. 20 cells/μL). Briefly, a volume of 5 μL of cell suspension was dispensed into a low-binding vial (0.5 mL) and the final volume was adjusted to 20 μL with PBS diluent and centrifuged at 96 × g for 3 min. After centrifugation, the PBS buffer was carefully discarded. Cells were then lysed, reduced and alkylated by dispensing 20 μL of RapiGest cocktail lysis buffer (0.3% RapiGest, 10 mM TCEP, 40 mM CAA) and heating at 70 °C for 30 min at 750 r.p.m. Protein digestion was performed using Lysine-C (protein/Lysine-C 20:1 w/w) and trypsin (protein/trypsin 10:1 w/w) for 16 h at 37 °C. Afterward, the samples were acidified with FA to a final concentration of (5% v/v) and incubated at 37 °C (for 1 h) to cleave RapiGest surfactant for downstream analysis. Finally, the peptides were dried and desalted through the C18 Ziptip. Finally, the clean peptide samples were analyzed by LC–MS/MS using the Orbitrap Eclipse mass spectrometer.

*Dilution-based processing.* PC-9 cells were collected with a concentration of $5 \times 10^5$/ mL in a 10 mL tube and centrifuged at 194 × g for 3 min to remove the cell culture media. The cell pellet was further washed with 5 mL of PBS buffer three times. Briefly, $10^6$ cells in PBS were dispensed into vials and centrifuged at 96 × g for 3 min. After centrifugation, the PBS buffer was carefully discarded. Cells were then lysed, reduced, and alkylated by dispensing RapiGest cocktail lysis buffer (0.3% RapiGest, 10 mM TCEP, 40 mM CAA) and heating at 70 °C for 30 min and sonicated with 5-cycles for 5 min using a tip sonicator (Bioruptor Plus, Diagenode). Cells were then centrifuged at 16,000 g for 20 min at 4 °C and the supernatant lysate was collected. BCA assay was performed using a BCA protein assay kit to measure the concentration of protein lysate, and a ~50 μg lysate was digested using Lysine-C (protein/Lysine-C 20:1 w/w) and Trypsin (protein/trypsin 10:1 w/w) and incubated for 16 h at 37 °C. Afterward, the samples were acidified with FA to a final concentration of (5% v/v) and allowed to incubate at 37 °C (for 1 h) to cleave RapiGest surfactant for downstream analysis. Finally, the peptides were dried and desalted through the C18 Ziptip. Finally, the sample was diluted to the desired concentration and small aliquots corresponding to 1.5 ng (~10 cells) and 15 ng (~100 cells) were injected and subsequently analyzed in triplicate by LC−MS/MS using the Orbitrap Eclipse mass spectrometer, respectively.

*Bulk conventional processing.* PC-9 cells were washed by PBS at least three times and lysed by phase transfer surfactants (PTS) buffer containing 12 mM SDC, 12 mM SLS, 100 mM Tris-HCl (pH 9.0), phosphatase inhibitor cocktail, and

protease inhibitor. The cell lysate was collected in a 1.5 mL tube. The lysates were heated at 95 °C for 5 min and sonicated for 10 min (30 s on, 30 s off) using a tip sonicator (Bioruptor Plus, Diagenodes). The cell pellet was removed by centrifugation (16,200 × g for 30 min at 4 °C). Then, methanol/chloroform protein precipitation was performed on the supernatant as follows: 1× sample, 4× methanol, 1× chloroform and 3× ddH$_2$O were sequentially added and vortexed. After centrifugation (10 min at room temperature, at 16,200 × g), the upper aqueous phase was removed. Then, 3× methanol was added to wash the protein pellet and centrifuged again. The supernatant was removed and the pellet was resolved in 8 M urea in 50 mM TEABC. The protein concentration was measured by the BCA protein assay kit. Next, final concentration of 10 mM DTT was added to the protein lysate and incubated at 29 °C for 30 min. Then, a final concentration of 50 mM IAM was added and incubated at 29 °C for 45 min. For protein digestion, 8 M urea was diluted to 4 M with 50 mM TEABC, and then Lys-C (100:1) was added and digested for 3 h at 29 °C. Finally, the urea was further diluted, followed by trypsin (50:1) digestion at 29 °C overnight. After protein digestion, a final concentration of 0.5% TFA was added to quench the reaction. Finally, the peptide desalting was performed by Stage-tip packed by the SDB-XC membrane, which was pre-activated with 80% ACN, 0.1% TFA, and then conditioned with 5% ACN, 0.1% TFA. After desalting, the sample was eluted with 80% ACN, 0.1% TFA and collected in a lo-binding tube.

**LC-MS/MS analysis**. The Orbitrap Eclipse mass spectrometer (ThermoFisher Scientific, Xcalibur Ver. 4.3.73.11) coupled with an Ultimate 3000 RSLCnano system (Thermo Fisher Scientific) was used for LC-MS/MS analysis in this study. The desalted peptides were resuspended to 5 μL in the loading buffer (0.1% FA) spiked with iRT peptides (Biognosys, Schlieren, Switzerland) and 4 μL was loaded to autosampler for LC-MS/MS analysis. The nanoflow Ultimate 3000 UHPLC (Thermo Fisher Scientific) with a capillary C18 column (Waters, nanoEase, 130 Å, 1.7 μm, 75 μm × 250 mm) was employed for peptide separation at 300 nL/min using buffer A (0.1% FA in water) and buffer B (0.1% FA in ACN). The peptides were separated through gradients from 3% to 25% ACN in 137.5 min, followed by a 4 min increase to 40% and 2 min increase to 95% ACN. After washout for 5 min at 95% ACN, the C18 column was re-equilibrated at 1% ACN for 10 min. The MS instrument was operated in the positive ion mode with spray voltage set to 1.75 kV, RF lens level set at 30%, and ion transfer tube heated at 305 °C. For DDA mode, top N multiply charged precursors were automatically isolated and fragmented according to their intensities within the cycle time of 3 s. The intensity threshold was set to 8E3. Full MS was scanned at a resolution of 120,000 with automatic gain control (AGC) target of 1E6 and a max injection time of 50 ms. Mass range was set to 375–1500 m/z and isolation width for MS/MS analysis was set to 1.4 m/z with advanced peak determination. Normalized collision energy (CE) of high-energy collision dissociation (HCD) was set to 30%. MS/MS was scanned in orbitrap at a resolution of 120,000 with an AGC target of 1.25E5 and a max injection time of 254 ms. For DIA mode, the full MS resolution, AGC target, and max injection time are the same as the DDA mode. The mass range for DIA MS/MS analysis was set to 400–800 m/z and overall 40 scan events of 10 m/z isolation window were employed with an overlap of 1 m/z. The MS/MS scan was performed in HCD mode using the following parameters: normalized CE = 30%; resolution = 30,000; AGC target = 4E5; max injection time = 54 ms. For single-cell samples (processed through SciProChip), the LC-MS/MS analysis time length is shortened to a 90 min gradient using the DIA mode, with higher MS/MS resolution at 60,000 and max ion injection times of 118 ms. All data were acquired in profile mode using positive polarity.

**Spectral library construction**. A set of project-specific spectra libraries (both at large-scale and small-scale) were constructed for lung cancer cell line (PC-9), and human CLL cell line (MEC-1) processed in iProChip or in vial-based processing. The raw files for library generation were acquired in both data-dependent as well as DIA mode to obtain project-specific hybrid libraries. Large-scale libraries were constructed by using bulk samples (~1 μg peptide, n = 10 raw files for PC-9) while small-scale libraries were generated using ~10 cells processed through iProChip (n = 9 raw files for PC-9) and aliquots of 1.5 ng (~10 cells, n = 13 raw files for PC-9) obtained from dilution. To further maximize the identification results, another small-scale library using 5 and 50 cells DIA runs (n = 6 raw files) processed through iProChip was also constructed. Similarly, for MEC-1 cells, large-scale libraries from bulk (n = 8 raw files) and small-scale from ~10 cells (n = 12 raw files) were constructed. Protein identification was performed using Spectronaut Pulsar software (v13; Biognosys, Switzerland)[36] filtering at 1% FDR PSM, peptide, and protein level. Trypsin was selected as a digestion enzyme with maximum two miscleavage sites. The minimum and maximum allowed peptide lengths in search space were 7 and 52, respectively. Variable modifications of acetylation on protein N-terminus and oxidation on methionine were included, while carbamidomethylation of cysteine was set as a fixed modification. For database search, the SwissProt/UniProt human proteome database (2015_12 release, Homo Sapiens = 20,193 entries) with the inclusion of 11 iRT peptide sequences was used. Project-specific spectral libraries were generated using standard parameters in Spectronaut. Briefly, Normalized retention time was obtained using segmented regression to determine iRT in each run by the precision iRT function. The six most intense fragment ions were included with iRT retention time normalization.

Fragment ions of minimum m/z 300, maximum m/z 1800, and minimal relative intensity of 5% were included. Fragment ions with less than three amino acid residues were not considered.

**Data analysis**. The DIA raw files were analyzed using Spectronaut software (version 13.11200127.43655) against home-built project spectra libraries as well as in library-free mode (dirDIA) using standard settings. For library-free strategy (direct DIA), protein identification from the DIA dataset was performed by database search against the SwissProt/UniProt human proteome database (Homo Sapiens: 20,193 entries) using Pulsar search engine in Spectronaut. For library-based strategy using our project spectra libraries, protein identification was performed using the Spectronaut Pulsar software[36] filtering at 1% FDR PSM and protein level described in the previous session (See Spectral library construction). For protein quantitation, peak area at MS2 level was calculated using top 3 peptides per protein and minor peptide grouping was adapted based on the striped sequence. For comparison, the DDA raw files were analyzed with Maxquant software (version 1.5.6.5)[45] using standard settings for Orbitrap MS and LFQ protein quantification. The first search and main search peptide tolerance were set as 20 and 4.5 p.p.m., respectively. The protein and peptide were both filtered at 1% FDR. Variable modifications of acetylation on protein N-terminus and oxidation on methionine were included, while carbamidomethylation of cysteine was set as a static modification. The SwissProt/UniProt human proteome database (downloaded on 2015/12/15, Homo Sapiens: 20,193 entries) with iRT peptides sequence was used. The pathway analyses were performed using the KEGG Mapper from the KEGG online database (https://www.kegg.jp/kegg/mapper.html). The kinome tree was drawn using the KinMap online tool from the Kinhub database platform (http://www.kinhub.org/kinmap)[38].

**Reporting summary**. Further information on research design is available in the Nature Research Reporting Summary linked to this article.

## Data availability
The mass spectrometry raw datasets, reference spectral libraries, and Spectronanut quantification outputs have been deposited in the Japan ProteOme Standard Repository (jPOST) and can be accessed through ProteomeXchange consortium (http://www.proteomexchange.org/). The dataset identifier is JPST000971 for JPOST and PXD023325 for ProteomeXchange. The protein sequence fasta file was obtained from the UniProtKB (https://www.uniprot.org/) human proteome database. The iRT peptides fasta file was downloaded from Biognosys website (https://biognosys.com/product/irt-kit/#SupportMaterials). For pathways analysis and functional annotation, the following databases were used: KEGG database (https://www.genome.jp/kegg/) and kinase families of KinMap database (http://www.kinhub.org/kinmap/). All data for iProChip/PC-9, iProChip/MEC-1, and SciProChip/PC-9 are available from the corresponding authors and are provided herein as Supplementary Data 1, 2, and 3, respectively. Source data are provided with this paper.

## Code availability
The code for chip control is available at https://doi.org/10.5281/zenodo.5656445.

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

## Acknowledgements

We would like to thank Jing Lin for developing an initial chip control program; Yi-Wen Fang, Pin-Lian Jiang, and Hsin-Ju Chan for assisting the PC-9 cell culture; Dr. Ying Li for SEM measurements; and Pei-Yi Lin and Dr. Hsiang-En Hsu for assisting LC-MS/MS measurement. This work was funded by the Ministry of Science and Technology (Grant MOST 107-2113-M-001-032-MY3; MOST 107-2113-M-001-023-MY3; MOST 110-2113-M-001-019-MY2; MOST 110-2113-M-001-020-MY3) and Academia Sinica (AS-TP-108-ML06; AS-iMATE-108-21). The master molds for the iProChip were fabricated at the Advanced Nano/Micro Fabrication and Characterization Laboratory at the Institute of Physics, Academia Sinica, Taiwan. The LC-MS/MS data acquisition and analysis by Orbitrap ECLIPSE mass spectrometer was performed in the Mass Spectrometry Facility located at the National Biotechnology Research Park (NBRP), Academia Sinica, Taiwan.

## Author contributions

S.T.G. and A.A.S. performed experiments and acquired and analyzed the data; A.A.S., R.B.K., and E.S.-W.C. generated spectral libraries, performed LC-MS/MS measurement, and analyzed the data; B.E., T.A., and K.-I.L. provided reagents; H.-L.T. and Y.-J.C. conceived and supervised the work. S.T.G., A.A.S., Y.-J.C., and H.-L.T. wrote the initial draft. All authors commented and contributed to the final editing of the manuscript.

## Competing interests

The authors declare no competing interests.
