## [Peer Review File · Nature Communications]

Reviewers' comments:

Reviewer #1 (Remarks to the Author):

The authors report a novel microfluidic platform for isolation of intact cells, followed by cell lysis and peptide digestion/desalting. The sample of peptides can subsequently be retrieved from the device and subject to DIA mass spectrometry. In the manuscript, authors demonstrate that their device can capture different numbers of cells in each sample chamber by tuning different experimental parameters and device design parameters, and can achieve capture of single cells if desired. They also show the device performance metrics in terms of mixing efficiency in the peptide digestion chamber, and show desalting performance. The performance of the coupled DIA m/s is also demonstrated with triplicate experiments using 13 or 14 cells, and triplicate experiments of single cells.

The device itself quite novel - it quite nicely accomplishes cell capture, cell lysis, peptide digestion and desalting. But it is unclear from the manuscript whether this device actually can practically address the issue of limited sample quantity for protein mass spec. Most of the time, the initial sample quantity (i.e. the number of cells available for processing) poses a problem because the sample processing steps introduce too much loss, and the experiment cannot be carried out if there are too few cells to begin with as there will not be enough in the final ready-for-ms library after all the processing steps. Although the authors show that in the device, they can capture as little as single cells, there are usually losses associated with loading of the microfluidic device. To show that this device is really able to address the sample input quantity issue, it would be better to discuss how much total input is loaded into the device for capture. I.e. what is the minimum cell number that is needed, and how that compares to standard practice or current frontier technology in mass spec.

Secondly, there was no raw data showing the effectiveness of the peptide desalting. Although the quantity of peptide recovered is an important metric, some validation of the salt or pH before and after the column is needed to confirm that the desalting has worked. Furthermore, the experiments showing the efficiency of peptide recovery only goes down to 0.2ug of loaded peptide, which is equivalent of peptides from on the order of ~1000 cells. But the experiments conducted throughout the paper are using from 1 to 100 cells, which would generate orders of magnitude less peptides. It has been well-demonstrated that at microfluidic volumes and with ultra-low input quantities of sample, adsorption is often a major contributing factor to sample loss when using microbead packed columns. The recovery efficiency of 89% may no longer hold at single cell level (200pg), but given the purpose of the device, it seems important to test the recover efficiency at the levels of peptide loading that the device will be used for.

Third, the comparison to bulk experiments is lacking, which is particularly problematic when the number of replicates of low-cell number experiments is also low ($n=3$ for single cells or $n=6$ for 13/14 cells, if we combine the DIA and DDA experiments). The reason why comparisons to conventional bulk (or even mini bulk such as 1,000 or 10,000 cells) is important is to be able to determine the sensitivity and robustness of the low cell number assay. Single cell assays are usually much lower in sensitivity than bulk assays; but as the user, it is important to know the degree of loss of sensitivity. A comparison to the bulk can tell us what fraction of the bulk proteins/peptides can be captured with the low-cell-number assay, and whether the peptides that are lost are due to low abundance, or subject to other technical bias. Without the bulk comparison, it is impossible to judge. In addition, usually bioinformatically pooled low-cell experiments are able to replicate bulk experiments due to the nature of such sampling. Therefore, increasing the number of replicates to a level that can be used to bioinformatically re-construct the bulk is a useful way to demonstrate the technical sensitivity of the assay. Currently there is no comparison to any external reference dataset generated using conventional gold standards. Given that the cells used were all cell lines, even a comparison to existing bulk datasets in the literature would be acceptable (in fact the authors mention such a study by Rieckmann et al, but do not perform any kind of comprehensive analysis to show how their data compares).

There is a similar problem with the number of replicates being too low, and trying to estimate the FDR or accuracy of the method (e.g. comparison of overlapping peptides between different experiments). For the number of proteins being sampled, a triplicate experiment is actually likely to under-estimate the accuracy of the assay; a larger n that when summed can generate bulk-like data is a better demonstration of the accuracy. With larger n , further statistical methods can be used to gauge the accuracy of the assay, such as looking at the CV of peptides detected across many cells, etc. While I appreciate the challenges of performing many single cell or 10 cell mass spec experiments to establish such statistical power, I strongly feel it is necessary to establish the reliability and accuracy of the method.

Analysis methods for establishing sensitivity and accuracy of the single cell -omics compared to bulk has been demonstrated extensively in the single-cell RNA-sequencing literature. Papers published from 2009-2015 on single-cell RNA-seq feature many ways to compare single-cell RNA measurements to bulk RNA measurements. I feel that the same analyses need to be applied to single-cell proteomics to establish feasibility. Authors can reference those methods as a roadmap.

Overall, I feel that the novelty of this manuscript is high, but much more rigour in terms of experimental design and data analysis are needed to establish this method as viable, feasible, and accurate for single-cell proteomics.

Minor comments:

Line 26 - “enables profiling of 1160 protein groups from triplicate analysis of a single mammalian cell” - this is written in a confusing way to suggest that the same single cell was analyzed 3 times. suggest re-writing to remove this confusion, e.g. “analysis of three individual/single mammalian cells”. It is impossible to have triplicate of the same single cell...

Line 153 - suggest changing or removing the word “prudently”; contextually this is not the right word use here.

Figure 2f - consider adding the unity line to the graph and making the graph square, with the axes using the same marks to enhance readability

Figure 3a - are the columns in the top and bottom panels matched in terms of the experiment? Suggest changing labels to experiment number followed by a parenthesis with notation of number of cells.

Figure 4b - the pink dot is quite confusing. Suggest changing the color of the right y-axis (cell area) to pink or some other color that matches the dots; also suggest using cell diameter instead of area, or an estimated cell volume based on the area measured by the image. Diameter is more commonly used in biology to convey cell size.

Reviewer #2 (Remarks to the Author):

In this manuscript, Gebreyesus and colleagues present a microfluidics approach for proteomic sample preparation of small cell populations – down to single cells. Specifically, they have constructed a PDMS chip (iProChip) that captures a pre-defined number of cells (5-10-100), and executes subsequent handling steps including cell lysis, protein reduction/alkylation, digestion and clean-up, which can be performed for 8 samples in parallel. Resulting peptides were analyzed by LC-MS using a data-independent approach (DIA), demonstrating the ability to identify ~4000 proteins from 100 cells, down to ~1000 proteins from single cells. They show that this includes various biologically and

clinically relevant proteins, including cell surface proteins and proteins involved in cell signaling. They conclude that the integrated sample preparation by the iProChip provides a powerful workflow for reproducible proteome analysis of low-input samples down to single cells.

This is an interesting manuscript, providing an alternative workflow compared to previous work using micro-pipetting in open array-based systems (e.g. nanoPOTS, OAD). This is a welcome development to show that alternative approaches are conceivable that may have advantages depending on the use case. The authors show the construction of the chip with an assessment of performance (sensitivity, consistency) when used in conjunction with DIA-MS. A real biological application is lacking – which is fair at this stage, instead showing proof of principle. Yet there are a few main concerns with regard to fitting the chip in a genuine single-cell workflow, and with regard to the identification of low-abundance (signaling) proteins presented in the manuscript – along with various smaller remarks:

Main concerns:

1. Although the authors present their system as a way to study single cells, strictly speaking the chip is not designed to do so. In fact, it is quite unclear how single cells were obtained: Line 230 states that this is done using a trap for 10 cells, which probably means that trapping needs to be monitored visually, and stopped as soon as exactly 1 cell has been trapped? This would denote a cumbersome procedure of which it is hard to imagine how one would achieve this for hundreds of cells?
2. Related to the previous point, it is unclear what application the authors have in mind by providing the ability to collect discrete numbers of cells (10, 50, 100 cells), especially since a (much) larger input is needed from a cell suspension to obtain these numbers. For instance, assuming a trapping efficiency of 10% (?), why would 900 cells be wasted to collect 100? In addition, when having just 100 cells available, would the chip still be effective to capture 10, 50 or all 100 of these? Or is there a minimal number of cells that the system requires as an input?
3. An important concern is the authors claim that many proteins were identified in low-input samples that are typically of low abundance, and therefore difficult to detect. For instance (line 212) they say that EGFR, NRAS, CDK4 and TP53 were 'readily identified', however these proteins were not identified in a published proteome analysis of B-cell leukemia cells, covering ~6000 proteins <https://doi.org/10.1074/mcp.RA117.000539>. Therefore it is highly unlikely that said proteins were among the top-2000 proteins in the author's sample. Similarly, it is very unlikely that proteins mentioned in line 264-286, including EGFR and TP53, were solidly identified in single cells. This raises a crucial question about the confidence of protein identification in single cells, which heavily leans on transfer of peptide identity from the library. Ideally, to make this more credible, annotated MS2 spectra should be shown for these peptides/proteins. Alternatively, but only serving as a sanity check, authors should indicate the expression level of said proteins in in-depth proteomic data

obtained from bulk cells, which should then indicate that they are expressed among the top-1000 most abundant proteins in this cell type.

Other comments:

4. Although with regard to text and figures the emphasis of the manuscript is on performance of the chip, the real innovation is in the chip itself (and not in e.g. advantages of DIA, which has been shown by others before). In that sense, more details could have been provided on the design and metrics of the chip (e.g. volumes) especially in comparison to alternative approaches (see comment below). Also the authors do not describe how they envision further development of the chip, e.g. to increase throughput or speed). Furthermore, a main concern relates to the claim that in single-cell analyses proteins can be identified that typically are of low abundance and therefore difficult to detect, raising some doubts about the confidence of protein identification.

5. The design and construction of the iProChip is novel, however some critical aspects are missing to give insight into its characteristics, also compared to alternative platforms. For instance, the dimensions of the chip are provided, but not the volumes of the subsequent 'chambers'. How does the volume of the chamber for cell trapping/lysis compare to the droplet size in nanoPOTS? And how to surface areas compare, potentially causing adsorptive protein losses? Can the SPE column be re-used or does it need to be re-packed for every sample? In addition, the design of the 'pillars' to trap defined numbers of cells is not described in detail. Was this based on trapping structures described before (citation missing), or were they newly designed for this purpose? In the latter case it will be interesting to hear how size, position and shape of the pillars were optimized to achieve efficient trapping.

6. Interfacing of the chip to LC-MS seems cumbersome: drying of peptides and resuspension in 5ul seems counterproductive, where all efforts have been geared towards avoiding losses and keeping volumes low. Is there any other way that peptides can be collected from the chip without drying and dilution?

7. The authors do not describe disadvantages of the iProChip, nor do they mention how its performance can be improved. For instance, processing 8 samples (cells) in parallel is nice but in the end is a low number, and the need for mixing on a plate shaker is inconvenient. In addition, the chip is not specifically designed to capture a single cell (see point 1) – is this foreseen to be implemented?

8. The workflow implemented on the chip rather strictly follows the 'classical' procedure for sample preparation that is broadly used for bulk samples, including lysis with detergent/Rapigest, protein reduction/alkylation, digestion, and peptide cleanup. Miniaturizing these steps on a chip is an achievement in itself (especially the clean-up on columns that are packed on-chip), however it is

questionable if all steps are required. Specifically, there is a clear trend to simplify the procedure since requirements for processing single cells and bulk samples are different. For instance, lysis of single cells can be done by a freeze-heat cycle in regular buffer even omitting reduction/alkylation (e.g. <https://doi.org/10.1101/399774>; PMID 33504367; <https://doi.org/10.1101/2020.12.22.423933>) resulting in minimal contamination avoiding the need for any clean-up. It will be interesting if the authors can comment how the iProChip can be adjusted to accommodate simplified workflows.

9. Although it is not the intention of this manuscript to compare iProChip to nanoPOTS, it needs to be seen if overall performance of iProChip exceeds that of nanoPOTS, which is also determined to a large extent by the LC-MS system at the back end. Nevertheless, in this light authors need to be careful in their statement with regard to 'ultra-streamlined' and 'ultra-sensitivity', for 2 reasons: i) the level of streamlining and sensitivity is similar in both approaches, with regard to the number and type of handling steps that are included in the procedure, and the number of proteins that can be identified; ii) on an absolute scale, and at least in theory, streamlining and sensitivity can be much improved, thereby going beyond the 'ultra' classification – for which it will then be hard to find an appropriate term. In addition, throughput of the presented system is limited when compared to array-based platforms (8 vs 96 samples in parallel).

10. Line 115-120: discrete numbers of cells can be trapped from a cell suspension, however the efficiency of trapping is unclear. That is, what is the absolute number of cells that need to be input to capture 10/100/1000 cells?

11. Line 132: mixing is rather slow, and also other steps in the procedure do not seem particularly fast. It will be helpful if the authors can include a scheme indicating the time for each handling step, including overhead time, as well as net time for processing 1 sample (or 8 in parallel) from cell trapping to peptide collection.

12. It is surprising that digestion is performed for 16h, which is typically shorter even for bulk samples. I would expect that digestion time can be reduced benefitting from low volumes and high enzyme/substrate ratio. Did the authors look into this?

13. Line 508: LC gradients used for separation of peptides obtained from single cells is extremely long. All in all this amounts to ~3h per cell, i.e. allowing to do 8 cells per day. It is very unlikely that this cannot be shortened, and it is very unclear why the authors chose this type of gradient.

14. Line 312: the comparison of the data to those by Rieckmann et al does not seem very relevant, since they analysed immune cells, not leukemic/cancer cells, hence this is not representative for the MEC-1 cell line.

15. Line 558-560: this sentence needs some clarification: does this mean that the authors require at least 3 peptides to be quantified per protein? And what does 'peak area at MS2 level' mean exactly? Is it the cumulative area of all detected fragments for each peptide? And what does 'minor peptide grouping' imply?

Reviewer #3 (Remarks to the Author):

The authors present an engineering marvel in the form of a chip based on multilayer soft lithography to prepare samples comprising one or more cells for proteome analysis. If this were submitted to e.g. Analytical Chemistry or Lab on a Chip I would be enthusiastic about its publications based on the engineering and novelty. However, to meet the bar for publication in Nature Communications, the work should also provide a meaningful advance over the state of the art, which this does not. The approach has numerous and severe downsides, which impede practical implementation beyond this proof of concept demonstration and which the authors neglect to mention. Suggestions for improvement of the manuscript and a more comprehensive discussion of both the benefits and limitations of the approach are provided below:

1. First, these devices are incredibly labor intensive to fabricate, failure prone and single use. Once we take into account the many, many hours spent to cast PDMS, align and bond the layers, punch holes for valves and reservoirs, clean out the holes to eliminate debris, manually make the 34 connections required for fluidic control, pack SPE media and passivate the surface of the chip for 1 h with BSA prior to use, all just to prepare 9 samples, it is abundantly clear that there is nothing "ultra-streamlined" about this platform and that rather this is likely the most difficult way to prepare proteomic samples that has ever been developed. In contrast, other microproteomics approaches based on micro and nanowell plates are arbitrarily scalable to dozens or hundreds of samples with either reusable or disposable devices. As such, the minimal manual intervention required during processing in competing approaches is orders of magnitude less than the up front cost required by this approach before initiating the preparation.

2. A primary rationale for miniaturizing sample preparation for proteomics is to minimize surface exposure. Unfortunately, enclosed microfluidic devices have a large surface-to-volume ratio that results in high surface adsorption from small volumes. The authors should calculate and compare the surface area that they expose their sample to, both in the trapping/lysis chamber, the subsequent digestion chamber, and the collection reservoir to the ~1 mm² surface exposure of other small-sample proteomics studies.

3. Another challenge with an enclosed microfluidic device and one utilizing sized-based cell trapping is the limitation regarding what samples can be analyzed. For example, the platform is incompatible with laser microdissection and FACS sorting, which are both widely utilized for cell isolation. The cell trap geometry may need to be modified for each different cell size, and a single small cluster of undissociated cells can foul the channel and render that portion of the device useless. An open well format is readily interfaced with common cell isolation techniques and can easily analyze single cells from tissue sections in addition to dissociated cells. Further, trying to analyze single cells based on trapping chambers of 10 traps or more requires the researcher to “get lucky” to trap a single cell. Since the title indicates that single cell proteomics is the primary objective for the work, I wonder how many chips would have to be fabricated and tested to analyze 100 cells. I also wonder how the chip would perform with primary cells that may not behave as nicely as the cultured cells used here. The authors should be up front about these limitations, which do not exist for open well plate based methodologies.

4. The authors need to present proteome coverage from a blank control containing cell supernatant and going through all the same sample preparation steps. Otherwise, we cannot know whether the identified proteins originated from a cell or were inflated by cellular debris and the contents of lysed cells present in the supernatant.

5. PDMS is gas permeable. The authors should discuss how they prevented evaporation through the channel walls during extended incubations.

6. Reporting total coverage across multiple replicates as the authors do in the abstract and elsewhere in the manuscript is an artificial means of reporting greater coverage than is actually obtained. The reported coverage using this approach is not meaningful and is not the convention of the field except for TMT studies. This is especially critical as the authors are comparing coverage of their platform identified across replicates to average proteome coverage reported in other studies.

7. Where average coverage is reported (e.g., line 230), the authors should indicate how many replicates this average is based on.

8. The ability to directly image cells before analysis (line 235) is common to most other single cell proteomics platforms as well, so this is not an actual advantage.

9. In the abstract and elsewhere, the authors refer to 5 orders of proteome coverage. Do they mean dynamic range?

10. On page 2, line 42, the statement “which theoretically extends to the single cell sensitivity” is an outdated statement since single-cell MS proteomics is now well established with many papers published.

11. The coverage reported for nanoPOTS on p. 3 of 1517 identified for ~10 cells is inaccurate. 1517 identifications includes only those identified by MS/MS and excludes those identified by library matching using match between runs. The number of proteins identified is actually 3092. The authors could potentially argue against including identifications based on library matching, but since their DIA method also relies on library matching, this would be disingenuous.

12. Similarly, the number of proteins identified on average per single cell was 1475 in Ref. 21 when library matching was included, which should be acknowledged by the authors.

13. On p. 3, line 64, the statement should include the underlined text or something similar to recognize that not all microfluidic devices incorporate hydraulic actuations. "Microfluidic devices based on multilayer soft lithography use custom chip integration and hydraulic actuations to achieve precise μL -to-nL fluid manipulation"

14. On p. 3, line 75, I don't understand what this statement means: "enabled all retrospective peptides mapping against spectral libraries and offered superior coverage". Please clarify.

Point-by-point responses to Reviewers' comments

Highly streamlined single-cell proteomics by all-in-one chip and data-independent acquisition mass spectrometry

We thank all reviewers for their kind and constructive comments and suggestions to improve our manuscript. We have attempted to address the comments and questions either in the body of the revised manuscript, or in the point-by-point reply letter. We hope that our revisions and explanations adequately respond to the comments and further enhance the manuscript. In addition to the point-by-point reply shown below, the revised text is highlighted in red font in the revised manuscript.

Reply: blue

Revised text in main text/Method/SI

Reviewers' comments:

Reviewer #1 (Remarks to the Author):

The authors report a novel microfluidic platform for isolation of intact cells, followed by cell lysis and peptide digestion/desalting. The sample of peptides can subsequently be retrieved from the device and subject to DIA mass spectrometry. In the manuscript, authors demonstrate that their device can capture different numbers of cells in each sample chamber by tuning different experimental parameters and device design parameters, and can achieve capture of single cells if desired. They also show the device performance metrics in terms of mixing efficiency in the peptide digestion chamber, and show desalting performance. The performance of the coupled DIA m/s is also demonstrated with triplicate experiments using 13 or 14 cells, and triplicate experiments of single cells.

1. The device itself quite novel - it quite nicely accomplishes cell capture, cell lysis, peptide digestion and desalting. But it is unclear from the manuscript whether this device actually can practically address the issue of limited sample quantity for protein mass spec. Most of the time, the initial sample quantity (i.e. the number of cells available for processing) poses a problem because the sample processing steps introduce too much loss, and the experiment cannot be carried out if there are too few cells to begin with as there will not be enough in the final ready-for-ms library after all the processing steps. Although the authors show that in the device, they can capture as little as single cells, there are usually losses associated with loading of the microfluidic device. To show that this device is really able to address the sample input quantity issue, it would be better to discuss how much total input is loaded into the device for capture. I.e. what is the minimum cell number that is needed, and how that compares to standard practice or current frontier technology in mass spec.

Reply: We thank the reviewer for the comment and appreciate the importance of our device and effort. As the reviewer pointed out, initial sample quantity indeed poses challenges for performing proteomics on limited input samples, which is also why and what we hope to address in this study. Conventional cell sorters such as FACS have been widely used to isolate cells for most existing microproteomics approaches. Although FACS is capable of sorting a single cell, it typically requires >tens-to-hundred μ L of cell solution and relatively high cell density for effective instrument operation, which can at times be infeasible to prepare samples due to limited input quantity.

To show that our device can significantly **improve low input quantity** limitation, we have performed additional experiments using the iProChip. To characterize minimal numbers of cell input in this chip, we prepared samples containing PC-9 cells with a cell density of 25 cells/uL and a total volume of either 5 or 10 uL (which equals 125 or 250 cells in each sample vial). Note that such low cell density and volume pose challenges for conventional cell sorting instruments since they often require tens-to-hundreds uL and higher cell density for effective operation. The cell vial was then connected to the chip, and directly flowed into the cell chamber to determine sample volume and cells needed for capturing desired cell numbers (1, 5, 10, 50 and 100 cells) in individual chambers. As shown in **Response Figure 1** and **Response Table 1**, our result showed ~4-8% cell usage efficiency (defined as number of trapped cells/number of total injected cells) for capturing 1, 5 and 10 cells and 33-44% for trapping 50 and 100 cells, respectively. As such, by using as low as ~250 cells in total as initial sample quantity, iProChip is capable of parallel processing of 9-10 single cell samples for proteomic analysis, which will be very difficult (if possible) to achieve using any external cell sorter. It is foreseeable that such cell usage efficiency can be further increased by collecting cell flow-through and re-inject into the chip if desired.

Response Figure 1. The cell usage efficiency for capturing 1, 5, 10, 50 and 100 cells as determined by using either 5 or 10 uL cell solutions with a density of 25 cells/uL.

Response Table 1. Characterization of cell usage efficiency and input cells needed for operating the iProChip for 1, 5, 10, 50 and 100 cells.

Cell density (cells/uL)	Operational Volume (uL)	Injected Time (s)	No. of cells in operational volume ^a	Injected volume (uL) ^b	No. of cells injected	No. of cells trapped ^c	Cell usage efficiency (%)
25	5	3 ± 1	125	1 ± 0.2	25 ± 06	1 ± 0	4 ± 0.00
25	5	12 ± 3	125	4 ± 1.0	100 ± 25	5 ± 1	5 ± 0.04
25	10	24 ± 7	250	5 ± 1.0	125 ± 25	9 ± 1	8 ± 0.04
25	10	39 ± 7	250	6 ± 2.0	150 ± 50	50 ± 3	33.3 ± 0.06
25	10	36 ± 3	250	9 ± 1.0	225 ± 25	100 ± 4	44.4 ± 0.16

a. Operational volume is the volume of cell solution for running the chip.

b. Injected volume is determined by measuring the remaining volume of cell solution in the vial.

c. Cell usage efficiency is defined as number of trapped cells/number of total injected cells X 100.

In the revised manuscript, we have provided the following revisions:

(1) Updated the text (Page 5, Line 136-140): *“The cell trapping capability of iProchip was also evaluated to characterize the cell usage efficiency (defined as numbers of trapped cells/numbers of total injected cells) and minimum numbers of cells needed for iProChip operation (Methods). Using a*

total of either 5 or 10 μ L cell solution (25 cells/ μ L), the result showed that cell usage efficiency ranges from ~4 - 44 % for capturing 1 - 100 cells (Supplementary Table 2 and Supplementary Fig. 4).”,

(2) We have added a Method section (Page 16, Line 539-552): “In order to characterize minimal numbers of cells needed to operate in this chip, we prepared vials containing PC-9 cells with a cell density of 25 cells/ μ L, and used a total volume of either 5 or 10 μ L (which equals to 125 or 250 cells in each vial). The cell vial was then connected to the chip, and directly flowed into the cell chamber at 3 psi to determine how much sample volume and cells is needed for capturing desired cell numbers (including 1, 5, 10, 50 and 100 cells) in individual chambers. The cell solution was injected into each chamber so that a fraction of cells were captured by wedge-shaped traps while remaining cells simply passed by to the waste-outlets. A real-time chip monitoring by a brightfield microscope was used to inspect the cell flow and capture; the valves were closed when desired numbers of cells were captured in the chambers. Note that the imaging-ready feature of iProChip enables easy quantification of trapped cells. At this cell density, it took 3-40 s to capture 1-100 cells respectively (Supplementary Table 2 and Supplementary Fig. 4). Our result showed 4-8% cell usage efficiency for capturing 1, 5 and 10 cells and 33-44% for trapping 50 and 100 cells.”

(3) We have added the Response Table 1 (as Supplementary Table 2) and Response Figure 1 (as Supplementary Figure 4) in the revised manuscript accordingly.

2. Secondly, there was no raw data showing the effectiveness of the peptide desalting. Although the quantity of peptide recovered is an important metric, some validation of the salt or pH before and after the column is needed to confirm that the desalting has worked. Furthermore, the experiments showing the efficiency of peptide recovery only goes down to 0.2 μ g of loaded peptide, which is equivalent of peptides from on the order of ~1000 cells. But the experiments conducted throughout the paper are using from 1 to 100 cells, which would generate orders of magnitude less peptides. It has been well-demonstrated that at microfluidic volumes and with ultra-low input quantities of sample, adsorption is often a major contributing factor to sample loss when using microbead packed columns. The recovery efficiency of 89% may no longer hold at single cell level (200pg), but given the purpose of the device, it seems important to test the recover efficiency at the levels of peptide loading that the device will be used for.

Reply: We thank the reviewer for raising the comment regarding the peptide desalting, and agree it is important to characterize as many aspects of the embedded desalting module as possible. Unfortunately, by traditional protein quantification methods such as the BCA assay, our attempt to quantify recovery efficiency at <1000 cells did not succeed due to the challenge of processing extremely limited amounts of peptides at the level of <1000 cells, which goes beyond the detection limit of BCA assay. Nevertheless, as per reviewer’s suggestion, we performed additional experiments to compare the desalting effects of ~10 cells through the iProchip with or without desalting column. As shown from the following profiles (**Response Figure 2**), the total ion chromatogram (TIC) of the 10-cell sample without passing the desalting column shows the presence of detergent peaks which suppressed the peptides ions. By the desalting step in the iProchip, the desalted sample shows very reproducible TIC profiles of typical peptide ions. The results revealed the effective removal of detergent/buffer by the desalting function in the iProchip at 10-cell level. This observation is consistent with the literature that the presence of Rapigest surfactant in sample may suppresses the MS signal, which can be removed after desalting column in the iProchip. This additional comparison revealed the effect of desalting for good LC-MS/MS signals.

Response Figure 2. Comparison of the total ion chromatogram (TIC) profiles of (a) desalted (DS) sample and (b) non-desalted (ND) 10-cell samples.

3. Third, the comparison to bulk experiments is lacking, which is particularly problematic when the number of replicates of low-cell number experiments is also low ($n=3$ for single cells or $n=6$ for 13/14 cells, if we combine the DIA and DDA experiments). The reason why comparisons to conventional bulk (or even mini bulk such as 1,000 or 10,000 cells) is important is to be able to determine the sensitivity and robustness of the low cell number assay. Single cell assays are usually much lower in sensitivity than bulk assays; but as the user, it is important to know the degree of loss of sensitivity. A comparison to the bulk can tell us what fraction of the bulk proteins/peptides can be captured with the low-cell-number assay, and whether the peptides that are lost are due to low abundance, or subject to other technical bias. Without the bulk comparison, it is impossible to judge. In addition, usually bioinformatically pooled low-cell experiments are able to replicate bulk experiments due to the nature of such sampling. Therefore, increasing the number of replicates to a level that can be used to bioinformatically re-construct the bulk is a useful way to demonstrate the technical sensitivity of the assay. Currently there is no comparison to any external reference dataset generated using conventional gold standards. Given that the cells used were all cell lines, even a comparison to existing bulk datasets in the literature would be acceptable (in fact the authors mention such a study by Rieckmann et al, but do not perform any kind of comprehensive analysis to show how their data compares).

Reply: We thank the reviewer for the comment and suggestion. For microproteomics based studies, bulk comparison is often precluded due to the challenges associated with sample preparation at such small-scale, that is where techniques like microfluidics-based platforms leverage.

Comparison to Bulk Sample Following the suggestion, we have designed three experiments to prepare the “Bulk sample” and compared with our data. (1) We added new experiments to profile bulk samples containing ~100 cells in **vial-based preparation** (in triplicate). As expected, the identification results were much worse. Compared to the 4,722 identified proteins obtained by iProchip (**Manuscript Figure 4a**), only an average of 86 protein groups were obtained when we processed ~100 cells in the vial-based format (**Response Figure 3**). (2) We also performed a **dilution-based processing of cells in bulk preparation** (~50 ug lysate) and only injected small aliquots corresponding to 1.5 ng (~10 cells) and 15 ng (~100 cell) of digested peptides for triplicate analysis. While the result showed a comparable protein and peptide identification at 15 ng in both iProChip and bulk-dilution preparation, at 1.5 ng, 2-fold higher identification results were obtained from iProChip workflow compared to the bulk-dilution preparation (**Response Figure 4**). This comparison demonstrated that performance gain of iProChip versus bulk-dilution processing appeared much higher for smaller cell numbers.

(3) Additionally, we compared our data (1-106 cells) with large-scale ~1 ug PC-9 samples (around ~6000 cells), that can be considered as **a conventional proteomics bulk sample** processed in bulk. Compared to this large-scale ~1 ug study at the depth of >6,000 proteins., 90% - 96% of the proteins identified from lower cells (1-106) were in common with some additional proteins uniquely identified from iProChip (**Response Figure 5a**). Interestingly, at 106 cells level, it was noted that 75% of 6000 proteins were already identified from the 106 cells processed through iProChip, suggesting the bulk-like comparable identification at such a small sample scale. In addition, the fraction of the bulk proteins captured with the low-cell-number assay indicated an increasing trend correlated with increasing cell numbers (**Response Figure 5b**). Further analysis of abundance among commonly identified proteins shows that mainly abundant proteins were observed at lower cells, suggesting challenges of detecting low abundant proteins at low cell samples (**Response Figure 5c**). These additional insights enhance the understanding about comparison of low-cell-assay versus bulk-sample processing. Lastly, the Reviewer suggested to compare our study to the work by Rieckmann et al which largely differs in terms of objectives; thus, performing such comprehensive analysis is beyond the scope of current study. Furthermore, we want to clarify that the said study was cited for providing the insights related to the coverage of surface markers in immune cells.

Response Figure 3. Processing of ~100 cells in vial-based preparation.

Response Figure 4. Processing of dilution aliquots through bulk-dilution processing and proteome coverage comparison. The number of proteins and peptides identified across average of triplicates processed through bulk-dilution samples and their comparison with 14 cells, and 106 cells as processed through iProChip.

Response Figure 5. Comparison of triplicate analyses results obtained from iProChip (1-100 cells) and bulk sample preparation. (a) Venn diagram showing proteins commonly identified from bulk sample preparation and iProChip. (b) Number (fraction) of proteins identified in bulk sample preparation that were also captured in the low-cells-number assay using iProChip. (c) Comparison of protein abundance distribution in commonly overlapped proteins among low-cells-data from iProChip and bulk analysis.

In response to this comment, we have revised the manuscript (Page 8, Line 267-281).. *“To compare the iProChip with bulk sample processing, we performed dilution-based processing in vial (~50ug lysate) and injected small aliquots corresponding to 1.5 ng (~10 cells) and 15 ng (~100 cell) for LC-MS/MS analysis (Method). While the result showed a comparable protein and peptide identification at 15 ng in both iProChip and bulk-dilution preparation, 2-fold more identifications were obtained from iProChip workflow at 1.5 ng. (Supplementary Fig. 11). This higher performance gain of iProChip for smaller cells further validates the efficiency of our approach beneficial for limited cells. Additionally, when the data (1-106 cells) was compared to a conventional proteomics sample (~1ug PC-9, ~6000 cells, >6,000 proteins), 90%-96% of the proteins identified from lower cells (1-106) using iProChip were common (Supplementary Fig. 12a). Interestingly, 75% of these 6000 proteins were identified from 106 cells processed by iProChip, suggesting the bulk-like comparable identification at*

a small sample scale. In addition, the fraction of the bulk proteins captured with iProChip indicated an increasing trend correlated with cell numbers (Supplementary Fig. 12b). Further analysis of protein abundance shows that mainly abundant proteins were observed at lower cells, suggesting challenges of detecting low abundant proteins at low cell samples (Supplementary Fig. 12c)."

We have also added the **Response Figure 4** (as **Supplementary Figure 11**) and **Response Figure 5** (as **Supplementary Figure 12**) in the revised manuscript accordingly.

4. There is a similar problem with the number of replicates being too low, and trying to estimate the FDR or accuracy of the method (e.g. comparison of overlapping peptides between different experiments). For the number of proteins being sampled, a triplicate experiment is actually likely to under-estimate the accuracy of the assay; a larger n that when summed can generate bulk-like data is a better demonstration of the accuracy. With larger n, further statistical methods can be used to gauge the accuracy of the assay, such as looking at the CV of peptides detected across many cells, etc. While I appreciate the challenges of performing many single cell or 10 cell mass spec experiments to establish such statistical power, I strongly feel it is necessary to establish the reliability and accuracy of the method.

Analysis methods for establishing sensitivity and accuracy of the single cell -omics compared to bulk has been demonstrated extensively in the single-cell RNA-sequencing literature. Papers published from 2009-2015 on single-cell RNA-seq feature many ways to compare single-cell RNA measurements to bulk RNA measurements. I feel that the same analyses need to be applied to single-cell proteomics to establish feasibility. Authors can reference those methods as a roadmap.

Analytical Merits of Assay We thank the reviewer for the concern of estimating the assay accuracy using "lower" sample size, yet acknowledging challenges for proteomics profiling of lower cells. In the present study, iProChip is introduced for the first time as an alternative approach to complement existing single cell proteomics workflows. Therefore, we had followed the standard practice in MS-based proteomics to use triplicate dataset for estimating the assay performance. We would like to clarify that the protein identification was performed at 1% false-discovery rate (FDR) at precursors and protein levels that are commonly adopted gold standard to ensure stringent protein identification. About the number of replicates, all our datasets were produced in triplicates, which is in agreement with other similar microproteomics studies. We agree that if available, larger sample size can allow further examination by statistical methods to better represent assay accuracy. Following the reviewer's suggestion, we have performed an additional 10-cell experiments (in triplicate), and compared its result with the 10-cells data presented in the submitted manuscript (Manuscript Fig. 4a and Supplementary Fig. 8b). Although the new experiments were done using a different chip and a different cell batch, as shown in the figure, a highly reproducible protein identification result was observed (**Response Figure 6a**), suggesting robust and reliable operation in iProChip. Additional comparison among triplicate data also shows high overlaps of protein and peptide in both experiments (**Response Figure 6b**). Moreover, comparison of total proteins from this current experiment with the manuscript data showed more than 71% overlapping proteins, suggesting good reproducibility among different batches of experiments (**Response Figure 6c**). The additional data suggests good reproducibility between different experiments to ensure the robustness of the iProChip platform and its repeatability between chip-to-chip experiments.

We also appreciate the reviewer's suggestion to follow single cell-omics literature for comparison to bulk samples. This experiment has been added in **Reply 3** (**Comparison to Bulk Sample**).

Response Figure 6. Comparison of identification coverage and performance of proteomic profiling of additional 10-cells data (PC-9). (a) comparison of identification of protein groups and peptides in additional 10-cell data with data in the submitted manuscript. (b) The Venn diagram classification of proteins and peptides among triplicates in both experiments (c) Comparison of total protein groups among both experiments.

Overall, I feel that the novelty of this manuscript is high, but much more rigour in terms of experimental design and data analysis are needed to establish this method as viable, feasible, and accurate for single-cell proteomics.

Reply: We thank the reviewer for acknowledging the novelty of our approach, and believe the additional data and assay verification per the request of the review comments shall further clarify the features of our approach and improve the manuscript.

Minor comments:

Line 26 - “enables profiling of 1160 protein groups from triplicate analysis of a single mammalian cell” - this is written in a confusing way to suggest that the same single cell was analyzed 3 times. suggest re-writing to remove this confusion, e.g. “analysis of three individual/single mammalian cells”. It is impossible to have triplicate of the same single cell...

Reply: We thank the reviewer for the correction of the confusing statement and suggestion of revision. We originally intended to emphasize that this protein identification result was obtained from the accumulated results of three different/individual replicates (the same single cell was not analyzed 3 times).

In the revised manuscript, we have revised the sentence accordingly (Page 2, Line 28-29): “.....the iProChip-DIA enables profiling of 1160 protein groups from triplicate analysis of three individual single mammalian cells”.

Line 153 - suggest changing or removing the word “prudently”; contextually this is not the right word use here.

Reply: We thank the reviewer for the comment and kind suggestion. We have removed the word “prudently” from the text in the manuscript.

Figure 2f - consider adding the unity line to the graph and making the graph square, with the axes using the same marks to enhance readability

Reply: We thank the reviewer for the kind suggestion. We have updated Figure 2f according to the suggestion, as shown below.

Figure 2f. Desalting recovery efficiency of the on-chip SPE column. Error bars, s.d. ($n= 3$ independent experiments).

Figure 3a - are the columns in the top and bottom panels matched in terms of the experiment? Suggest changing labels to experiment number followed by a parenthesis with notation of number of cells.

Reply: We thank the reviewer for the suggestion. It is correct that the top and bottom panel corresponds to the same experiment. The figure shows triplicate experiments which have the exact number of cells, i.e. 13 or 14 cells to compare the identification results (both protein groups and peptides) from DDA and DIA modes. Following the suggestion, we have updated the label to “DDA1 (13), DDA2 (13), DDA3(13), DIA1 (14), DIA2 (14) and DIA3 (13)” in the revised Figure 3a, as shown below.

Figure 4b - the pink dot is quite confusing. Suggest changing the color of the right y-axis (cell area) to pink or some other color that matches the dots; also suggest using cell diameter instead of area, or

an estimated cell volume based on the area measured by the image. Diameter is more commonly used in biology to convey cell size.

Reply: We thank the reviewer for the suggestion. We have changed the color of the right y-axis to match that of the data points (both in orange now), and presented this data as “cell diameter” to replace “cell area” for presenting cell size in the revised Figure 4b and Supplementary Figure 8a.

Figure 4b. Cell size and triplicate analysis results of identified protein groups from single cells.

Supplementary Figure 8a. Peptide identification summary for a single PC-9 cell.

Reviewer #2 (Remarks to the Author):

In this manuscript, Gebreyesus and colleagues present a microfluidics approach for proteomic sample preparation of small cell populations – down to single cells. Specifically, they have constructed a PDMS chip (iProChip) that captures a pre-defined number of cells (5-10-100), and executes subsequent handling steps including cell lysis, protein reduction/alkylation, digestion and clean-up, which can be performed for 8 samples in parallel. Resulting peptides were analyzed by LC-MS using a data-independent approach (DIA), demonstrating the ability to identify ~4000 proteins from 100 cells, down to ~1000 proteins from single cells. They show that this includes various biologically and clinically relevant proteins, including cell surface proteins and proteins involved in cell signaling. They conclude that the integrated sample preparation by the iProChip provides a powerful workflow for reproducible proteome analysis of low-input samples down to single cells.

This is an interesting manuscript, providing an alternative workflow compared to previous work using micro-pipetting in open array-based systems (e.g. nanoPOTS, OAD). This is a welcome development to show that alternative approaches are conceivable that may have advantages depending on the use case. The authors show the construction of the chip with an assessment of performance

(sensitivity, consistency) when used in conjunction with DIA-MS. A real biological application is lacking – which is fair at this stage, instead showing proof of principle. Yet there are a few main concerns with regard to fitting the chip in a genuine single-cell workflow, and with regard to the identification of low-abundance (signaling) proteins presented in the manuscript – along with various smaller remarks:

Reply: We thank the reviewer for acknowledging our effort for developing an alternative approach to complement existing approaches towards single cell proteomics analysis. We are excited by the demonstrated performance in this study, as iProChip design is flexible and can readily incorporate other microfluidics features and proteomics methods to realize more advanced proteomics profiling at single cell level. We also thank the reviewer for providing constructive comments to improve our study and manuscript. In the following, please find our response to your comments:

Main concerns:

1. Although the authors present their system as a way to study single cells, strictly speaking the chip is not designed to do so. In fact, it is quite unclear how single cells were obtained: Line 230 states that this is done using a trap for 10 cells, which probably means that trapping needs to be monitored visually, and stopped as soon as exactly 1 cell has been trapped? This would denote a cumbersome procedure of which it is hard to imagine how one would achieve this for hundreds of cells?

Reply: 1.1 We thank the Reviewer for raising the comment and concerns. The first generation of chip was designed with capacity for proteomics analysis of 1-100 cells. As such, in this particular chip, single cell trapping was achieved by controlling low cell density (25 cells/uL) with optimized flow pressure of 3 psi. In response to reviewer's comments, we have performed additional experiments which showed that using as low as 5uL cell solution with a density of 25 cells/uL, iProChip can routinely realize single cell capture in a few seconds and achieve good cell usage efficiency of ~4% for single cell operation (**Response Table R1**; see also Reviewer 1, comment 1). Note that the imaging-ready feature of iProChip enables absolute quantification of exact sample size being analyzed. Therefore, using as low as ~250 cells in total as initial sample quantity, iProChip is capable of parallel processing 9-10 single cell samples for proteomic analysis, which will be highly challenging (if possible) for most existing single cell workflows since they typically require the use of external cell sorting such as the FACS. To the best of our knowledge, a platform with integrated cell trapping capacity, exact cell counting and verification by cell imaging has not been demonstrated in existing single cell workflows.

Following the comment, we have added description on the cell capture process in Method section of the revised manuscript (Page 16, Line 539-552): *"In order to characterize minimal numbers of cells needed to operate in this chip, we prepared vials containing PC-9 cells with a cell density of 25 cells/uL, and used a total volume of either 5 or 10 uL (which equals to 125 or 250 cells in each vial). The cell vial was then connected to the chip, and directly flowed into the cell chamber at 3 psi to determine how much sample volume and cells is needed for capturing desired cell numbers (including 1, 5, 10, 50 and 100 cells) in individual chambers. The cell solution was injected into each chamber so that a fraction of cells were captured by wedge-shaped traps while remaining cells simply passed by to the waste-outlets. A real-time chip monitoring by a brightfield microscope was used to inspect the cell flow and capture; the valves were closed when desired numbers of cells were captured in the chambers. Note that the imaging-ready feature of iProChip enables easy quantification of trapped cells. At this cell density, It took 3-40 s to capture 1-100 cells respectively. The result is summarized in Supplementary Table 2 and Supplementary Fig. 4. Our result showed 4-8% cell usage efficiency for capturing 1, 5 and 10 cells and 33-44% for trapping 50 and 100 cells."*

1.2 On a side note, inspired by the promising result using the iProChip, we recently developed a new chip dedicated for single cell proteomics application and demonstrated it can realize faster and easier single cell capture (**Response Figure 7**, unpublished). Note that in this new chip, each chamber contains only 1 trap so that one cell can be precisely captured in each unit in a shorter time, which will further facilitate full automation of overall workflow and increase cell usage efficiency.

Response Figure 7. Single cell capturing chamber from PDMS chip before (left) and after (right) capturing one PC-9 cell.

2. Related to the previous point, it is unclear what application the authors have in mind by providing the ability to collect discrete numbers of cells (10, 50, 100 cells), especially since a (much) larger input is needed from a cell suspension to obtain these numbers. For instance, assuming a trapping efficiency of 10% (?), why would 900 cells be wasted to collect 100? In addition, when having just 100 cells available, would the chip still be effective to capture 10, 50 or all 100 of these? Or is there a minimal number of cells that the system requires as an input?

Reply: We thank the reviewer for raising these questions and comments. The iProChip was initially designed to serve as a single, integrated platform for performing robust and sensitive proteomic analysis for the range of 1-100 cells, when only extremely limited input cells such as circulating tumor cells and clinical samples are available, which pose challenges in conventional proteomics workflow involving the use of external cell sorter.

We agree with the reviewer that it is important to know how much input cells will be needed for operating the iProChip. In response to the comment, we thus performed additional experiments to evaluate the minimal input cell quantity and cell usage efficiency (defined as number of trapped cells/numbers of total injected cells). We firstly prepared samples containing PC-9 cells with a cell density of 25 cells/ μ L, and used a total volume of either 5 or 10 μ L (which equals 125 or 250 cells in each sample vial). Note that such low cell density and volume pose challenges for the conventional cell sorter since they often require tens-to-hundreds μ L and higher cell density for effective operation. The cell vial was then connected to the chip, and directly flowed into the cell chamber to determine how much sample volume and cells is needed for capturing desired cell numbers (1, 5, 10, 50 and 100 cells) in individual chambers. The result is summarized in **Response Figure 1** and **Response Table 1** shown below. Our result showed 4-8% cell usage efficiency for capturing 1, 5 and 10 cells and 33-44% for trapping 50 and 100 cells, respectively. As such, by using as low as \sim 250 cells in total as initial sample quantity, iProChip is capable of parallel processing of 9-10 single cell samples for proteomic analysis, which will be very difficult (if possible) to achieve using any external cell sorter. It is also

foreseeable that such cell usage efficiency can be further increased by collecting cell flow-through and re-inject into the chip if desired.

Response Figure 1. The cell usage efficiency for capturing 1, 5, 10, 50 and 100 cells was determined by using either 5 or 10 μL cell solution with a density of 25 cells/ μL .

Response Table 1. Characterization of cell usage efficiency and input cells needed for operating the iProChip for 1, 5, 10, 50 and 100 cells.

Cell density (cells/ μL)	Operational Volume (μL)	Injected Time (s)	No. of cells in operational volume ^a	Injected volume (μL) ^b	No. of cells injected	No. of cells trapped ^c	Cell usage efficiency (%)
25	5	3 ± 1	125	1 ± 0.2	25 ± 06	1 ± 0	4 ± 0.00
25	5	12 ± 3	125	4 ± 1.0	100 ± 25	5 ± 1	5 ± 0.04
25	10	24 ± 7	250	5 ± 1.0	125 ± 25	9 ± 1	8 ± 0.04
25	10	39 ± 7	250	6 ± 2.0	150 ± 50	50 ± 3	33.3 ± 0.06
25	10	36 ± 3	250	9 ± 1.0	225 ± 25	100 ± 4	44.4 ± 0.16

a. Operational volume is the volume of cell solution for running the chip.

b. Injected volume is determined by measuring the remaining volume of cell solution in the vial.

c. Cell usage efficiency is defined as number of trapped cells/number of total injected cells X 100.

In the revised manuscript, we have provided the following revisions:

(1) Updated the text (Page 5, Line 136-140): “The cell trapping capability of iProchip was also evaluated to characterize the cell usage efficiency (defined as numbers of trapped cells/numbers of total injected cells) and minimum numbers of cells needed for iProChip operation (Methods). Using a total of either 5 or 10 μL cell solution (25 cells/ μL), the result showed that cell usage efficiency ranges from ~4 - 44 % for capturing 1 - 100 cells (Supplementary Table 2 and Supplementary Fig. 4).”,

(2) We have added a Method section (Page 16, Line 539-552): “In order to characterize minimal numbers of cells needed to operate in this chip, we prepared vials containing PC-9 cells with a cell density of 25 cells/ μL , and used a total volume of either 5 or 10 μL (which equals to 125 or 250 cells in each vial). The cell vial was then connected to the chip, and directly flowed into the cell chamber at 3 psi to determine how much sample volume and cells is needed for capturing desired cell numbers (including 1, 5, 10, 50 and 100 cells) in individual chambers. The cell solution was injected into each chamber so that a fraction of cells were captured by wedge-shaped traps while remaining cells simply passed by to the waste-outlets. A real-time chip monitoring by a brightfield microscope was used to inspect the cell flow and capture; the valves were closed when desired numbers of cells were captured in the chambers. Note that the imaging-ready feature of iProChip enables easy quantification of

trapped cells. At this cell density, it took 3-40 s to capture 1-100 cells respectively (Supplementary Table 2 and Supplementary Fig. 4). Our result showed 4-8% cell usage efficiency for capturing 1, 5 and 10 cells and 33-44% for trapping 50 and 100 cells."

(2) We have added the **Response Table 1** (as **Supplementary Table 2**) and **Response Figure 1** (as **Supplementary Figure 4**) in the revised manuscript accordingly.

3. An important concern is the authors claim that many proteins were identified in low-input samples that are typically of low abundance, and therefore difficult to detect. For instance (line 212) they say that EGFR, NRAS, CDK4 and TP53 were 'readily identified', however these proteins were not identified in a published proteome analysis of B-cell leukemia cells, covering ~6000 proteins <https://doi.org/10.1074/mcp.RA117.000539>. Therefore, it is highly unlikely that said proteins were among the top-2000 proteins in the author's sample. Similarly, it is very unlikely that proteins mentioned in line 264-286, including EGFR and TP53, were solidly identified in single cells. This raises a crucial question about the confidence of protein identification in single cells, which heavily leans on transfer of peptide identity from the library. Ideally, to make this more credible, annotated MS2 spectra should be shown for these peptides/proteins. Alternatively, but only serving as a sanity check, authors should indicate the expression level of said proteins in in-depth proteomic data obtained from bulk cells, which should then indicate that they are expressed among the top-1000 most abundant proteins in this cell type.

Reply: 3.1 Cell **Type-dependent Protein Expression** Thanks for the comments to further clarify the identification result of biology-relevant proteins in our assay. This is an important issue related to the cell type-specific protein expression. According to the Human Protein Atlas (HPA) database (<https://www.proteinatlas.org/>), the protein expression levels are dependent on the cell type. In the example of EGFR protein, it has high expression in lung cancer cells (A549) and was not detected in the immune cells of blood. Therefore, we did not detect the EGFR in the MEC-1 cells, a human chronic B cell leukemia cell. We also checked the expression of TP53 in the immune cell profiles from our own B cell datasets (unpublished), which is below the detection limit in immune cell profiling. Therefore, it is expected that endogenous EGFR and TP53 may not be present or with extremely low abundance, if any, in the B-cell from CLL patients by Johnston et.al, despite that good coverage of 6000 proteins were observed in that study. As suggested by the reviewer, we have further compared the proteome coverage of our data with the B-cell leukemia cells from Johnston et. al. (*Molecular & Cellular Proteomics* 17.4, 776, 2018). As shown below (**Response Figure 8**), even though MEC-1 cells also belong to human B-cells leukemia, only 51% of protein groups commonly overlap between the two datasets. Nevertheless, the important B-cell markers (e.g., CD19, CD20 and CD22) were commonly detected in both Johnston et al., and our MEC-1 data, while they were not detected in the PC-9 data (**Response Table 2**). Similarly, the important lung cancer markers (e.g., EGFR and TP53) were not detected in MEC-1 and Johnston et al., study (**Response Table 2**). Nevertheless, EGFR and TP53 are critical proteins for lung cancer and have high expression in the lung cancer cells; therefore, they were detected in the PC-9 lung cancer cells at as low as ~5 cells. These comparisons again support the argument of cell-dependant specificity.

Response Figure 8. Venn diagram showing overlap of commonly identified proteins among Johnston et al., study and our MEC-1 data.

Response Table 2. Comparison of the identification of B-cell surface markers and important NSCLC proteins among datasets from PC-9, MEC-1 and Johnston et. al, study.

Important Protein Comparison (Our Study Vs. Johnston et al.,)

Cells	Key Proteins	Accession Number	Our study (PC9)	Our study (MEC1)	Johnston et al.,
PC9	EGFR	P00533	✓	X	X
	TP53	P04637	✓	X	X
	MAP2K1	Q02750	✓	✓	✓
	MAPK1	P28482	✓	✓	✓
	CDK1	P06493	✓	✓	✓
	ITGB1	P05556	✓	X	✓
	PGK1	P00558	✓	✓	✓
	CDK4	P11802	✓	✓	X
MEC1	CD19	P15391	X	✓	✓
	CD20	P11836	X	✓	✓
	CD21	P20023	X	✓	✓
	CD22	P20273	X	✓	✓
	CD81	P60033	✓	✓	X
	CD47	Q08722	✓	✓	✓
	CD74	P04233	X	✓	✓
	HLA-DRA	P01903	X	✓	✓
	HLA-DRB5	Q30154	X	✓	✓

We have added the **Response Table 2** (as **Supplementary Table 4**) and following text in the revised manuscript (Page 11, Line 364-372): “According to the Human Protein Atlas (HPA) database (<https://www.proteinatlas.org/>), the protein expression levels are dependent on the cell type. We have further compared the proteome coverage of our data with the B-cell leukemia cells from Johnston et. al. Even the MEC-1 cell line also belongs to human B-cells leukemia, only 51% of protein groups commonly overlapped between the two datasets (Supplementary Table 4). Nevertheless, the important B-cell markers (e.g., CD19, CD20 and CD22) were readily and commonly detected in both datasets, while they were not detected in our PC-9 data. Similarly, the important lung cancer markers (e.g., EGFR and TP53) were not detected in MEC-1 and leukemia cells (Supplementary Table 4). This supports the notion of cell-dependent specificity of protein expression”.

3.2 FDA Druggable Proteins at 1-5 Cells We would like to further clarify the description (Line 212-220): “Notably, FDA approved druggable targets for lung cancer, such as EGFR, MAP2K1, MAP2K2 and proteins involved in NSCLC pathway were readily identified in DIA, whereas only TP53 and CDK1 were detected in DDA using our approach (Fig. 3e)”. This statement aims to highlight the superior identification depth of DIA compared to the DDA mass spectrometry. We hope to evaluate the

potential applicability of our approach for clinical proteomics. Thus, the data interpretation focuses on the FDA approved druggable protein targets and the detection limit of very low abundant proteins is beyond the scope of this session.

In this study, EGFR was detected up to 5 cells, whereas TP53 was identified till single cell level. As mentioned in the manuscript *“Even at low cell numbers (14±1 cells), the drug targets EGFR, MAP2K1, MAP2K2, MAPK1, MAPK3, KIF5B, tumor suppressor TP53, and other key signaling components (KRAS, CDK4, CDKN2A, EML4, KIF5B, NRAS, BAX, RB1) were identified. In terms of sensitivity, EGFR, MAPK1, MAP2K1, MAP2K2, CDKN2A, TP53, KIF5B and GRB2 proteins were still detected down to as low as 5 cells, whereas MAP2K1, KRAS and TP53 were even identified at single-cell level.”* On the lung cancer cell model, the proteome coverage allows mapping to relevant proteins in cancer biology.

3.3. Protein Identification Stringency Notably, the identification and quantification analysis of the dataset were performed at a statistically stringent criteria of 1% false-discovery rate (FDR) at precursors and protein levels. For protein quantitation, peak areas of MS2 transitions from top six most intense fragment ions were calculated using top 3 peptide sequences per protein. Per suggestions to ensure the identification confidence, we have listed example DIA spectra below from the selected proteins relevant to lung cancer and added them in the Supporting Information (also shown below as **Response Figure 9** and **Response Table 3**).

Response Figure 9. Annotated MS2 spectra for the indicated peptides of proteins identified in 1-cell, 5-cell, and 10-cells of non-small lung cancer PC-9 cells, respectively. The corresponding peptide Q-value and quantity are shown in the Supplementary Table 3.

Response Table 3. Summary of peptide Qvalue and quantity for the indicated proteins identified in 1-cell, 5-cell, and 10-cells of non-small lung cancer PC-9 cells, respectively.

Protein Groups	Peptide Groups	Peptide Qvalue			Peptide Quantity		
		1-cell	5-cells	10-cells	1-cell	5-cells	10-cells
TP53	ELNEALELK	5.4E-03	2.1E-29	2.1E-69	1.1E+00	6.8E+03	9.6E+03
	KPLDGEYFTLQIR	N/D	5.9E-12	2.7E-08	N/D	3.8E+03	4.6E+03
	LGFLHSGTAK	N/D	2.5E-04	6.1E-03	N/D	7.6E+02	1.1E+03
	TYQGSYGFR	2.3E-04	6.8E-19	5.6E-15	2.5E+03	6.4E+03	6.5E+03
CDK1	IGEGTYGVVYK	3.2E-04	2.4E-12	4.2E-29	5.6E+02	2.3E+03	5.4E+03
	LESEEEGVPSTAIR	8.6E-42	2.6E-25	8.6E-84	5.4E+03	1.3E+04	3.6E+04
	MLIYDPAK	N/D	N/D	7.0E-03	N/D	N/D	8.2E+03
	NLDENGLDLSK	3.7E-04	9.1E-19	3.8E-57	5.6E+02	1.8E+03	1.3E+04
	SPEVLLGSAR	N/D	6.3E-05	9.8E-72	N/D	3.0E+04	4.1E+04
ITGB1	GEVFNELVGK	2.6E-15	9.3E-29	1.1E-56	2.2E+03	1.1E+04	2.1E+04
	LKPEDITQIQPQLVLR	N/D	1.6E-13	1.1E-42	N/D	8.2E+02	1.5E+04
	LSEGVTSYK	3.5E-03	1.7E-19	2.0E-46	6.9E+02	5.6E+03	1.3E+04
	NVLSLTNK	1.8E-03	7.0E-16	3.3E-03	4.8E+02	5.8E+03	7.8E+03
MAP2K1	DVKPSNILVNSR	N/D	3.8E-04	4.2E-27	N/D	2.1E+03	5.8E+03
	ISELGAGNGGVVFK	N/D	5.7E-11	1.0E-58	N/D	6.6E+03	1.1E+04
	KLEELELDEQQR	1.5E-03	1.6E-17	1.9E-39	4.5E+02	6.4E+03	8.2E+03
	LEAFLTQK	N/D	3.7E-04	6.8E-29	N/D	2.4E+03	2.8E+03
	RLEAFLTQK	N/D	5.9E-05	N/D	N/D	1.5E+03	N/D
EGFR	EISDGDVHISGNK	N/D	6.2E-03	1.3E-43	N/D	7.8E+02	6.2E+03
	IPLLENLQIIR	N/D	1.5E-07	3.2E-33	N/D	2.9E+03	1.4E+04
	VLGSGAFGTVYK	N/D	3.4E-04	2.3E-16	N/D	9.7E+02	3.2E+03
	YLVIQGDER	N/D	9.3E-09	1.8E-13	N/D	4.9E+03	8.1E+03
	RPAGSVQNPVYHNQPLNPAPSR	N/D	N/D	2.7E-05	N/D	N/D	3.9E+02
MAPK1	ALDLLDK	N/D	N/D	9.6E-03	N/D	N/D	4.5E+02
	ELIFEETAR	N/D	2.8E-09	3.6E-30	N/D	9.1E+03	2.0E+04
	GQVFDVGPR	N/D	4.2E-05	3.2E-07	N/D	4.0E+03	7.9E+03
	NYLLSLPHK	N/D	N/D	7.5E-03	N/D	N/D	2.8E+03

Note: N/D means "Not Detected"

In response to the comments, we have revised the text (Page9, Line 313-317): *“Notably, the identification and quantification analysis of the dataset were performed and obtained at a statistically stringent criteria of 1% false-discovery rate (FDR) at precursors and protein levels. The confidence of protein identification for representative proteins relevant to lung cancer is depicted and the MS/MS spectra for the identification were shown (Supplementary Fig. 16 and Supplementary Table 3)”*.

We have also added the **Response Figure 9** (as **Supplementary Figure 16**) and **Response Table 3** (as **Supplementary Table 3**) in the revised manuscript accordingly.

Other comments:

4. Although with regard to text and figures the emphasis of the manuscript is on performance of the chip, the real innovation is in the chip itself (and not in e.g. advantages of DIA, which has been shown by others before). In that sense, more details could have been provided on the design and metrics of the chip (e.g. volumes) especially in comparison to alternative approaches (see comment below). Also the authors do not describe how they envision further development of the chip, e.g. to increase throughput or speed). Furthermore, a main concern relates to the claim that in single-cell analyses proteins can be identified that typically are of low abundance and therefore difficult to detect, raising some doubts about the confidence of protein identification.

Reply: 4.1 We thank the reviewer for the comment, suggestion and concern. Our manuscript had described the design and dimension of iProChip in the supplementary information and method section. Following this suggestion, we have updated **Supplementary Figure 1b** to directly show the (x,y and z-) dimensions of the chip and added description on the main text (please see revised text below and in main text). The 10, 50 and 100 cell chambers have volumes of 5, 8.5 and 11 nL, respectively, and volume for the reaction vessel is ~312 nL, which is comparable to other single cell proteomics workflows such as nanoPOTS that exhibited ~200 nL volume. Compared to droplet-based nanoPOTS, though, enclosed chambers of iProChip can more effectively minimize reagent evaporation to maintain stable reaction condition, and its reaction volume can be easily modified to accommodate smaller volume if needed. We had described our perspective on further chip developments in the Discussion section of the manuscript. One future immediate development includes the direct coupling of the chip to the LC-MS/MS that can further reduce sample loss and increase sensitivity. Other developments include the chip re-design to achieve optimal volume, combining proteomics methods (e.g. isobaric labeling methods) and interfacing MS instruments (ultra-low-flow nanoLC and FAIMS), which are anticipated to increase assay sensitivity and throughput.

Supplementary Figure 1b. A single operational unit showing the dimensions for main sections in the iProChip.

4.2 The protein identification was performed at 1% false-discovery rate (FDR) at precursors and protein levels that are the commonly adopted gold standard to ensure stringent protein identification. For protein quantitation, peak area of peptide fragments was used to calculate peptide abundance

and then protein abundance was obtained by using the average value of top 3 peptides per protein. Please also refer to the Comment 2.3.3 that we have listed example DIA spectra from the selected proteins relevant to lung cancer and added them in the Supporting Information (Supplementary Figure 16, and Supplementary Table 3).

In response to the comments, we have (1) updated **Supplementary Figure 1b**, (2) revised the text (Page 4, Line 107-114): *“The cell trap is made up of arrays of 10, 50 and 100 wedge-shaped twin pillars spaced by 5 μm for rapid size-based cell capture in a 5, 8.5 and 11 nL chamber, respectively. A circular chamber with a radius of 1 mm and height of 100 μm (312 nL) was fabricated to ..., A 2.5 cm long desalting column with a cross section of 200 μm x 25 μm”* and (3) added the text (Page 9, Line 313-317): *“Notably, the identification and quantification analysis of the dataset were performed and obtained at a statistically stringent criteria of 1% false-discovery rate (FDR) at precursors and protein levels. The confidence of protein identification for representative proteins relevant to lung cancer is depicted and the MS/MS spectra for the identification were shown (Supplementary Fig. 16 and Supplementary Table 3)”* in the revised manuscript.

5. The design and construction of the iProChip is novel, however some critical aspects are missing to give insight into its characteristics, also compared to alternative platforms. For instance, the dimensions of the chip are provided, but not the volumes of the subsequent ‘chambers’. How does the volume of the chamber for cell trapping/lysis compare to the droplet size in nanoPOTS? And how to surface areas compare, potentially causing adsorptive protein losses? Can the SPE column be re-used or does it need to be re-packed for every sample? In addition, the design of the ‘pillars’ to trap defined numbers of cells is not described in detail. Was this based on trapping structures described before (citation missing), or were they newly designed for this purpose? In the latter case it will be interesting to hear how size, position and shape of the pillars were optimized to achieve efficient trapping.

Reply: We thank the reviewer for the comment and appreciate the novelty of the iProChip.

5.1 Following reviewer’s suggestion, we have updated the chip illustration in Supplementary Figure 1b to provide more detail dimensions, and have calculated the volumes and surface area of chambers in the iProChip and compared to nanoPOTS (**Response Table 4**). The reaction vessel has a volume of 312 nL, which is comparable to ~200 nL droplet size in nanoPOTS. Meanwhile, the surface areas for the 10, 50 and 100 cell trapping chambers and reaction vessels are 0.46, 0.81, 1.1 and 6.4 mm^2 respectively, whereas the surface area for the nanoPOTS chip is ~0.8 mm^2 (*Nat Commun* 9, 882 (2018)). Although the iProChip exhibits a higher surface-to-volume ratio than nanoPOTS, compared to a typical sample preparation volume in a 0.5mL vial (~130 mm^2), it exhibits substantially reduced surface area by 95 %, which effectively reduces adsorptive losses. Furthermore, our proteomics analysis at 1-100 cells level further demonstrated that BSA coating of the chip also effectively minimizes adsorptive loss for enhanced sensitivity in proteomic profiling.

5.2 In the present study, our major effort was to develop an integrated platform for proteomics analysis at the capacity of 1-100 cells, particularly focusing on high sensitivity and streamlined sample preparation. Therefore, at the first stage of method development, we do not intend (and suggest) to reuse the SPE column which requires detailed evaluation on the complete cleaning up of the column and sample recovery, which will be part of the future work to establish an economical chip. The cell capturing pillars were designed to have a wedge-shape in the front with a vent distance of 5 μm , so it can capture cells with diameter >5 μm , which is suited to study most mammalian cells. These types of narrow body traps are more convenient to arrange in a confined space, especially when multiple

cell captures in the same chamber are needed. The position (arrangement) of the traps within the chamber was adopted from literature to achieve optimal trapping efficiency (*Lab Chip* 6, 1445, 2006).

Response Table 4. Surface area and volume for iProChip and nanoPOT.

	iProChip				nanoPOTS
	Cell capturing chamber			Digestion chamber	
	10	50	100		
Volume (nL)	5	8.5	11	312	200
Surface area (mm ²)	0.46	0.81	1.1	6.4	0.8
Surface-to-Volume ratio	0.092	0.095	0.1	0.02	0.004

In response to this comment, we have added (1) discussion in the revised manuscript (Page 4, Line 111-113): “Compared to a typical sample preparation volume in a 0.5mL vial (~130 mm²), iProChip exhibits substantially reduced surface area by 95 %, which effectively reduces adsorptive losses”, (2) updated the text (Page 4, Line 107-114): “The cell trap is made up of arrays of 10, 50 and 100 wedge-shaped twin pillars spaced by 5 μm for rapid size-based cell capture in a 5, 8.5 and 11 nL chamber, respectively. A circular chamber with a radius of 1 mm and height of 100 μm (312 nL) was fabricated to ..., A 2.5 cm long desalting column with a cross section of 200 μm x 25 μm....”, (3) added the **Response Table 4** (as **Supplementary Table 1**) to list surface area and volume for iProChip, as well as (4) updated **Supplementary Fig. 1b**.

6. Interfacing of the chip to LC-MS seems cumbersome: drying of peptides and resuspension in 5ul seems counterproductive, where all efforts have been geared towards avoiding losses and keeping volumes low. Is there any other way that peptides can be collected from the chip without drying and dilution?

Reply: We thank the reviewer for the comment to simplify our workflow. The step of sample drying and dilution with 0.1% FA is for buffer exchange purpose, which is a prerequisite step for subsequent MS analysis. Recently, some studies demonstrated direct coupling with LC-MS/MS by using custom-made autosampler and automatic switching (*Anal. Chem.* 93, 1658–1666, 2021; *Anal. Chem.* 92, 10588–10596, 2020; *Anal. Chem.* 92, 2665–2671, 2020). In fact, direct coupling of the chip to LC-MS/MS is also our upcoming effort along the way. Briefly, direct transfer to autosampler of LC-MS can be implemented by an external two-position six-port switch valve to switch between the reagent vial with loading buffer (0.1% FA) and a PEEK tubing connected to individual processing units on the chip. User-defined program can be designed to control the valve switch and sample injection into the sample loop for subsequent LC-MS analysis. The new single cell chip (ongoing) has been designed to include only ~1 cm long SPE columns, which will be suitable to apply descent pressure for sample transfer through PEEK tubing connected to individual processing units on the chip. The PEEK tubing will in turn be connected to customizable commercially available LC autosampler through tubing connectors to the LC-MS/MS.

7. The authors do not describe disadvantages of the iProChip, nor do they mention how its performance can be improved. For instance, processing 8 samples (cells) in parallel is nice but in the end is a low number, and the need for mixing on a plate shaker is inconvenient. In addition, the chip is not specifically designed to capture a single cell (see point 1) – is this foreseen to be implemented?

Reply: We thank the reviewer for the comment.

7.1 In the Discussion section, we had discussed the limitations and possible directions we foresee that can elevate the performance of iProChip specifically for single cell proteomics, both in terms of improved sensitivity and higher throughput (please also see our reply to Reviewer 2, comment 4). Like other newly developed miniaturized platforms, a trained scientist is needed to fabricate and operate the iProchip. We also thank the reviewer's comment that mixing on a plate shaker may be inconvenient. In our first-generation chip, we did not test the possibility of performing the 16 h digestion step without a shaker. However, as shown in our mixing test (**Manuscript Figure 2d**), mixing of food dyes by passive diffusion can be completed in ~0.5 hr. Therefore, it may be feasible to operate the iProChip-DIA protocol without the use of a shaker in our upcoming efforts. Another disadvantage for the iProchip could be its relatively low throughput although this proof of concept study did not aim to high throughput assay. It is expected to further improve the throughput of iProChip by accommodating more single cell chambers in the chip (see point 7.2), and by incorporating isobaric labeling methods such as tandem mass tags (TMT), through which multiplexed single cell profiling can be realized.

7.2 We recently developed a new chip dedicated for single cell proteomics and demonstrated it can realize faster and easier **single cell capture** (**Response Figure 7**, unpublished, also see Reply 1.2 to Reviewer 2). In this new chip, each chamber contains only 1 trap so that a single cell can be precisely captured in a shorter time, which will further facilitate full automation of overall workflow and increase the cell usage efficiency.

8. The workflow implemented on the chip rather strictly follows the 'classical' procedure for sample preparation that is broadly used for bulk samples, including lysis with detergent/Rapigest, protein reduction/alkylation, digestion, and peptide cleanup. Miniaturizing these steps on a chip is an achievement in itself (especially the clean-up on columns that are packed on-chip), however it is questionable if all steps are required. Specifically, there is a clear trend to simplify the procedure since requirements for processing single cells and bulk samples are different. For instance, lysis of single cells can be done by a freeze-heat cycle in regular buffer even omitting reduction/alkylation (e.g. <https://doi.org/10.1101/399774>; PMID 33504367; <https://doi.org/10.1101/2020.12.22.423933>) resulting in minimal contamination avoiding the need for any clean-up. It will be interesting if the authors can comment how the iProChip can be adjusted to accommodate simplified workflows.

Reply: We thank the reviewer for valuable suggestions and insights. We are intrigued by the reviewer's view, and agree that the ability to simplify proteomics procedures, such as through the ProteOmic sample Preparation (mPOP) (H. Specht *et. al.*, bioRxiv (2019); <https://doi.org/10.1101/399774>) that uses freeze-heat cycle to extract proteins in pure water, and thus obviating cleanup before MS analysis, may further facilitate enhancement of analysis sensitivity. In the present study, our major effort was devoted into developing a highly integrated, miniaturized device for carrying out the entire proteomics workflow in a streamlined fashion. Therefore, at the first stage of development, iProChip is designed to follow a rather classical workflow.

Nonetheless, since iProChip design is highly versatile and can be organized on a module-by-module basis, other simplified workflows, such as mPOP mentioned above, can be easily integrated into the iProChip. Inspired by the promising result of iProChip, in the next phase we aim to further simplify current workflow and interface to LC-MS/MS, in conjugation with additional microfluidics features and proteomics methods, to realize more sensitive proteomics analysis with added cell assay functionality. With the demonstrated robustness and reproducibility through iProChip operation, any incorporated simplified workflow can be efficiently verified and further prototyped into a next generation chip.

9. Although it is not the intention of this manuscript to compare iProChip to nanoPOTS, it needs to be seen if overall performance of iProChip exceeds that of nanoPOTS, which is also determined to a large extent by the LC-MS system at the back end. Nevertheless, in this light authors need to be careful in their statement with regard to ‘ultra-streamlined’ and ‘ultra-sensitivity’, for 2 reasons: i) the level of streamlining and sensitivity is similar in both approaches, with regard to the number and type of handling steps that are included in the procedure, and the number of proteins that can be identified; ii) on an absolute scale, and at least in theory, streamlining and sensitivity can be much improved, thereby going beyond the ‘ultra’ classification – for which it will then be hard to find an appropriate term. In addition, throughput of the presented system is limited when compared to array-based platforms (8 vs 96 samples in parallel).

Reply: We thank the reviewer for the comment and suggestion to improve our statement. As pointed out by the reviewer, a direct comparison between iProChip and nanoPOTS in terms of overall performance can be influenced by many factors, such as the utilized LC-MS/MS system, DIA approach and others. Therefore, direct comparison on the numbers of proteins may be somewhat misleading, which is not the objective of this study. We highly appreciate many features by the pioneering approach nanoPOTS, such as the high throughput, facile adoption by liquid handling system and excellent analytical merits. More generally, the aim in this study is to introduce iProChip, microfluidics-based alternative, for the all-in-one design principle to accommodate the entire sample processing workflow with minimal intervention. Compared to nanoPOTS, iProChip can realize from cell trapping and quantification to the proteomics processing without the need of external instruments, hence we use the word “streamlined”. Although unexplored in the present study, cells can be further manipulated biochemically through microfluidics operations, e.g. by cell sorting on a surface functionalized trapping chamber, flowing in ligands to stimulate or inhibit cells. Such flexibility in additional cell assay capacity is currently lacking in existing single cell proteomics approaches.

We agree with the reviewer’s suggestion about the wording on “ultra-streamlined” and “ultra-high sensitivity” is valid, since both approaches achieve a comparable level of performance by using similar type and handling protocols. Therefore, we have revised both words throughout the manuscript.

In response to this comment, we have revised the title and text containing the word “ultra-streamlined” to “*highly streamlined*”, as well as revised “ultra-high sensitivity” to “*high sensitivity*” throughout the manuscript.

10. Line 115-120: discrete numbers of cells can be trapped from a cell suspension; however, the efficiency of trapping is unclear. That is, what is the absolute number of cells that need to be input to capture 10/100/1000 cells?

Reply: We thank the reviewer for the question. As mentioned in our response to your second comment, we have performed additional experiments to determine (1) absolute numbers of cell needed, and (2) cell usage efficiency for the iProChip operation. The result showed that ~125, ~150 and ~225 cells will be sufficient for iProChip to capture 10, 50 and 100 cells respectively. The measured cell usage efficiency (numbers of trapped cells/numbers of total injected cells) is ~8%, ~33% and ~44% for 10, 50 and 100 cell chambers respectively. (Please also see our reply to reviewer 2, 2nd comment for more detail and associated text revision in the revised manuscript).

11. Line 132: mixing is rather slow, and also other steps in the procedure do not seem particularly fast. It will be helpful if the authors can include a scheme indicating the time for each handling step,

including overhead time, as well as net time for processing 1 sample (or 8 in parallel) from cell trapping to peptide collection.

Reply: We thank the reviewer for the suggestion, and agree that a scheme with clearly indicated times for each operational step will be very helpful. Previously, we included handling times for individual steps in Supplementary Figure 2.

Following the reviewer's suggestion, we have updated both Figure 1f and Supplementary Figure 2 to clearly indicate the time for each step, and overall processing time in the method section (Page 17, Line 577): *"The overall time for parallel processing of 9 samples was ~20 hours"*.

12. It is surprising that digestion is performed for 16h, which is typically shorter even for bulk samples. I would expect that digestion time can be reduced benefiting from low volumes and high enzyme/substrate ratio. Did the authors look into this?

Reply: We thank the reviewer for providing the insight and suggestion. Indeed, the digestion efficiency is expected to be higher for the enzymatic reaction in such miniaturized volumes. However, in this study, we had not tested the digestion time thoroughly and rather adopted the routine practice in our laboratory. We nevertheless thank the reviewer for the suggestion, and indeed plan to further optimize iProChip operation including the digestion time.

13. Line 508: LC gradients used for separation of peptides obtained from single cells is extremely long. All in all this amounts to ~3h per cell, i.e. allowing to do 8 cells per day. It is very unlikely that this cannot be shortened, and it is very unclear why the authors chose this type of gradient.

Reply: We thank the reviewer for pointing this out. The dataset for construction spectra library was generated with a long LC gradient. Although we spiked iRT peptides, a similar LC gradient was used for the DIA analysis for better retention time alignment with the spectral library. For DIA acquisition we employed a narrow 10 Da m/z isolation window (40 cycles) for high precursor selection specificity and chose a cycle time of ~3 seconds. This allows obtaining enough data points per peak for more accurate peak area integration. For the next step of optimization to benefit for future studies, we will take the suggestion to further decrease gradient time along with optimization of DIA acquisition parameters.

14. Line 312: the comparison of the data to those by Rieckmann et al does not seem very relevant, since they analyzed immune cells, not leukemic/cancer cells, hence this is not representative for the MEC-1 cell line.

Reply: We thank the reviewer for the comment, as mentioned earlier in Reviewer's 2 comment 3.1, we have further compared the proteome coverage of our data with the B-cell leukemia cells from Johnston et. al., which is more similar to the MEC-1 cell. As shown below (**Response Table 2**), even though MEC-1 cell line also belongs to human B-cells leukemia, only 51% of protein groups commonly overlapped between the two datasets. Nevertheless, the important B-cell markers (e.g. CD19, CD20 and CD22) were readily and commonly detected in both Johnston et al., and our MEC-1 data, while they were not detected in our PC-9 data. Similarly, the important lung cancer markers (e.g., EGFR and TP53) were not detected in MEC-1 and Johnston et al., study (as shown in the table below). This again supports the argument of cell-dependant specificity. On the other hand, EGFR and TP53 are both the critical proteins for lung cancer and have high expression in the lung cancer cell line. Therefore, we were able to detect EGFR and TP53 in the PC-9 lung cancer cells at as low as ~5 cells.

Response Figure 8. Venn diagram showing overlap of commonly identified proteins among Johnston et al., study and our MEC-1 data.

Response Table 2 | Comparison of the identification of B-cell surface markers and important NSCLC proteins among datasets from PC-9, MEC-1 and Johnston et. al, study.

Important Protein Comparison (Our Study Vs. Johnston et al.,)

Cells	Key Proteins	Accession Number	Our study (PC9)	Our study (MEC1)	Johnston et al.,
PC9	EGFR	P00533	✓	X	X
	TP53	P04637	✓	X	X
	MAP2K1	Q02750	✓	✓	✓
	MAPK1	P28482	✓	✓	✓
	CDK1	P06493	✓	✓	✓
	ITGB1	P05556	✓	X	✓
	PGK1	P00558	✓	✓	✓
	CDK4	P11802	✓	✓	X
MEC1	CD19	P15391	X	✓	✓
	CD20	P11836	X	✓	✓
	CD21	P20023	X	✓	✓
	CD22	P20273	X	✓	✓
	CD81	P60033	✓	✓	X
	CD47	Q08722	✓	✓	✓
	CD74	P04233	X	✓	✓
	HLA-DRA	P01903	X	✓	✓
	HLA-DRB5	Q30154	X	✓	✓

We have added the **Response Table 2** (as **Supplementary Table 4**) and following text in the revised manuscript (Page 11, Line 364-372): *“According to the Human Protein Atlas (HPA) database (<https://www.proteinatlas.org/>), the protein expression levels are dependent on the cell type. We have further compared the proteome coverage of our data with the B-cell leukemia cells from Johnston et. al. Even the MEC-1 cell line also belongs to human B-cells leukemia, only 51% of protein groups commonly overlapped between the two datasets (Supplementary Table 4). Nevertheless, the important B-cell markers (e.g., CD19, CD20 and CD22) were readily and commonly detected in both datasets, while they were not detected in our PC-9 data. Similarly, the important lung cancer markers (e.g., EGFR and TP53) were not detected in MEC-1 and leukemia cells (Supplementary Table 4). This supports the notion of cell-dependent specificity of protein expression”.*

15. Line 558-560: this sentence needs some clarification: does this mean that the authors require at least 3 peptides to be quantified per protein? And what does 'peak area at MS2 level' mean exactly? Is it the cumulative area of all detected fragments for each peptide? And what does 'minor peptide grouping' imply?

Reply: We thank the reviewer for the comment. From quantification of DIA dataset, peptide precursor quantification values are initially computed by summing the integrated peak area of top 6 intense interference free fragment ions (called MS2 level). Then average peak areas of peptides of different charge states or modification are obtained (called minor grouping) based on unique peptide sequence. To infer the abundance of a protein, the measured intensities are aggregated by averaging the 3-most intense peptides per protein. Here, we followed a standard targeted DIA data analysis approach using Spectronaut and further information can be obtained from Ludwig et. al. (*Mol Syst Biol* **14**, e8126, 2018).

Reviewer #3 (Remarks to the Author):

The authors present an engineering marvel in the form of a chip based on multilayer soft lithography to prepare samples comprising one or more cells for proteome analysis. If this were submitted to e.g. Analytical Chemistry or Lab on a Chip I would be enthusiastic about its publications based on the engineering and novelty. However, to meet the bar for publication in Nature Communications, the work should also provide a meaningful advance over the state of the art, which this does not. The approach has numerous and severe downsides, which impede practical implementation beyond this proof of concept demonstration and which the authors neglect to mention. Suggestions for improvement of the manuscript and a more comprehensive discussion of both the benefits and limitations of the approach are provided below:

Reply: We thank the reviewer for the comment on both the benefits and limitations of our approach. We would like to further summarize the following points to clarify the chip details and its advancement.

1. **Microfluidics hold great potential as a all-in-one proteomics station**

Developing microfluidic chips has enabled many advancements in the biological and biomedical research, such as its implementation enabling the study of interactome using ultra-low input lysate cells (*Nat Commun* **10**, 1525, 2019), realizing ultra-sensitive detection of proteins and RNA in single cells (*Nat. Commun* **10**, 3544, 2019) and uncovering logical rule in stem cell differentiation (*Sci Adv* **5**, eaav7959, 2019). From these literatures, it is the feature of custom design and construction in microfluidic devices that facilitates a wide range of implementations. However, its utility in the microproteomics is limited. In fact, even Dr. **Ruedi Aebersold** from ETH, a world-leading proteomics expert, shared his perspective on the choice of a microfabricated PDMS device for single cell proteomics profiling in a recent article entitled "***A dream of single-cell proteomics***" (*Nature Methods*, **16**, 809, 2019). Dr. Aebersold commented on the microfabricated device: "*Cells flow into the device with one cell per compartment, which is confirmed with imaging. Cells can be manipulated and lysed, proteins can be washed, and the sample can then be worked up for mass spec.*" These established works and leading opinions suggest great potential of microfluidics devices to serve as an all-in-one station for single cell proteomics profiling, although it was not demonstrated until the introduction of this study.

2. **Positive evaluation and acknowledgement by Reviewer #1 and #2.**

Reviewer #1: “Overall, I feel that the novelty of this manuscript is high, but much more rigour in terms of experimental design ...; The device itself quite novel - it quite nicely accomplishes cell capture, cell lysis, peptide digestion and desalting””

Reviewer #2: “The design and construction of the iProChip is novel, ...; Miniaturizing these steps on a chip is an achievement in itself ...; This is an interesting manuscript, providing an alternative workflow... ; This is a welcome development to show that alternative approaches are conceivable that may have advantages....”

The comments from Reviewer #1 and #2 were positive and indeed highly supportive to welcome the development of alternative single cell proteomics workflows that can provide features to complement existing ones. We would also like to clarify that **our work was inspired and based on the knowledge from all pioneering single cell proteomics studies**. We do NOT neglect previously reported efforts.

3. Fundamental advancements and future development of iProChip

As the very first microfluidics-based single cell proteomics processor, we have (1) provided all details in chip design and validation, (2) fully characterized its analytical performance, and (3) systematically demonstrated its all-in-one station capability to achieve highly sensitive proteomics profiling at 1-100 cells levels without external cell sorting and other instrument. At its current state, it shows proteome depth comparable to existing methods, 5 orders of proteome coverage, highly consistent ~100-fold protein quantification (1-100 cells) and high reproducibility with low missing values (<16%) in both adherent and non-adherent cell types. Like other newly developed platforms, the iProChip has limitations in demand of a trained scientist for device fabrication, rather low throughput although this proof of concept study did not aim to increase the throughput. Future improvement can be achieved, e.g. direct interfacing of iProChip to the LC-MS/MS (ultra-low-flow nanoLC and FAIMS) to increase sensitivity, chip re-design to achieve optimal volume and higher throughput. It is the good performance and promising potential of iProChip that we hope to share with the single cell proteomics community. We appreciate the opportunity to clarify these points.

1. First, these devices are incredibly labor intensive to fabricate, failure prone and single use. Once we take into account the many, many hours spent to cast PDMS, align and bond the layers, punch holes for valves and reservoirs, clean out the holes to eliminate debris, manually make the 34 connections required for fluidic control, pack SPE media and passivate the surface of the chip for 1 h with BSA prior to use, all just to prepare 9 samples, it is abundantly clear that there is nothing “ultra-streamlined” about this platform and that rather this is likely the most difficult way to prepare proteomic samples that has ever been developed. In contrast, other microproteomics approaches based on micro and nanowell plates are arbitrarily scalable to dozens or hundreds of samples with either reusable or disposable devices. As such, the minimal manual intervention required during processing in competing approaches is orders of magnitude less than the up front cost required by this approach before initiating the preparation.

Reply: We thank the reviewer for the comment and would like to address the comments as follows:

1 Concerning microfluidics devices for single cell proteomics The comment regarding devices of this kind are “*incredibly labor intensive to fabricate, failure prone and single use*” may need clarification. For instance, one of the most widely used and enabling single cell sequencing platforms, C1 system by Fluidigm, was conceptualized and developed based upon layered microfluidics devices (*Nat Biotechnol.* 32, 1053, 2014); <https://www.fluidigm.com/products/c1-system>). Since its introduction, its utility has greatly enabled numerous highly important discoveries in the single cell community,

such as single nucleus RNA sequencing (*Science* **352**, 1586, 2016), noise level characterization in expression patterns (*Nat Commun* **6**, 8687, 2015) and study of mouse embryo transcriptome dynamics (*Nature* **583**, 760, 2020).

In a recent technology feature article “**A dream of single-cell proteomics**” (*Nature Methods*, **16**, 809, 2019), in fact, even Dr. **Ruedi Aebersold** from ETH, a world-leading proteomics expert, shared his perspective on the choice of a microfabricated PDMS device for single cell proteomics profiling: “The goal is to eliminate sample handling losses”, he says, His team hunted for the least absorbent material, chose polydimethylsiloxane (PDMS) and built microfabricated devices that “work quite well,” he says, though the team is still testing the device. Aebersold also commented on the microfabricated device “*Cells flow into the device with one cell per compartment, which is confirmed with imaging. Cells can be manipulated and lysed, proteins can be washed, and the sample can then be worked up for mass spec.*”

2 Rationale to develop alternatives for single cell proteomics We fully agree with the Reviewer that the micro- and nanowells-based approach offer good merit of easy scale up. Nevertheless, single cell proteomics is an emerging topic, and development of other alternative strategies is essential to implementation for various applications, particularly when the new strategy shows promising performance. Furthermore, developing microfluidic chips for various biological applications is a highly active research field, and many advanced devices are being developed to address unmet needs in various settings, such as its implementation enables the study of interaction proteomics using ultra-low input lysate (*Nat Commun* **10**, 1525, 2019), single immune cell secretion (*Cell Reports* **15**, 411, 2016), uncovering the logical rule in stem cell differentiation (*Sci Adv* **5**, eaav7959, 2019), realizing ultra-sensitive proteins and RNA in single cells (*Nat. Commun* **10**, 3544, 2019) and etc (*Nature* **507**, 181, 2014; *Cell* **160**, 381, 2015; *Nat Commun* **9**, 212, 2018; *Nat Commun* **11**, 5271, 2020). From the above-mentioned literature, it is the feature of custom design and construction in microfluidic devices that facilitates the wide range of implementations. Our microfluidics-based iProChip is conceptually different from existing single cell proteomic methods and exhibits complementary features to existing methods (e.g. no need to use external cell sorter which may not be accessible to some users). Moreover, the design of microfluidics chips is highly versatile and can be easily prototyped; additional microfluidics features such as introducing advanced cell manipulation and sorting before proteomics workflow can be easily performed. In summary, these established results inspired us to eagerly introduce the iProChip to the single cell proteomics community.

3 Chip engineering issues We would also like to clarify that “streamlined” as stated in our approach refers to the fact that iProChip can accommodate the entire proteomics workflow, starting from initial cell trapping to final peptide desalting, in a fully automation manner without the use of additional instruments, which can serve as alternative to complement existing approaches. As mentioned in the manuscript, operation of iProChip is facilitated by a custom controller and GUI, so that it can execute the entire proteomic workflow in an automatic fashion. **The whole iProChip fabrication takes real on-bench time of ~4 hours** and it has the potential of re-use after careful evaluation on the sample recovery. We appreciate that the Reviewer reminded the failure-prone issue of the chip fabrication; performance validation on a chip-by-chip basis is a basic criteria for microfluidics device development. Factors such as overall chip design, channel/chamber dimension for optimal sample processing, feature density for reaction compatibility and cleanliness to ensure absorption/contamination-free process during chip fabrication should all have to be considered and fine-tuned to obtain optimal devices. As shown in our response to the reviewer 1 (comment 4),

protein identification of 10 cells obtained from two different chips showed reproducible results (Response Figure 6, also shown below).

Response Figure 6. Comparison of identification coverage and performance of proteomic profiling of additional 10-cells data (PC-9). (a) comparison of identification of protein groups and peptides in additional 10-cell data with data in the submitted manuscript. (b) The Venn diagram classification of proteins and peptides among triplicates in both experiments (c) Comparison of total protein groups among both experiments.

In summary, given the importance of single cell proteomics, we do feel the effort put into developing a new approach that provides additional advantage over current methods is worthwhile. Standing on the very fundamental methodological point of view, what is behind the rationale of the submitted manuscript is to introduce a new platform with analytical capability of single cell proteomics as well as extendable functionality in cell manipulation and assay. The device design and fabrication and analytical merits were fully explored and discussed in the report. **Our work is an alternative that is complementary to existing tools, NOT neglecting previously reported efforts.**

2. A primary rationale for miniaturizing sample preparation for proteomics is to minimize surface exposure. Unfortunately, enclosed microfluidic devices have a large surface-to-volume ratio that results in high surface adsorption from small volumes. The authors should calculate and compare the surface area that they expose their sample to, both in the trapping/lysis chamber, the subsequent digestion chamber, and the collection reservoir to the ~1 mm² surface exposure of other small-sample proteomics studies.

Reply: We thank the reviewer for the suggestion, and agree that reducing surface exposure can minimize adsorptive loss and enhance analytical sensitivity. Therefore, iProChip is designed to be a miniaturized, integrated station capable of performing the entire proteomics workflow for single cell profiling. Following the suggestion, we have calculated the volume and surface area of the chambers in the iProChip and compared to nanoPOTS (Response table 4). The iProChip reaction vessel has a volume of 312 nL, which is slightly larger yet comparable to ~200 nL droplet size in nanoPOTS. Meanwhile, the surface areas for the 10, 50 and 100 cell trapping chambers and reaction vessel are 0.46, 0.81, 1.1 and 6.4 mm² respectively, whereas the surface area for the nanowell in the nanoPOTS chip is ~0.8 mm² (Nat Commun 9, 882, 2018). Although the iProChip exhibits a higher surface-to-

volume ratio than nanoPOTS, compared to a typical sample preparation volume in a 0.5mL vial (~130 mm²), it exhibits substantially reduced surface area by ~95 %, which effectively reduces adsorptive losses. Furthermore, our proteomics analysis at 1-100 cells level further demonstrated that BSA coating of the chip also effectively minimizes adsorptive loss for enhanced sensitivity in proteomic profiling.

Response Table 4. Surface area and volume for iProChip and nanoPOT.

	iProChip				nanoPOTS
	Cell capturing chamber			Digestion chamber	
	10	50	100		
Volume (nL)	5	8.5	11	312	200
Surface area (mm ²)	0.46	0.81	1.1	6.4	0.8
Surface-to-Volume ratio	0.092	0.095	0.1	0.02	0.004

In response to this comment, we have added discussion in the revised manuscript (Page 4, Line 111-113): “Compared to a typical sample preparation volume in a 0.5mL vial (~130 mm²), iProChip exhibits substantially reduced surface area by 95 %, which effectively reduces adsorptive losses”, and added a new **Supplementary Table 1** to list surface area and volume for iProChip.

3. Another challenge with an enclosed microfluidic device and one utilizing sized-based cell trapping is the limitation regarding what samples can be analyzed. For example, the platform is incompatible with laser microdissection and FACS sorting, which are both widely utilized for cell isolation. The cell trap geometry may need to be modified for each different cell size, and a single small cluster of undissociated cells can foul the channel and render that portion of the device useless. An open well format is readily interfaced with common cell isolation techniques and can easily analyze single cells from tissue sections in addition to dissociated cells. Further, trying to analyze single cells based on trapping chambers of 10 traps or more requires the researcher to “get lucky” to trap a single cell. Since the title indicates that single cell proteomics is the primary objective for the work, I wonder how many chips would have to be fabricated and tested to analyze 100 cells. I also wonder how the chip would perform with primary cells that may not behave as nicely as the cultured cells used here. The authors should be up front about these limitations, which do not exist for open well plate based methodologies.

Reply: Utility of Microfluidics in Biomedical Research We thank the reviewer for the comment on the considerations for applications. Enclosed microfluidics chips have been widely developed as miniaturized total analysis systems (μTASs) to streamline complex assay protocols (*Nature* **507**, 181, 2014) to study various biological or biomedical systems (*Nat Commun* **9**, 212, 2018; *Sci Adv* **5**, eaav7959, 2019; *Nat. Commun* **10**, 3544, 2019; *Nat Commun* **10**, 1525, 2019; *Nat Commun* **11**, 5271, 2020; *Cell* **160**, 381, 2015; *Cell Reports* **15**, 411, 2016). Based on these well-documented features of μTASs, therefore, we aim to extend the microfluidics chips as a miniaturized proteomic processor for single cell proteomics application. Designed to be an integrated platform, a main distinguishing feature of iProChip is its ability to capture cells without an external sorter. Therefore, interfacing with a laser microdissection or FACS sorter is not needed for iProChip workflow. In addition, creating inlet ports with a well-defined dimension on PDMS chips is common practice during fabrication, such ports can be used to accommodate additional instruments when necessary. Meanwhile, since iProChip is the first of this kind of device developed for single cell proteomics profiling, we choose to isolate cells based on their size in the first-generation chip. Alternative cell isolation strategies such as affinity binding can be integrated to realize multiplexed cell sorting directly on the chip, which is a feature not easily implementable using an external cell sorter.

Sample Input and Single Cell Capture In response to the comment, we have performed additional experiments to evaluate the single cell trapping. The results showed that using as low as 5uL cell solution with a density of 25 cells/uL, iProChip can routinely realize single cell capture in a few seconds and achieve good cell usage efficiency of ~4% for single cell operation (**Table R1**; see also **Reviewer 1, comment 1**). Note that the imaging-ready feature of iProChip further enables absolute quantification of exact sample size being analyzed. Therefore, using as low as ~250 cells in total as initial sample quantity, iProChip is capable of parallel processing 9-10 single cell samples for proteomic analysis. For the entire study, we used a total of 5 chips to acquire all data shown in the manuscript. We expect that iProChip throughput can be enhanced by chip re-design to accommodate more single cell chambers, and through incorporating isobaric labeling methods.

iProChip for Primary Cell Characterization We also thank the reviewer's question about how iProChip would perform when analyzing primary cells, which are materials of high clinical relevance. In this study, we have not yet tested primary cells in iProChip. However, layered microfluidics has been developed to study various primary cells, such as the immune cell, neural stem cell, mesenchymal and hematopoietic stem cell (*Sci Adv* **5**, eaav7959, 2019; *ACS Nano* **14**, 15094, 2020; *Dev Cell* **48**, 293, 2019; *PNAS* **115**, E10907, 2018; *Front Immunol* **9**, 2373, 2018). Based on these established protocols and flexibility of iProChip design, it is expected that iProChip can be implemented to analyze primary cells.

In response to the comment, we have added description on the cell capture process in Method section of the revised manuscript (Page 16, Line 539-552): *"In order to characterize minimal numbers of cells needed to operate in this chip, we prepared vials containing PC-9 cells with a cell density of 25 cells/uL, and used a total volume of either 5 or 10 uL (which equals to 125 or 250 cells in each vial). The cell vial was then connected to the chip, and directly flowed into the cell chamber at 3 psi to determine how much sample volume and cells is needed for capturing desired cell numbers (including 1, 5, 10, 50 and 100 cells) in individual chambers. The cell solution was injected into each chamber so that a fraction of cells were captured by wedge-shaped traps while remaining cells simply passed by to the waste-outlets. A real-time chip monitoring by a brightfield microscope was used to inspect the cell flow and capture; the valves were closed when desired numbers of cells were captured in the chambers. Note that the imaging-ready feature of iProChip enables easy quantification of trapped cells. At this cell density, It took 3-40 s to capture 1-100 cells respectively. The result is summarized in Supplementary Table 2 and Supplementary Fig. 4. Our result showed 4-8% cell usage efficiency for capturing 1, 5 and 10 cells and 33-44% for trapping 50 and 100 cells."*

4. The authors need to present proteome coverage from a blank control containing cell supernatant and going through all the same sample preparation steps. Otherwise, we cannot know whether the identified proteins originated from a cell or were inflated by cellular debris and the contents of lysed cells present in the supernatant.

Reply: We thank the reviewer for suggesting this control experiment, which further clarified the validity of our approach. As per reviewer's request, we have performed analysis on a blank control sample using the identical sample preparation steps without trapped cells. The results show that only an average of ~58 proteins were identified. Compared to the average of 88 proteins from similar studies reported recently in single-cell proteomics studies (*Anal Chem* **92**, 2665, 2020; *Nat Commun* **11**, 5632, 2020), our results presented comparatively less proteins from blank sample (**Response Figure 10a**). Further comparison of identified proteins shows a significant overlap between the 1-cell and the 5-cell samples and minimal overlap with 0-cell (blank) samples (**Response Figure 10b, c**). These comparisons suggest low cross contamination and false identifications.

Response Figure 10. Protein groups identification in triplicate analyses of blank (zero cell) sample. (a) The number of proteins identified across triplicates of blank samples. (b) Comparison of protein identification among blank, 1-cell and 5-cells samples. (c) Venn diagram of overlapping protein identified from 0-cell, 1-cell and 5 cells.

In response to this comment, we have (1) added following text: (Page 8, Line 259-266): *“To evaluate the background of the single cell measurement, blank control samples (cell-free supernatant without trapped cell) were prepared using iProChip to go through all sample preparation steps and analyzed. Compared to the average of 88 proteins from similar studies (Anal. Chem. 92, 2665, 2020; Nat Commun 11, 5632, 2020), our results of an average of 58 proteins presented comparatively less proteins from the blank sample (Supplementary Fig. 10). Further comparison showed a significant overlap between the 1 cell and 5 cell samples, both minimally overlapped with the blank sample. These results suggest low cross contamination and false identifications”*, and (2) added the **Response Figure 10** (as **Supplementary Figure 10**) in the revised manuscript accordingly.

5. PDMS is gas permeable. The authors should discuss how they prevented evaporation through the channel walls during extended incubations.

Reply: We thank the reviewer for the comment. Gas permeability of PDMS can be influenced by several factors, such as the type of gas, vapor pressure, environment temperature, PDMS (monomer:catalyst) ratio and membrane thickness (*RSC Adv* 4, 61415, 2014; *J Membr Sci* 481, 1, 2015). In our operation, channel walls of the cell capture/lysis chamber and reaction vessel were thoroughly coated with BSA (0.1 %) before each experiment; such treatment not only effectively minimizes adsorptive loss of peptides but also prevents evaporation of reagents. It is also noted that during the extended incubation at 40C, the vapor pressure of water is ~0.07 atm which is rather low to yield significant evaporation for an enclosed chamber filled with solution. Furthermore, we also attribute the minimal evaporation (if any) observed during our iProChip operation to a rather thick (~6 mm) PDMS membrane, the close-chamber geometry of iProChip, as well as the enclosed compartment during extended incubation inside the shaker.

In response to this comment, we have added following text on Page 6, Line 176-179: *“Note that before each experiment, the cell chamber and reaction vessel were coated with bovine serum albumin (BSA) to minimize adsorptive losses of peptides and evaporation of reagents during extended incubation.”*

6. Reporting total coverage across multiple replicates as the authors do in the abstract and elsewhere in the manuscript is an artificial means of reporting greater coverage than is actually obtained. The reported coverage using this approach is not meaningful and is not the convention of the field except for TMT studies. This is especially critical as the authors are comparing coverage of their platform identified across replicates to average proteome coverage reported in other studies.

Reply: We thank the reviewer for the comment. We would like to clarify that we adapted the reporting style from previous literature. (e.g., please check these as proofs: *Anal Chem* **90**, 5430, 2018; *Anal Chem* **92**, 2997, 2020; *Mol Cell Proteomics* **14**, 1672, 2015). Throughout the manuscript, the proteomic coverage has been reported both in average of triplicate as well as sum of triplicates to present the overall identification, so the identification using both approaches are easily distinguished.

For better clarity, we have revised the Abstract in the revised manuscript (Page 2, Line 28-29): “By mapping to project-specific spectra libraries, the iProChip-DIA enables profiling of 1160 protein groups from triplicate analysis of three individual single mammalian cells”.

7. Where average coverage is reported (e.g., line 230), the authors should indicate how many replicates this average is based on.

Reply: We thank the reviewer for pointing this out. All the average coverage results throughout this manuscript are based on triplicate (n=3) measurement analysis along with standard deviation, as per standard identification reporting manner. In response to the comment, we have added “triplicate” wherever average identification coverage has been reported throughout the manuscript.

8. The ability to directly image cells before analysis (line 235) is common to most other single cell proteomics platforms as well, so this is not an actual advantage.

Reply: We thank the reviewer for the comment. We agree with the reviewer that imaging cells is also possible in other single cell proteomics platforms. One difference we would like to note is that horizontally well-arranged arrays of cell traps in iProChip offer easier imaging assessment and analysis of all trapped cells, which is in contrast to cells being randomly distributed in a three-dimensional droplet that may sometimes be difficult to image directly.

9. In the abstract and elsewhere, the authors refer to 5 orders of proteome coverage. Do they mean dynamic range?

Reply: Yes, the 5 order of magnitude coverage refers to the dynamic range in protein abundances across the cell number and corresponding identified proteins.

10. On page 2, line 42, the statement “which theoretically extends to the single cell sensitivity” is an outdated statement since single-cell MS proteomics is now well established with many papers published.

Reply: We thank the reviewer for pointing out this and our apology for the outdated statement. In response, we have revised the text (Page 2 and Line 45-46): “*which has been shown in several studies to reach the single cell sensitivity*”, and added references on previous single cell proteomics studies.

11. The coverage reported for nanoPOTS on p. 3 of 1517 identified for ~10 cells are inaccurate. 1517 identifications include only those identified by MS/MS and excludes those identified by library matching using match between runs. The number of proteins identified is actually 3092. The authors could potentially argue against including identifications based on library matching, but since their DIA method also relies on library matching, this would be disingenuous.

Reply: We thank the reviewer for the correction on the data reported in the literature. We have revised the text (Page 3 and Line 60-62) as “...(*nanoPOTS*) achieved a proteome coverage of over 3000 proteins from 10-100 cells by incorporating match-between-runs (MBR) feature.” to acknowledge that 3092 proteins were identified in ~10 cells using nanoPOTS by match between runs (MBR).

12. Similarly, the number of proteins identified on average per single cell was 1475 in Ref. 21 when library matching was included, which should be acknowledged by the authors.

Reply: We thank the reviewer for the correction. In the revised manuscript, we have revised the text (Page 3 and Line 64) as “...reported sensitive profiling of 1475 protein groups with MBR from a single cell...” to acknowledge 1475 proteins were identified in a single cell when using MBR.

13. On p. 3, line 64, the statement should include the underlined text or something similar to recognize that not all microfluidic devices incorporate hydraulic actuations. “Microfluidic devices based on multilayer soft lithography use custom chip integration and hydraulic actuations to achieve precise μL -to-nL fluid manipulation”

Reply: We thank the reviewer for pointing this out and agree that not all microfluidic devices use hydraulic actuations for fluid/chip control and it is necessary to distinguish the device we introduced in this study. In the revised manuscript (Page 3, Line 70), we have updated the sentence kindly suggested by the reviewer.

14. On p. 3, line 75, I don't understand what this statement means: “enabled all retrospective peptides mapping against spectral libraries and offered superior coverage”. Please clarify.

Reply: We apologize if the description was not clear enough. As in DIA acquisition principle, the fragment spectra of all precursor ions detectable in the mass range of data acquisition are obtained, it is assumed to provide a permanent digital map. Then targeted signal extraction can be carried out by using the spectral library containing query peptide information. This approach allows data mining based on hypotheses using different query peptides of interest termed as retrospective here (*Mol Cell Proteomics* **11**, O111.016717, 2012).

REVIEWER COMMENTS

Reviewer #1 (Remarks to the Author):

Dear all,

In general, I think the authors did a great job of addressing my concerns, especially given the difficulty of some of the experiments involved using low cell numbers as input, and repeating the replicate numbers. I think with the new results, I am quite satisfied with the manuscript and the study.

Overall, I find this to be an important technology that demonstrates an important advance to the field, and recommend its publication. I have a few minor suggestions below, which the authors can decide if they wish to incorporate.

1. for the new section "Examination of cell usage efficiency of cell capturing chambers", I might change the heading to something more informative like "Examination of on-chip cell capture efficiency and validation of loading low concentration samples".
2. I think the discussion section would be enhanced greatly with a little bit more discussion about the current throughput (e.g. in comparison to single-cell genomics workflows/devices), and what the authors anticipate might help to improve this in the future to make the device more useful. Currently there is one sentence there, but I think a bit more discussion/future outlooks on this could excite the reader about the future possibility of more large-scale single-cell proteomic assays.

Finally, I hope that the authors will make the technology readily available for use by others, e.g. by providing detailed protocols and methods for constructing the device and operating it.

best,

Angela Wu

Reviewer #2 (Remarks to the Author):

In their revised manuscript, the authors have addressed several of the raised issues by describing the use and operation of the chip in more detail, and quantifying its efficiency in capturing cells. Yet a few questions remain:

1. They now describe in more detail that the number of captured cells is monitored visually, which seems technically doable however not very convenient in the long run, i.e. doing this sequentially for individual samples. For single cell analyses, the authors have developed a specific layout of the chip (Response Fig 7), however this figure is not included in the revised manuscript itself. This is highly unfortunate, since it would elevate the perceived impact of the system, presumably allowing unattended capture of single cells. Adding this figure would be a distinct improvement of the manuscript.

2. Data are now included to show the efficiency of cell capture (Response Fig 1 and Response Table 1). This is valuable information, however it also indicates that capture of low numbers of cells is quite inefficient (4% for single cells). In a typical scenario of a single cell experiment, usually multiple individual cells are to be investigated (e.g. many hundreds <https://www.biorxiv.org/content/10.1101/665307v3>). It remains to be seen if this can be conveniently done by the iProChip (9 samples in parallel), requiring 25x more cells as an input than are eventually analysed.

Another question that I raised under point 2 but that was not addressed (possibly because of unclear phrasing on my end), is in what scenarios one would actually need/want to analyse discrete numbers of cells (e.g. 100 cells, and not 99 or 101). Moreover, when working in this range, one would rather collect all cells that are available (e.g. 200), rather than wasting half to capture 100. This is a crucial question, since isolation of small number of cells (instead of single cells) seems to be the main application area of the iProChip. The authors should comment on this to place the use of the chip in the perspective of biological applications.

Overall, the iProChip is an ingenious device, and the revised manuscript now describes in more detail the metrics and numbers to better judge performance. In fact, some of the additional data that are now provided in a way expose some of its shortcomings (e.g. new Suppl Fig 4), however this is essential information for prospective users to take into account. It will be then up to the community and the test of time to determine if the iProChip will find broader acceptance, or if manufacture and

operational limitations present major hurdles. After all, several other alternative approaches are emerging, e.g. using single-cell dispensing systems capable to deposit hundreds of individual cells on conventional plates or glass slides, combined with subsequent sample prep using the same device – all of which is commercially available (e.g. <https://www.biorxiv.org/content/10.1101/2021.04.14.439828v1>)

Reviewer #3 (Remarks to the Author):

Concerns from the initial submission have gone largely unaddressed. The platform remains a significant engineering accomplishment and one that can clearly be used to prepare and analyze low-input proteomic samples including single cells, albeit with significant challenges. The work is certainly worth publishing and should sail through the review process at Lab on a Chip or Analytical Chemistry (as mentioned previously). However, to be worthy of publication in Nature Communications, the method should not only be different from other techniques, but also show significant performance advantages relative to existing ones. This work in no way sets the bar for throughput, proteome coverage, ease of use, etc. As an example, Zhu, Mechtler and Slavov groups have recently published methods that use the commercial CellenONE platform for single cell proteomics sample preparation. That platform can obtain an image of each cell being analyzed, isolate cells having desired characteristics based on brightfield and 4-color fluorescence at a rate of ~dozens of cells per minute, prepare them in a semiautomated fashion and readily interface with commercial autosamplers. The authors' platform cannot compete with that in any way other than perhaps cost of the liquid handler, and there is no path forward to doing so. Even the cost of that system is low compared to the labor and cleanroom requirements of the present work. Additional concerns are as follows:

1. Concerning exposed surface area, the authors compare their method to a 500 μ L vial. This is disingenuous, as nobody uses such large volumes for single cell proteomics. The authors should compare to techniques actually used for single cell proteomics. A better comparison was provided against one technique in the rebuttal but was omitted from the revised manuscript as it was not favorable. Also, the surface exposure in the lysis chamber is not the total surface to which these samples are exposed.
2. The fact that Ruedi Aebersold thought this approach was a good idea as stated in a commentary does not strengthen the authors' case, nor does the fact that Reviewers 1 and 2 said nice things about it. The current approach needs to stand on its own relative to existing designs, which it does not. In addition, it appears that the Aebersold group has abandoned their efforts using a very similar platform, likely due to the inherent limitations of needing to prepare a highly complex microfluidic device to analyze one or a few samples. If the authors need to spend 4 hours fabricating a device to make one disposable chip, the platform will never find a real world application.

3. The fact that multilayer soft lithography and microfluidics in general are extremely useful in many cases is not in dispute. The authors need to demonstrate that they have developed a platform having significant performance advantages for single cell proteomics, which they have not done.

To summarize, the authors provide no compelling reason that this work should be published in Nature Communications.

Point-by-point responses to Reviewers' comments

~~Highly~~ Streamlined single-cell proteomics by all-in-one chip and data-independent acquisition mass spectrometry

We thank all reviewers for the kind and constructive comments and suggestions to improve our manuscript. We have attempted to address the comments and questions both in the body of the revised manuscript, and in this point-by-point reply letter. We hope that our new result of SciProChip, together with all other revisions and explanations can adequately respond to the comments and further improve the manuscript. In addition to the point-by-point reply shown below, the revised text is also highlighted in two formats: "~~light gray font with strike line~~" indicating deleted text (for removing primacy words or for conciseness), and "**red font**" indicating newly added content in the revised manuscript (tracked version).

Reply text: blue

Revised text in main text/Method/SI

Reviewers' comments:

Reviewer #1 (Remarks to the Author):

In general, I think the authors did a great job of addressing my concerns, especially given the difficulty of some of the experiments involved using low cell numbers as input, and repeating the replicate numbers. I think with the new results, I am quite satisfied with the manuscript and the study.

Overall, I find this to be an important technology that demonstrates an important advance to the field, and recommend its publication. I have a few minor suggestions below, which the authors can decide if they wish to incorporate.

Reply: We thank the reviewer for appreciating our effort and new chip-based device for single cell proteomics. Based on the performance of iProChip and extended version of SciProChip dedicated for single cell capacity, we are excited to share this technology with the community.

1.1 For the new section "Examination of cell usage efficiency of cell capturing chambers", I might change the heading to something more informative like "Examination of on-chip cell capture efficiency and validation of loading low concentration samples".

Reply: We thank the reviewer for the suggestion. We have changed the heading accordingly.

In the revised manuscript, aforementioned heading has been changed to (Page 17, Line 604-605): "*Examination of on-chip cell capture efficiency and validation of loading low concentration samples*"

1.2 I think the discussion section would be enhanced greatly with a little bit more discussion about the current throughput (e.g. in comparison to single-cell genomics workflows/devices), and what the authors anticipate might help to improve this in the future to make the device more useful. Currently there is one sentence there, but I think a bit more discussion/future outlooks on this could excite the reader about the future possibility of more large-scale single-cell proteomic assays.

Reply: We thank the reviewer for the comment and suggestion. As pointed out by the reviewer, the throughput of microfluidics-based single cell genomics/ transcriptomics is now extended to hundreds or even thousands of single-cells. The single-cell proteomics is still in the developmental stage to further increase its multiplexity. In the revised version, we have added results on the newly introduced SciProChip, optimized for single cell capacity, and its greatly enhanced performance. Specifically, the multiplexity has been increased to from 3 to 20, with elevated proteome coverage from an average of 976 to 1500 protein groups (~1.53-fold increase) by using iProChip and SciProChip, respectively.

For future chip developments, assay throughputs can be increased by two directions: (1) incorporating more single-cell units in the SciProChip, which can be achieved by multiplexed input for valve control and compact chip design. (2) Introducing multiplexed isotopic labeling techniques such as TMT tagging with 4-16 plex ability, which has been demonstrated by other strategies such as the SCoPE-MS/SCoPE2. Altogether, it seems that assay throughput for single-cell proteomics will likely be elevated by integrated effort through both technology developments (e.g., nanoPOTS, SciProChip and others) and novel workflow implementations.

Following the comment, we have added the following statement in the discussion (Page 14, Line 469-473): *“High throughput ability is important to single cell characterization from a heterogeneous sample. The multiplexity of 20 single-cells in the presented platform remains to be further increased. Such throughput may be further increased either by incorporating more single-cell processing units, or by implementing protocols that allow multiplexed analysis such as TMT labeling.”*

Finally, I hope that the authors will make the technology readily available for use by others, e.g. by providing detailed protocols and methods for constructing the device and operating it.

Reply: We again thank the reviewer for appreciating this study. We have provided all the details of chip fabrication and proteomics workflow in the manuscript. We will follow guidelines to make relevant protocols and methods available, and we may prepare educational materials for the protocols in the future to help other scientists and researchers to set up and use this technology.

best,

Angela Wu

Reviewer #2 (Remarks to the Author):

In their revised manuscript, the authors have addressed several of the raised issues by describing the use and operation of the chip in more detail, and quantifying its efficiency in capturing cells. Yet a few questions remain:

2.1 They now describe in more detail that the number of captured cells is monitored visually, which seems technically doable however not very convenient in the long run, i.e. doing this sequentially for individual samples. For single cell analyses, the authors have developed a specific layout of the chip (Response Fig 7), however this figure is not included in the revised manuscript itself. This is highly unfortunate, since it would elevate the perceived impact of the system, presumably allowing un-attended capture of single cells. Adding this figure would be a distinct improvement of the manuscript.

Reply: We thank the reviewer for the comment and suggestion. Following the previous revision, we have optimized the design and performance of a new chip, termed single-cell iProChip (SciProChip) dedicated for 20 single cell capacity (Figure 6, as shown below). Compared to the iProChip, the SciProChip demonstrated an increased cell capture efficiency from ~4% to ~40% and enhanced proteome coverage from an average of 976 to 1500 protein groups.

The image you kindly suggested to include is now also added as the new Figure 6c. Additionally, in the revised manuscript, we have added a new section (Page 12, Line 400-428) entitled “Enhanced sensitivity by single-cell integrated proteomic chip (SciProChip) and DIA MS” to describe its design, chip characteristics, and analytical merits for single cell proteomics analysis. The chip design and new features are summarized as follows:

Figure 6. Single-cell integrated proteomic chip (SciProChip) for 20-plex single cell proteomic analysis.

Higher Cell Usage Efficiency: The design of the 20 operational units in SciProChip is similar to the iProChip except their reduced sizes. Importantly, the single cell capturing chambers now allow unattended single cell capture. Notably, the cell usage efficiency in SciProChip has improved ~10-fold (from ~4% to ~40%) compared to iProChip (Fig. 6c and d), which is mainly due to improved positioning of twin pillars and narrower dimensions of the chamber. In the digestion chamber, we kept their heights at 25 μm (compared to 100 μm in iProChip), reducing total processing volume to 25% (from 312 to 78.5 nL). In addition, the length of the C18-packed columns was reduced from 2.5

cm to 1 cm. All these modifications are expected to help reduce sample loss.

Figure 6c and d. (c) A bright-field image of a capturing chamber with a trapped cell. Scale bar: 120 μm. (d) The cell usage efficiency of SciProChip. Error bars: s.d. ($n = 28$ measurements).

Increased Proteome Coverage For compatibility of small cell input, the LC-MS/MS gradient time for DIA-MS was reduced to 90 mins. The combined contribution from the modified chip and optimization in the DIA LC-MS greatly facilitated the increased sensitivity of our approach with an average protein identification from 976 proteins obtained by iProChip (triplicate runs) to ~1500 across 20 replicates of single cell analysis.

2.2 Data are now included to show the efficiency of cell capture (Response Fig 1 and Response Table 1). This is valuable information, however it also indicates that capture of low numbers of cells is quite inefficient (4% for single cells). In a typical scenario of a single cell experiment, usually multiple individual cells are to be investigated (e.g. many hundreds <https://www.biorxiv.org/content/10.1101/665307v3>). It remains to be seen if this can be conveniently done by the iProChip (9 samples in parallel), requiring 25x more cells as an input than are eventually analysed.

Reply: We thank the reviewer for the important feedback. As mentioned in **Reply 2.1**, in the revised manuscript we have developed SciProChip, which is dedicated for single cell capacity. Note that SciProChip exhibits better positioned single-cell capturing pillars and narrower chamber dimensions, both of which can facilitate efficient single-cell capture. Using SciProChip, our result demonstrated an improved cell usage efficiency of ~40% for capturing single-cells, which is ~10-fold enhancement than that of iProChip (Figure 6c and d shown above). Thus, on average, about 2-3 cells are required to capture a single cell using SciProChip.

It is also foreseeable that such single-cell usage efficiency may be further enhanced by collecting cell flow-through and re-inject if desired. Additionally, since this is the first report of microfluidics-based single-cell proteomics, in the present study our major effort was devoted into developing an integrated device for streamlined and sensitive proteomics workflow. Therefore, current throughput of such assay is at a level tens of single-cells. Though the current throughput is designed for 20-plex parallel processing, chip design is flexible and can be custom designed to incorporate more single cell units for higher assay throughput. It is also possible to increase throughput by implementing protocols that allow multiplexed analysis, such as by TMT labeling.

In response to this comment, we have added the following text: (1) (Page 12, Line 403-405) “*This SciProChip was designed to include 20 chambers with each containing a single-cell trap to facilitate precise and unattended capture of one cell for proteomic processing (Fig. 6c).*”, (2) (Page 12, Line 405-407) “*The chip showed improved cell usage efficiency of ~40% for single-cell capture by optimal positioning of the cell traps and the narrower dimension of the chamber (Fig. 6d and Methods).*”, (3) (Page 14, Line 469-473) “*High throughput ability is important to single cell characterization from a heterogeneous sample. The multiplexity of 20 single-cells in the presented platform remains to be further increased. Such throughput may be further increased either by incorporating more single-cell processing units, or by implementing protocols that allow multiplexed analysis such as TMT labeling*”, and (4) added a new Figure 6c and 6d in the revised manuscript.

2.3 Another question that I raised under point 2 but that was not addressed (possibly because of unclear phrasing on my end), is in what scenarios one would actually need/want to analyse discrete numbers of cells (e.g. 100 cell, and not 99 or 101). Moreover, when working in this range, one would rather collect all cells that are available (e.g. 200), rather than wasting half to capture 100. This is a crucial question, since isolation of small number of cells (instead of single cells) seems to be the main application area of the iProChip. The authors should comment on this to place the use of the chip in the perspective of biological applications.

Reply: We thank the reviewer for this important question and apologize for the oversight in the previous revision.

Proteome Coverage vs Cell Number: The current sensitivity of single cell proteomics to identify <1000-2000 proteins is limited to reach the disease-specific proteins or pathways that are usually very low abundant. To study the cellular or phenotypic status to provide biological insights, sufficient proteome coverage is critical and may require higher cell numbers due to the limited proteome coverage at single cell level. Thus, we have developed two chips with capacities for analyzing different numbers of cells, from a single cell (SciProChip) to 10-100 cells (iProChip) to allow different experiments that may require different cell inputs and sufficient proteomics depth to explore cell biology. Importantly, the number of cells to be captured and analyzed does not need to be “discrete”. With the imaging functionality, *in-situ* cell counting can quantify the number of cells. Thus, cell number can be variable to achieve the proteome coverage of interest. The iProChip is not designed to specifically analyze only discrete numbers of cells (such as 100, and not 99 or 101).

For example, as evident from our data, proteomics coverage of key proteins and biomarkers in the lung cancer and B cell signaling pathway is different upon processing different numbers of cells (1-100) (Fig. 4f and 5b). For a particular application, such as in mass-limited clinical samples, the protein-of-interest may only be detected at 10-cell or 50-cell levels, thus it is essential to perform proteomics with such input cell-level.

Assay Performance at Different Cell Input: For developing a new workflow for analyzing proteomic coverage of very low-input cell samples, it is crucial we first understand the assay resolution and characterize all relevant analytical performances.

The design of different number cells (1-100) can provide systematic characterization of proteomics performance at different numbers of input cells. In the current results of proteomic profiling of 1-100 cells, we have acquired a good understanding of the proteome coverage that can be robustly detected of different input cells.

In response to the comment, we have included following text in the manuscript:

Introduction (Page 3, Line 79-88): *“To study the cellular or phenotypic status, sufficient proteome coverage is critical and may require higher cell numbers due to relatively limited proteome coverages at single cell level. Therefore, in this study, chips with different cell capacity were constructed to facilitate experiments with optimal profiling depth for different cell inputs. Specifically, an integrated proteomics chip (iProChip, 1-100 cells) and its extended version for single-cell capacity (SciProChip) were designed and coupled with data-independent acquisition (DIA) MS as streamlined nanoproteomics (nanogram of cells) pipelines.”*

Discussion (Page 13, Line 433-438) *“Despite tremendous efforts, hundreds to approximately one-two thousands of proteome coverage is limited to reach disease-specific proteins or pathways that are usually very low abundant. Thus, two chips were designed for different capacities from a single cell (SciProChip) to 10-100 cells (iProChip) to allow experiments that may require different cell inputs for sufficient proteomic depth to explore cell biology.”*

Overall, the iProChip is an ingenious device, and the revised manuscript now describes in more detail the metrics and numbers to better judge performance. In fact, some of the additional data that are now provided in a way expose some of its shortcomings (e.g. new Suppl Fig 4), however this is essential information for prospective users to take into account. It will then be up to the community and the test of time to determine if the iProChip will find broader acceptance, or if manufacture and operational limitations present major hurdles. After all, several other alternative approaches are emerging, e.g. using single-cell dispensing systems capable to deposit hundreds of individual cells on conventional plates or glass slides, combined with subsequent sample prep using the same device – all of which is commercially available (e.g. <https://www.biorxiv.org/content/10.1101/2021.04.14.439828v1>)

Reply: We thank the reviewer for appreciating our effort and constructive comments to further improve our platform and present both pros and cons of the new device to the community. We also thank the reviewer for mentioning alternative approaches using single-cell dispensing systems such as the CellenONE platform for proteomics workflow as shown in the biorxiv manuscript, which indicates the promise and need of single cell proteomics. Indeed, its utility and impact can already be observed from several new and on-going studies by prominent researchers in the field. We are indeed glad to see more promising technologies are being developed for single cell proteomics research. With a newly introduced SciProChip in the revised manuscript, we have further shown advantages of the enclosed chamber-based proteomics workflow for single cell analysis, including (1) greatly improved single cell proteomics coverage (now ~1500 protein groups/cell) with high reproducibility, and (2) substantially improved cell usage efficiency (46% versus 4% in iProChip). With the expandable and flexible cell experiment in microfluidics devices, we also believe that

the scientific community shall benefit from more (rather than less) choices for nanoproteomics and single cell proteomics, so the pursuit of a promising, alternative strategy that can complement existing technologies should be encouraged.

Reviewer #3 (Remarks to the Author):

Concerns from the initial submission have gone largely unaddressed. The platform remains a significant engineering accomplishment and one that can clearly be used to prepare and analyze low-input proteomic samples including single cells, albeit with significant challenges. The work is certainly worth publishing and should sail through the review process at Lab on a Chip or Analytical Chemistry (as mentioned previously). However, to be worthy of publication in Nature Communications, the method should not only be different from other techniques, but also show significant performance advantages relative to existing ones. This work in no way sets the bar for throughput, proteome coverage, ease of use, etc. As an example, Zhu, Mechtler and Slavov groups have recently published methods that use the commercial CellenONE platform for single cell proteomics sample preparation. That platform can obtain an image of each cell being analyzed, isolate cells having desired characteristics based on brightfield and 4-color fluorescence at a rate of ~dozens of cells per minute, prepare them in a semiautomated fashion and readily interface with commercial autosamplers. The authors' platform cannot compete with that in any way other than perhaps cost of the liquid handler, and there is no path forward to doing so. Even the cost of that system is low compared to the labor and cleanroom requirements of the present work.

Reply: We thank the reviewer for the comments that significant improvements are critical to complement existing single cell proteomics workflows. We also thank the reviewer for mentioning the commercial CellenONE platform, which indicates the promise and need of single cell proteomics. Its utility and impact can also be clearly seen from several new and on-going studies by prominent researchers in the field. We are indeed very happy to see more promising technologies are being rapidly developed for single cell proteomics research. In the previous revision, we have attempted to address all concerns raised by the reviewer in either point-by-point written responses or performing additional experiments and analysis. With the expandable and flexible cell experiment in microfluidics devices, we aim to share an alternative strategy that can complement existing technologies and offer more choices for nanoproteomics and single cell proteomics in the community.

In this revised manuscript, we have introduced an extended version of a single-cell proteomics chip (termed SciProChip), and demonstrated its enhanced performance for single cell proteomics analysis (Fig. 6 and Page 12, Line (400-428) in the main text). In the current form, SciProChip allows unattended single cell capture and displays substantially 10-fold improved cell usage efficiency from 4% in iProChip to ~40% for single-cell capture, without the need to use the external cell sorter. Furthermore, new results showed that SciProChip-DIA workflow confidently identified 1324-1843 (on average $\sim 1500 \pm 131$) protein groups across 20 single cells from two batches of culture

using two independent SciProChips, which showed good reproducibility both within analyzed single-cells and between two experimental runs. In terms of numbers of protein groups identified, SciProChip has outperformed iProChip by ~50% while maintaining all good analytical merits including high overlap between single-cell runs (81%-86%), good analytical reproducibility (Pearson correlation 0.82-0.96) and consistent protein quantitation across all single-cells. To put into broader perspective, the performance of SciProChip-DIA workflow is among one of the most sensitive label-free single cell proteomics strategies with rather complete all-in-one features. Furthermore, the design and construction of microfluidics-based iProChip and SciProChip are conceptually different from existing single cell proteomic methods and exhibit complementary features to existing ones (e.g., accurate cell counting, cell imaging, spares the need for the external sorter and contains built-in desalting columns). The design of chips is versatile and can be readily prototyped with multiplexity, such as the extended SciProChip. Looking forward, it is possible to add additional features such as introducing affinity tag or surface marker-based cell sorting and signaling manipulation before the proteomics workflow starts.

Figure 6. Single-cell integrated proteomic chip (SciProChip) for 20-plex single cell proteomic analysis.

In summary, these established results inspired us to eagerly share the iProChip and SciProChip to the single cell proteomics community. Given the importance of single cell proteomics, we feel efforts putting into developing a new approach which can offer complementary features with noticeable performance to existing methods is worthwhile. From the methodology development point of view, the rationale of the

submitted manuscript is to introduce a conceptually distinct platform with good analytical capability as well as readily extendable functionality in cell manipulation and assay. The device design, fabrication and analytical merits were fully explored and discussed in the manuscript. We hope that the revised manuscript and point-by-point reply can adequately show that our work is an alternative to complement other similar tools, and NOT neglecting any previous efforts.

Additional concerns are as follows:

3.1 Concerning exposed surface area, the authors compare their method to a 500 μ L vial. This is disingenuous, as nobody uses such large volumes for single cell proteomics. The authors should compare to techniques actually used for single cell proteomics. A better comparison was provided against one technique in the rebuttal but was omitted from the revised manuscript as it was not favorable. Also, the surface exposure in the lysis chamber is not the total surface to which these samples are exposed.

Reply: We thank the reviewer for the comment and suggestion. It is our oversight that this comparison to nanoPOTS was not included in the previous revision, since we thought iProChip and nanoPOTS have fundamentally different design principles. As per the reviewer's suggestion, the details of the calculated volumes and contact surface areas (including all exposed surfaces such as lysis and digestion chambers and the connecting channels) of iProChip, the extended SciProChip, and nanoPOTS can be found in the updated Supplementary Table 1 in the revised manuscript.

Supplementary Table 1. Comparison of surface area and volume in iProChip, SciProChip and nanoPOTS. Note that connecting channel refers to the connection channel between the cell capturing chamber and the digestion vessel.

Device	iProChip					SciProChip			nanoPOTS
Module	Cell capturing chamber			Digestion chamber	Connecting channel*	Cell capturing chamber	Digestion chamber	Connecting channel*	well
No. cells	10	50	100	-	-	1	-	-	1
Volume (nL)	5	8.5	11	312	1.8	2.1	78.5	1.8	200
Surface area (mm ²)	0.46	0.81	6.4	6.9	0.18	0.2	6.4	0.18	0.8
Surface-to-volume ratio	0.09	0.09	0.02	0.02	0.1	0.09	0.08	0.1	0.004

We fully agree with the Reviewer that the contact surface area of iProChip with an enclosed-chamber geometry is larger than the nanoPOTS platform. Yet given that all the modules are concatenated within the same device (no need for performing sample transfer among different steps) and that almost the entire chip (except the SPE column) was coated with BSA, our data showed that sample loss due to surface adsorption is still substantially reduced by 90% compared to most conventional workflows. It is also noted that enclosed systems have several inherent advantages, such as being less prone to air impurities that are advantageous for handling clinical samples, high flexibility for automation, parallelization, and integration of lab routines in a single device.

In response to this comment, we have (1) added updated Supplementary Table S1 to summarize the comparison of iProChip and SciProChip to nanoPOTS and (2) revised the manuscript accordingly (Page 4, Line 124-127): “*Note that the calculated surface-to-volume ratio for iProChip is larger than that of existing single-cell devices, such as nanoPOTS, yet it still exhibits substantially reduced surface area by >90% in comparison to microscale vial-based workflow (Supplementary Table 1)*”

3.2 The fact that Ruedi Aebersold thought this approach was a good idea as stated in a commentary does not strengthen the authors’ case, nor does the fact that Reviewers 1 and 2 said nice things about it. The current approach needs to stand on its own relative to existing designs, which it does not. In addition, it appears that the Aebersold group has abandoned their efforts using a very similar platform, likely due to the inherent limitations of needing to prepare a highly complex microfluidic device to analyze one or a few samples. If the authors need to spend 4 hours fabricating a device to make one disposable chip, the platform will never find a real world application.

Reply: We thank the reviewer for the comments and reminder on the potential limitation of a microfluidics-based platform. We hope the following summary can better clarify the pros and cons of our platform.

- (1) Firstly, we would like to clarify that the concept and design of constructing integrated chips for proteomics analysis was BY NO MEANS following Prof. Ruedi Aebersold’s idea or commentary when we initiated this study in a thematic program funded in 2018. We were not aware of that commentary until later on. Meanwhile, we do not have the capacity to know any progress of Prof. Ruedi Aebersold’s group and whether they have abandoned the pursuit of such an approach. We quoted his commentary on *Nature Methods* article mainly to share and value his vision in encouraging the development of alternative methods and technologies, and that research endeavors putting into exploration of new methods/technology for addressing key research needs can be encouraged as long as it is scientifically sound and solid, which we believe this study has demonstrated.
- (2) In the previous manuscript, we had shown that by coupling integrated proteomic chip and data-independent acquisition MS workflow, such a fundamentally different platform can achieve among one of most sensitive proteomics profiling at 1-100 cells with various advantageous features including high reproducibility, good input-dependent linearity for protein quantification and low missing values. Thanks to the comments/suggestions from all Reviewers, in this revision, we have further extended the design of iProChip to introduce SciProChip dedicated for multiplexed single cell proteomics analysis. This SciProChip not only inherits key features of iProChip, such as streamlined operation and all-in-one functionality, but also has been updated to allow unattended single-cell capture, much improved cell usage efficiency (from ~4% in iProChip to ~40%). Most importantly, its proteomics profiling coverage was substantially enhanced to confidently identify an average of ~1500 protein groups with high reproducibility across 20 individual single-cells using two different batches of cell culture and 2 different chips.
- (3) Like other newly developed platforms, the iProChip/SciProChip does have

limitations in demand of a trained researcher for making devices and rather low-throughput. Although extending throughput is out of the scope of the current study, nevertheless, the assay throughputs can be further increased by incorporating more single-cell units. This can be achieved by (1) design of multiplexed valve control and (2) compatible chip space to accommodate more single-cell units. Meanwhile, the assay throughput can be enhanced by introducing multiplexed labeling techniques such as isobaric labeling, which has been demonstrated by other single cell proteomics strategies such as the SCoPE-MS/SCoPE2. Altogether, assay throughput for single-cell proteomics will likely be elevated by integrated effort through both technology developments (e.g., nanoPOTS, SciProChip and others) and novel workflow implementations.

3. The fact that multilayer soft lithography and microfluidics in general are extremely useful in many cases is not in dispute. The authors need to demonstrate that they have developed a platform having significant performance advantages for single cell proteomics, which they have not done.

Reply: We thank the reviewer for the comment and agree that multilayer soft lithography and microfluidics are useful platforms for various applications. Yet its implementation for streamlined and sensitive proteomics analysis at mass-limited input samples has remained underexplored. Thus, we aim to extend the microfluidics chips as a miniaturized proteomic processor for single cell proteomics application. With the demonstrated single cell proteomics profiling of ~1500 protein groups using SciProChip-DIA and various good analytical performance of using iProChip-DIA for analyzing 1-100 cells, we show that this integrated approach can realize streamlined and sensitive nanoproteomics and single-cell proteomics profiling. Meanwhile, since iProChip is the first generation, we expect that its flexible design will allow extendable functionality and enhanced performance. For instance, alternative cell isolation strategies such as affinity-based sorting to enrich sub-population of cells from a heterogeneous cell population can be integrated to realize multiplexed cell sorting directly on the chip, which is a feature not easily implementable using existing single cell proteomics platforms. In addition, although unexplored in present study, cell assays (for example, cell stimulation by delivering ligands) can be further incorporated through microfluidics operations before proteomics workflow takes place. Such flexibility in additional cell assay capacity is still currently lacking in most existing approaches.

To summarize, the authors provide no compelling reason that this work should be published in Nature Communications.

REVIEWERS' COMMENTS

Reviewer #2 (Remarks to the Author):

The authors have appropriately addressed my concerns, and especially the addition of the SciProChip and data generated with this single-cell version of the technology is a great improvement of the manuscript. Overall the paper brings a very interesting new technology to the field of single-cell proteomics with the potential for further development in the future to enhance throughput. I therefore recommend acceptance of the manuscript for publication.

Point-by-point responses to Reviewers' comments

Streamlined single-cell proteomics by ~~all-in-one~~ an integrated microfluidic chip and data-independent acquisition mass spectrometry

Reviewers' comments:

Reviewer #2 (Remarks to the Author):

The authors have appropriately addressed my concerns, and especially the addition of the SciProChip and data generated with this single-cell version of the technology is a great improvement of the manuscript. Overall the paper brings a very interesting new technology to the field of single-cell proteomics with the potential for further development in the future to enhance throughput. I therefore recommend acceptance of the manuscript for publication.

>> Reply: We thank the reviewer for the comment and all previous suggestions that really helped us to further improve our manuscript during the revision process. Based on the performance of the iProChip-DIA and SciProChip-DIA workflows, we are also excited to share this technology with the community.